# MIRROR DESCENT-ASCENT FOR MEAN-FIELD MIN-MAX PROBLEMS

## ABSTRACT

We study two variants of the mirror descent-ascent algorithm for solving min-max problems on the space of measures: simultaneous and alternating. We work under assumptions of convexity-concavity and relative smoothness of the pay-off function with respect to a suitable Bregman divergence, defined on the space of measures via flat derivatives. We show that the convergence rates to mixed Nash equilibria, measured in the Nikaidò-Isoda error, are of order $\mathcal{O}\left(N^{-1/2}\right)$ and $\mathcal{O}\left(N^{-2/3}\right)$ for the simultaneous and alternating schemes, respectively, which is in line with the state-of-the-art results for related finite-dimensional algorithms.

## 1 INTRODUCTION

Numerous tasks in machine learning can be framed as optimization problems for functions defined on the space of probability measures. For instance, in supervised learning, pioneering works (Chizat & Bach, 2018; Mei et al., 2018; Rotskoff & Vanden-Eijnden, 2018) showed that training a shallow neural network (NN) in the mean-field regime (i.e., an infinite-width one-hidden-layer NN) can be viewed as minimizing a convex function over the space of probability distributions of the parameters of the network. This key insight proved to be a fruitful approach in analyzing convergence of training algorithms for infinite-width one-hidden-layer NNs (see, e.g., (Hu et al., 2021; Chizat, 2022a; Nitanda et al., 2022; Suzuki et al., 2023)).

The paradigm of mean-field optimization has been extended to min-max settings in several works, e.g., (Hsieh et al., 2019; Domingo-Enrich et al., 2020; Wang & Chizat, 2023; Lu, 2023; Trillos & Trillos, 2023; Kim et al., 2024), which formulate the training of Generative Adversarial Networks (GANs) and adversarial robustness as a problem of finding mixed Nash equilibria (MNEs) of min-max games over the space of probability measures.

In this work, we study the convergence of an infinite-dimensional mirror descent-ascent algorithm (MDA) to mixed Nash equilibria of a min-max game with a convex-concave payoff function. In games, the design of learning algorithms heavily depends on the playing conventions the players can adopt: simultaneous (players move at the same time) or alternating (each player moves upon observing the opponents' moves). To our knowledge, the works concerned with studying the convergence of discrete-time algorithms for mean-field min-max games only analyze the case of simultaneous playing (see, e.g., (Hsieh et al., 2019; Wang & Chizat, 2023)). In contrast, we make a rigorous comparison between the simultaneous and alternating algorithms, and prove that alternating playing leads to faster convergence rate. This result theoretically underpins the common practice of training GANs in an alternating fashion. Moreover, we demonstrate that our framework extends naturally to additional settings, including two-player zero-sum Markov games (Example E).

### 1.1 NOTATION AND SETUP

For any $\mathcal{X} \subseteq \mathbb{R}^d$, let $\mathcal{P}(\mathcal{X})$ denote the set of probability measures on $\mathcal{X}$. In game theory, if $\mathcal{X}$ is the set of *(pure) strategies* available to the players, then $\mathcal{P}(\mathcal{X})$ is known as the set of *mixed strategies*. Let $\mathcal{C}, \mathcal{D} \subset \mathcal{P}(\mathcal{X})$ be nonempty and convex. We consider a convex-concave (cf. Assumption 1.5) payoff function $F : \mathcal{C} \times \mathcal{D} \to \mathbb{R}$ and the associated min-max game

$$\min_{\nu \in \mathcal{C}} \max_{\mu \in \mathcal{D}} F(\nu, \mu). \tag{1}$$

We are interested in finding *mixed Nash equilibria* (MNEs) for game (1), i.e., pairs of strategies $(\nu^*, \mu^*) \in \mathcal{C} \times \mathcal{D}$ such that, for any $(\nu, \mu) \in \mathcal{C} \times \mathcal{D}$, we have

$$F(\nu^*, \mu) \leq F(\nu^*, \mu^*) \leq F(\nu, \mu^*). \tag{2}$$

We observe that in the case in which $F$ is bilinear, i.e., $F(\nu, \mu) = \int_{\mathcal{X}} \int_{\mathcal{X}} f(x, y)\nu(\mathrm{d}x)\mu(\mathrm{d}y)$, for some $f : \mathcal{X} \times \mathcal{X} \to \mathbb{R}$, measures characterized by (2) are MNEs in the classical sense of two-player zero-sum games. Throughout, we assume that there exists at least one MNE for game (1).[1]

In min-max games, the distance between a pair of strategies $(\nu, \mu)$ and an MNE is typically measured using the Nikaidò-Isoda (NI) error (Nikaidô & Isoda, 1955), which, for all $(\nu, \mu) \in \mathcal{C} \times \mathcal{D}$, is defined by

$$\mathrm{NI}(\nu, \mu) := \max_{\mu' \in \mathcal{D}} F(\nu, \mu') - \min_{\nu' \in \mathcal{C}} F(\nu', \mu).$$

Straight from the definition, we see that $\mathrm{NI}(\nu, \mu) \geq 0$ for all $(\nu, \mu) \in \mathcal{C} \times \mathcal{D}$, and from (2) it follows that $\mathrm{NI}(\nu, \mu) = 0$ if and only if $(\nu, \mu)$ is an MNE.

## 1.2 Example: Training of GANs

Let $\hat{\xi} \in \mathcal{P}(\mathcal{Y})$ be the empirical measure of the i.i.d. sampled particles $\{x_i\}_{i=1}^{M} \subset \mathcal{Y}$, and let $\xi \in \mathcal{P}(\mathcal{Z})$ be a source measure. Consider the measurable parametrized transport map $T_\theta : \mathcal{Z} \to \mathcal{Y}$ (which typically can be viewed as a neural network with parameters $\theta \in \Theta \subset \mathbb{R}^d$). The *pushforward* of the measure $\xi$ on $\mathcal{Z}$ via $T_\theta$ is the measure $T_\theta \# \xi$ on $\mathcal{Y}$ characterized by $\int_{\mathcal{Y}} \varphi \mathrm{d}(T_\theta \# \xi) = \int_{\mathcal{Z}} (\varphi \circ T_\theta) \mathrm{d}\xi$, for any measurable function $\varphi : \mathcal{Y} \to \mathbb{R}$.

The aim of a GAN is to search for the optimal set of parameters $\theta^* \in \Theta$ that minimizes the distance between the generated measure $T_{\theta*} \# \xi$ and the empirical measure $\hat{\xi}$. In order to evaluate this distance, we define the function $D_w : \mathcal{Y} \to \mathbb{R}$ (which can also be viewed as a neural network with parameters $w \in \mathcal{W} \subset \mathbb{R}^d$), and solve the min-max problem

$$\min_{\theta \in \Theta} \max_{w \in \mathcal{W}} \left\{ \int_{\mathcal{Y}} D_w(y) \left( T_\theta \# \xi - \hat{\xi} \right) (\mathrm{d}y) \right\}.$$

For example, if the family of functions $\{D_w\}_{w \in \mathcal{W}}$ is either 1-Lipschitz continuous or uniformly bounded, the resulting GAN corresponds to the Wasserstein GAN or the Total Variation GAN, respectively (Arjovsky et al., 2017). On the other hand, if the family of functions $\{D_w\}_{w \in \mathcal{W}}$ belongs to the norm unit ball of a reproducing kernel Hilbert space (RKHS), we recover the Maximum Mean Discrepancy (MMD) GAN (Li et al., 2017).

Solving this problem on the finite-dimensional subspaces $\theta, w \subset \mathbb{R}^d$ may pose serious challenges such as the lack of existence of pure Nash equilibria. Instead, we lift the problem to the space of probability measures and search for MNEs, i.e., optimal distributions over the set of parameters. That is, by setting $f(\theta, w) := \int_{\mathcal{Y}} D_w(y) \left( T_\theta \# \xi - \hat{\xi} \right) (\mathrm{d}y)$, we solve the mean-field min-max game

$$\min_{\nu \in \mathcal{P}(\Theta)} \max_{\mu \in \mathcal{P}(\mathcal{W})} \left\{ \int_{\mathcal{W}} \int_{\Theta} f(\theta, w)\nu(\mathrm{d}\theta)\mu(\mathrm{d}w) \right\}. \tag{3}$$

We will demonstrate theoretically (cf. Theorem 2.1 and Theorem 3.6) that alternating updates speed up GANs training significantly. Note that the lifted problem is bilinear in $\nu$ and $\mu$, so an MNE for (3) exists under mild conditions (see footnote 1).

We stress, however, that our framework applies more broadly, and, while encompassing (3) as a special case, covers also more general nonlinear convex-concave functions $F$. A natural occurrence of a nonlinear $F$ is illustrated in Example D. Another setting where such nonlinearity appears is in two-player zero-sum Markov games, discussed in Example E.

---

[1] If $F$ is continuous and $\mathcal{D}$ is compact, then the existence of an MNE of (1) follows from Sion's minimax theorem (Sion, 1958). For the particular case when $F(\nu, \mu) = \int_{\mathcal{X}} \int_{\mathcal{X}} f(x, y)\nu(\mathrm{d}x)\mu(\mathrm{d}y)$, an MNE exists due to Glicksberg's minimax theorem (Glicksberg, 1952) if $f$ is continuous and $\mathcal{C}, \mathcal{D}$ are compact.

### 1.3 RELATED WORKS

Mirror descent (MD) was originally proposed in (Nemirovski & Yudin, 1983) for solving convex optimization problems and has been extensively studied on finite-dimensional vector spaces, see e.g. (Beck & Teboulle, 2003; Bubeck, 2015; Lu et al., 2018). One of its main advantages over traditional gradient descent is that, by utilizing Bregman divergence as a regularization term instead of the usual squared Euclidean norm, the MD method captures the geometry of the ambient space better than the gradient descent scheme (see (Beck & Teboulle, 2003) for a detailed discussion).

Recently, the MD algorithm has been extended to infinite-dimensional settings for studying optimization problems over spaces of measures, with applications in machine learning (e.g., Sinkhorn's and Expectation–Maximization algorithms, see (Aubin-Frankowski et al., 2022)) as well as in policy optimization for reinforcement learning (Tomar et al., 2021; Kerimkulov et al., 2025a).

By leveraging results from optimization on $\mathbb{R}^d$ (see (Bauschke et al., 2017; Lu et al., 2018)), the work of (Aubin-Frankowski et al., 2022) extends the convergence proof from (Lu et al., 2018) to the case of the infinite-dimensional MD method by showing that in order for the MD procedure to converge with rate $\mathcal{O}\left(N^{-1}\right)$, it suffices to require convexity and relative smoothness of $F$ (cf. Assumptions 1.5 and 3.3, respectively).

Other works such as (Hsieh et al., 2019; Dvurechensky & Zhu, 2024) studied infinite-dimensional MDA and Mirror Prox algorithms for finding MNEs of two-player zero-sum games. The most closely related work to ours is (Hsieh et al., 2019), which focuses on min-max games for bilinear objective functions and utilizes a particular case of the MDA algorithm with relative entropy regularization.

Our paper generalizes the setting of (Hsieh et al., 2019) by considering a possibly non-linear convex-concave objective function and the MDA algorithm with a general Bregman divergence. Moreover, while (Hsieh et al., 2019) proves an explicit convergence rate $\mathcal{O}\left(N^{-1/2}\right)$ only for the simultaneous MDA algorithm, we also prove a faster convergence rate $\mathcal{O}\left(N^{-2/3}\right)$ for the alternating scheme. For a brief discussion on recent results on related Mirror Prox algorithms (not studied in the present paper), see Appendix K.

### 1.4 OUR CONTRIBUTION

We provide a theoretical analysis of the proposed simultaneous and alternating MDA algorithms, establishing convergence rates under convexity–concavity and relative smoothness of the objective $F$ with respect to a Bregman divergence. In particular, Theorem 2.1 and 3.6 show that the alternating MDA scheme achieves faster convergence than the simultaneous one. We validate our results on simple numerical experiments.

From one perspective, our work extends (Aubin-Frankowski et al., 2022) to the setting of min–max games. A key obstacle we overcome is that, unlike in single-player MD in both infinite-dimensional (cf. (Aubin-Frankowski et al., 2022)) and finite-dimensional (cf. (Lu et al., 2018)) settings, the objective function for min-max problems is not monotonically decreasing along the iterates, which forces us to work with the NI error and requires different proof techniques.

From another perspective, we generalize the results of (Hsieh et al., 2019) by considering a possibly non-linear convex–concave objective function and MDA algorithms with respect to a general Bregman divergence. Whereas (Hsieh et al., 2019) derive an explicit convergence rate only for the simultaneous MDA algorithm in the context of GAN training, we establish a faster rate for the alternating variant. Moreover, our more general framework also covers applications other than GANs, such as adversarial training of neural networks (Example D) and two-player zero-sum Markov games (Example E).

At the technical level, our convergence proof for alternating MDA relies on a duality between the Bregman divergence on the space of measures and a corresponding dual Bregman divergence defined on the space of bounded continuous functions. To our knowledge, the use of this dual formulation of MDA on a function space is novel, and may be of independent interest.

## 1.5 BREGMAN DIVERGENCE ON THE SPACE OF PROBABILITY MEASURES

As noted in Section 1.3, the MD algorithm relies on the Bregman divergence. We now introduce this concept rigorously for the space of probability measures using the flat derivative (Definition G.1), following (Aubin-Frankowski et al., 2022), who defined it via directional derivatives.

Set $\mathcal{E} := \mathcal{C} \cup \mathcal{D} \subset \mathcal{P}(\mathcal{X})$ and let $h : \mathcal{P}(\mathcal{X}) \to \mathbb{R}$ satisfy the following assumption.

**Assumption 1.1** (Differentiability and convexity of $h$). *Assume that $h \in \mathfrak{C}^1(\mathcal{E})$ (cf. Definition G.1). Moreover, assume that $h : \mathcal{P}(\mathcal{X}) \to \mathbb{R}$ is $\alpha$-strongly convex on $\mathcal{E}$ relative to the total variation (TV) distance, i.e., there exists $\alpha > 0$ such that for all $\varepsilon \in [0,1]$ and $\nu', \nu \in \mathcal{E}$, we have*

$$h\left((1-\varepsilon)\nu + \varepsilon\nu'\right) \leq (1-\varepsilon)h(\nu) + \varepsilon h(\nu') - \frac{\alpha}{2}\varepsilon(1-\varepsilon)\operatorname{TV}^2(\nu', \nu).$$

If Assumption 1.1 holds, then we show in Lemma B.2 that for any $\nu', \nu \in \mathcal{E}$, we have

$$h(\nu') - h(\nu) \geq \int_{\mathcal{X}} \frac{\delta h}{\delta \nu}(\nu, x)(\nu' - \nu)(\mathrm{d}x) + \frac{\alpha}{2}\operatorname{TV}^2(\nu', \nu). \tag{4}$$

Under Assumption 1.1, we define the $h$-Bregman divergence (or simply Bregman divergence) on the space of probability measures.

**Definition 1.2** (Bregman divergence). *The $h$-Bregman divergence is the map $D_h : \mathcal{E} \times \mathcal{E} \to [0, \infty)$ given by*

$$D_h(\nu', \nu) := h(\nu') - h(\nu) - \int_{\mathcal{X}} \frac{\delta h}{\delta \nu}(\nu, x)(\nu' - \nu)(\mathrm{d}x).$$

As we discussed above, Assumption 1.1 implies (4) due to Lemma B.2 and hence, straight from the definition of $D_h$, we obtain $D_h(\nu', \nu) \geq \frac{\alpha}{2}\operatorname{TV}^2(\nu', \nu)$, for all $\nu', \nu \in \mathcal{E}$, and $D_h(\nu', \nu) = 0$ if and only if $\nu' = \nu$.

We now give two examples of a function $h$ and the corresponding sets $\mathcal{E}$ such that Assumption 1.1 is satisfied. For other examples of functions $h$ that verify the inequality $D_h(\nu', \nu) \geq \frac{\alpha}{2}\operatorname{TV}^2(\nu', \nu)$ and hence Assumption 1.1, see (Chizat, 2022b, Lemma 3.2).

**Example 1.3** (Relative entropy). *Suppose that $h$ is the relative entropy, i.e., $h(m) := \int_{\mathcal{X}} \frac{m(x)}{\pi(x)} \log \frac{m(x)}{\pi(x)} \pi(x)\mathrm{d}x$, where $m, \pi \in \mathcal{P}_\lambda(\mathcal{X})$, i.e., they are absolutely continuous with respect to the Lebesgue measure on $\mathcal{X}$ and $\pi$ is a fixed reference probability measure on $\mathcal{P}_\lambda(\mathcal{X})$. Fix $\beta > 0$ and define $\mathcal{E}_\beta := \left\{ m \in \mathcal{P}_\pi(\mathcal{X}) : \left\| \log \frac{m(\cdot)}{\pi(\cdot)} \right\|_{L^\infty(\mathcal{X})} \leq \beta \right\}$. Let $\mathcal{C} = \mathcal{D} = \bigcup_{\beta > 0} \mathcal{E}_\beta$. Then $\mathcal{E} = \mathcal{C} \cup \mathcal{D}$ is convex. Moreover, it is proved in (Kerimkulov et al., 2025b, Proposition 2.17) that $h$ admits the flat derivative*

$$\frac{\delta h}{\delta m}(m, x) = \log \frac{m(x)}{\pi(x)} - h(m), \tag{5}$$

*on $\mathcal{E}$, and for all $m, m' \in \mathcal{E}$, the Bregman divergence $D_h(m', m)$ is in fact the Kullback-Leibler divergence (or relative entropy) $\operatorname{KL}(m', m)$. Therefore, by Pinsker's inequality, that is, $\frac{1}{2}\operatorname{TV}^2(m', m) \leq \operatorname{KL}(m', m)$, Assumption 1.1 holds with $\alpha = 1$.*

**Example 1.4** ($\chi^2$-divergence). *Suppose that $h$ is the $\chi^2$-divergence, i.e., $h(m) := \frac{1}{2}\int_{\mathcal{X}} \left(\frac{m(x)}{\pi(x)} - 1\right)^2 \pi(x)\mathrm{d}x$, where $m, \pi \in \mathcal{P}_\lambda(\mathcal{X})$. Let $L_\pi^2(\mathcal{X})$ be the set of square integrable functions on $\mathcal{X}$ with respect to $\pi$. Fix $\eta > 0$ and define $\mathcal{F}_\eta := \left\{ m \in \mathcal{P}_\pi(\mathcal{X}) : \left\| \frac{m(\cdot)}{\pi(\cdot)} \right\|_{L_\pi^2(\mathcal{X})} \leq \eta \right\}$. Let $\mathcal{C} = \mathcal{D} = \bigcup_{\eta > 0} \mathcal{F}_\eta$. Note that $\mathcal{F} = \mathcal{C} \cup \mathcal{D}$ is convex. Moreover, it is proved in (Kerimkulov et al., 2025b, Proposition 2.20) that $h$ admits the flat derivative*

$$\frac{\delta h}{\delta m}(m, x) = \frac{m(x)}{\pi(x)} - \int_{\mathbb{R}^d} \frac{m(x)}{\pi(x)} m(x)\mathrm{d}x,$$

*on $\mathcal{F}$, and for all $m, m' \in \mathcal{F}$, the Bregman divergence $D_h(m', m)$ is in fact the $L^2$-distance $\frac{1}{2}\left\| \frac{m'(\cdot)}{\pi(\cdot)} - \frac{m(\cdot)}{\pi(\cdot)} \right\|_{L_\pi^2(\mathcal{X})}^2$. Since $\pi \in \mathcal{P}_\lambda(\mathcal{X})$, the Cauchy-Schwarz inequality implies $\frac{1}{2}\operatorname{TV}^2(m', m) \leq D_h(m', m)$. Thus, Assumption 1.1 holds with $\alpha = 1$.*

## 1.6 SIMULTANEOUS AND ALTERNATING MDA

In what follows, we state our standing assumptions and the necessary definitions for introducing the simultaneous and alternating MDA schemes. Let $F : \mathcal{C} \times \mathcal{D} \to \mathbb{R}$ be such that $F(\cdot, \mu) \in \mathfrak{C}^1(\mathcal{C})$ and $F(\nu, \cdot) \in \mathfrak{C}^1(\mathcal{D})$ (cf. Definition G.1).

**Assumption 1.5** (Convexity-concavity of $F$). *Assume that $F$ is convex in $\nu$ and concave in $\mu$, i.e., for any $\nu, \nu' \in \mathcal{C}$ and $\mu, \mu' \in \mathcal{D}$, we have*

$$D_{F(\cdot,\mu)}(\nu', \nu) = F(\nu', \mu) - F(\nu, \mu) - \int_{\mathcal{X}} \frac{\delta F}{\delta \nu}(\nu, \mu, x)(\nu' - \nu)(\mathrm{d}x) \geq 0,$$

$$D_{F(\nu,\cdot)}(\mu', \mu) = F(\nu, \mu') - F(\nu, \mu) - \int_{\mathcal{X}} \frac{\delta F}{\delta \mu}(\nu, \mu, y)(\mu' - \mu)(\mathrm{d}y) \leq 0.$$

**Assumption 1.6** (Uniform boundedness of the flat derivatives of $F$). *Suppose that there exists $C_1 > 0$ and $C_2 > 0$ such that for all $\nu, \mu \in \mathcal{P}(\mathcal{X})$, and all $x, y \in \mathcal{X}$, we have*

$$\left| \frac{\delta F}{\delta \nu}(\nu, \mu, x) \right| \leq C_1, \quad \left| \frac{\delta F}{\delta \mu}(\nu, \mu, y) \right| \leq C_2.$$

Assumptions 1.5 and 1.6 are standard in the mean-field optimization literature, see, e.g., (Chen et al., 2023; Lascu et al., 2025).

In Proposition C.1 and D.1, we verify that Assumptions 1.5 and 1.6 are satisfied by Examples 1.2, D and E. In Lemma B.4, we show that Assumption 1.5 corresponds to the intuition we have from optimization on $\mathbb{R}^d$, where convexity (concavity) is equivalent to the Hessian of $F$ being non-negative (non-positive).

For a given stepsize $\tau > 0$, and fixed initial pair of strategies $(\nu_0, \mu_0) \in \mathcal{C} \times \mathcal{D}$, for $n \geq 0$, the *simultaneous* and *alternating* MDA algorithms are respectively defined by

---

**Algorithm 1:** SIMULTANEOUS MDA

**Input:** Objective function $F$, initial measures $(\nu^0, \mu^0)$, stepsize $\tau > 0$
**for** $n = 0, 1, \ldots, N-1$ **do**

$\quad \nu^{n+1} = \underset{\nu \in \mathcal{C}}{\arg\min}\{\int_{\mathcal{X}} \frac{\delta F}{\delta \nu}(\nu^n, \mu^n, x)(\nu - \nu^n)(\mathrm{d}x) + \frac{1}{\tau}D_h(\nu, \nu^n)\},$

$\quad \mu^{n+1} = \underset{\mu \in \mathcal{D}}{\arg\max}\{\int_{\mathcal{X}} \frac{\delta F}{\delta \mu}(\nu^n, \mu^n, y)(\mu - \mu^n)(\mathrm{d}y) - \frac{1}{\tau}D_h(\mu, \mu^n)\}$

**Output:** $\left( \frac{1}{N}\sum_{n=0}^{N-1} \nu^n, \frac{1}{N}\sum_{n=0}^{N-1} \mu^n \right)$

---

**Algorithm 2:** ALTERNATING MDA

**Input:** Objective function $F$, initial measures $(\nu^0, \mu^0)$, stepsize $\tau > 0$
**for** $n = 0, 1, \ldots, N-1$ **do**

$\quad \nu^{n+1} = \underset{\nu \in \mathcal{C}}{\arg\min}\{\int_{\mathcal{X}} \frac{\delta F}{\delta \nu}(\nu^n, \mu^n, x)(\nu - \nu^n)(\mathrm{d}x) + \frac{1}{\tau}D_h(\nu, \nu^n)\},$

$\quad \mu^{n+1} = \underset{\mu \in \mathcal{D}}{\arg\max}\{\int_{\mathcal{X}} \frac{\delta F}{\delta \mu}(\nu^{n+1}, \mu^n, y)(\mu - \mu^n)(\mathrm{d}y) - \frac{1}{\tau}D_h(\mu, \mu^n)\}$

**Output:** $\left( \frac{1}{N}\sum_{n=0}^{N-1} \nu^{n+1}, \frac{1}{N}\sum_{n=0}^{N-1} \mu^n \right)$

---

Although we abuse the notation by denoting the iterates in both Algorithm 1 and Algorithm 2 by $(\nu^n, \mu^n)_{n \geq 0}$, we will make it clear from the context which algorithm we consider.

Algorithm 1 is referred to as *simultaneous* because both players update their strategy from step $n$ to $n+1$ at the same time, whereas Algorithm 2 is called *alternating* because the minimizing player is first updating their move from step $n$ to $n+1$, and then the maximizing player is acting upon observing the minimizing player's $(n+1)$-th action. Note that due to the symmetry of the players,

the analysis of Algorithm 2 also covers the case when the maximizing player moves first followed by the minimizing player.

We observe that by varying the choices of $h$ in Definition 1.2 we obtain a collection of different update rules in Algorithms 1 and 2. When $h$ is the relative entropy, we can view Algorithm 1 and Algorithm 2 as Euler discretizations of a Fisher-Rao gradient flow, whose continuous-time convergence with explicit rates for mean-field min-max games was proved in (Lascu et al., 2024) (cf. also (Liu et al., 2023) for single-agent convex optimization).

**Remark 1.7** (Connection to the continuous-time gradient flow). *In Appendix J, we provide an implicit MDA algorithm that achieves a convergence rate of $\mathcal{O}(N^{-1})$, which matches the rate $\mathcal{O}(t^{-1})$ of the continuous-time gradient flow obtained by letting $\tau \to 0$. However, this scheme is not implementable in practice, as it requires each player to know the next move of their opponent at every iteration. Consequently, while it provides a useful theoretical benchmark, practical algorithms rely on explicit (simultaneous or alternating) mirror schemes.*

## 2 CONVERGENCE OF THE SIMULTANEOUS MDA ALGORITHM 1

In this section, we state the main result on the convergence of the simultaneous MDA algorithm.

**Theorem 2.1** (Convergence of the simultaneous MDA Algorithm 1). *Let $(\nu^0, \mu^0)$ be such that $\sup_{\nu \in \mathcal{C}} D_h(\nu, \nu^0) + \sup_{\mu \in \mathcal{D}} D_h(\mu, \mu^0) < \infty$. Suppose that Assumption 1.1, 1.5, 1.6 hold. If $\tau = \Theta\left(N^{-1/2}\right)$[2], then*

$$
\mathrm{NI}\left(\frac{1}{N}\sum_{n=0}^{N-1}\nu^n, \frac{1}{N}\sum_{n=0}^{N-1}\mu^n\right) \leq 2\sqrt{\frac{2\left(C_1^2 + C_2^2\right)\left(\sup_{\nu \in \mathcal{C}} D_h(\nu, \nu^0) + \sup_{\mu \in \mathcal{D}} D_h(\mu, \mu^0)\right)}{\alpha N}}.
$$

**Remark 2.2.** *Theorem 2.1 is consistent with the already known convergence rate $\mathcal{O}\left(N^{-1/2}\right)$ of the MDA algorithm for min-max games with strategies in compact convex subsets of $\mathbb{R}^d$; see e.g. (Bubeck, 2015, Theorem 5.1).*

**Remark 2.3** (Initialization condition). *The initialization requirement in Theorem 2.1, namely, $\sup_{\nu \in \mathcal{C}} D_h(\nu, \nu^0) + \sup_{\mu \in \mathcal{D}} D_h(\mu, \mu^0) < \infty$ must be verified case by case, depending on the choice of $h$ and the admissible classes $\mathcal{C}, \mathcal{D}$. Such verifications for Examples 1.3 and 1.4 are carried out in Lemmas B.8 and B.9, respectively.*

**Remark 2.4** (About the proof of Theorem 2.1). *In their proof of convergence of the infinite-dimensional MD algorithm for convex $F$, (Aubin-Frankowski et al., 2022) show that relative smoothness is sufficient to prove that $F$ is monotonically decreasing along the sequence $(\nu^n)_{n \geq 0}$ generated by MD, i.e., $F(\nu^{n+1}) \leq F(\nu^n)$, for all $n \geq 0$. The monotonicity property is key to establishing that the MD scheme converges to a minimizer of $F$ with rate $\mathcal{O}(N^{-1})$.*

*In the case of Algorithm 1, the monotonicity property no longer holds, and we therefore prove convergence only for the time-averaged iterates. The corresponding convergence rate is expected to be the slower rate $\mathcal{O}\left(N^{-1/2}\right)$, as suggested by (Bubeck, 2015, Theorem 5.1).*

*Furthermore, in contrast to the proof strategy in (Aubin-Frankowski et al., 2022), our argument does not require relative smoothness of $F$ (a condition involving bounds on its second-order flat derivatives, see Assumption 3.3 and Lemma B.4), but only uniform boundedness of its first-order flat derivatives. This is made possible by requiring $h$ to be strongly convex relative to the $\mathrm{TV}^2$ distance – an assumption stronger than the strict convexity of $h$ used in (Aubin-Frankowski et al., 2022), yet still verifiable for typical choices of divergences, see Examples 1.3 and 1.4.*

*The choice of the $\mathrm{TV}^2$ distance in Assumption 1.1 is motivated by the fact that several functions $h$, as illustrated in Examples 1.3 and 1.4 and noted in (Chizat, 2022b, Lemma 3.2), satisfy the inequality $D_h(\nu', \nu) \geq \frac{\alpha}{2}\mathrm{TV}^2(\nu', \nu)$. By contrast, replacing $\mathrm{TV}^2$ with, for example, the squared $L^2$-Wasserstein distance $\mathcal{W}_2^2(\nu', \nu)$ would reduce the generality of our analysis. Apart from the relative entropy, we are not aware of any divergences satisfying $D_h(\nu', \nu) \geq \frac{\alpha}{2}\mathcal{W}_2^2(\nu', \nu)$, and even this would require a considerably stronger (and difficult to verify) condition that the iterates produced by Algorithm 1 satisfy the Talagrand inequality.*

---

[2]We say $f(n) = \Theta(g(n))$ if there exists $c_1, c_2, n_0 > 0$ such that $c_1 g(n) \leq f(n) \leq c_2 g(n)$, for all $n \geq n_0$.

## 2.1 SKETCH OF PROOF OF THEOREM 2.1

Here, we present the proof sketch of Theorem 2.1. The full proof is provided in Section A.

Applying the Bregman proximal inequality (Lemma B.1) to each of the simultaneous mirror steps yields two inequalities: one for the $\nu$-update and one for the $\mu$-update. Each inequality relates the corresponding linearized terms in (1) to three Bregman terms: $D_h(\nu, \nu^n)$, $D_h(\nu, \nu^{n+1})$ and $D_h(\nu^{n+1}, \nu^n)$, and likewise for the $\mu$-update.

Combining the linearized terms in (1) involving $\nu - \nu^n$ and $\mu - \mu^n$ with the convexity-concavity of $F$ via Assumption 1.5, we obtain upper bounds on the differences $F(\nu^n, \mu^n) - F(\nu, \mu^n)$ and $F(\nu^n, \mu) - F(\nu, \mu^n)$, respectively.

Assumptions 1.1 and 1.6 then convert the remaining linearized terms involving $\nu^{n+1} - \nu^n$, $\mu^{n+1} - \mu^n$ and the Bregman divergences $D_h(\nu^{n+1}, \nu^n)$, $D_h(\mu^{n+1}, \mu^n)$ into quadratic bounds in TV. More precisely, for the $\nu$-part one obtains the expression

$$\int_{\mathcal{X}} \frac{\delta F}{\delta \nu}(\nu^n, \mu^n, x)(\nu - \nu^{n+1})(\mathrm{d}x) - \frac{1}{\tau} D_h(\nu^{n+1}, \nu^n) \leq 2C_1 \operatorname{TV}(\nu^{n+1}, \nu^n) - \frac{\alpha}{2\tau} \operatorname{TV}^2(\nu^{n+1}, \nu^n),$$

and similarly for the $\mu$-part. Combining the two inequalities gives the unified bound

$$F(\nu^n, \mu) - F(\nu, \mu^n) \leq \frac{2\tau}{\alpha} \left( C_1^2 + C_2^2 \right) + \frac{1}{\tau} \left( D_h(\nu, \nu^n) + D_h(\mu, \mu^n) - D_h(\nu, \nu^{n+1}) - D_h(\mu, \mu^{n+1}) \right).$$

Summing over $n = 0, ..., N - 1$, telescoping the Bregman terms, dividing by $N$ and using Jensen's inequality then yields

$$\operatorname{NI}\left( \frac{1}{N} \sum_{n=0}^{N-1} \nu^n, \frac{1}{N} \sum_{n=0}^{N-1} \mu^n \right) \leq \frac{2\tau}{\alpha} \left( C_1^2 + C_2^2 \right) + \frac{1}{N\tau} \left( \sup_{\nu \in \mathcal{C}} D_h(\nu, \nu^0) + \sup_{\mu \in \mathcal{D}} D_h(\mu, \mu^0) \right).$$

Setting $\tau = \Theta\left( N^{-1/2} \right)$ leads to the final bound, establishing the claim.

## 3 CONVERGENCE OF THE ALTERNATING MDA ALGORITHM 2

Before we state the main result concerning the convergence of the alternating MDA Algorithm 2, we introduce the necessary notions on the dual space of the space of probability measures.

Let $(\mathcal{M}(\mathcal{X}), \| \cdot \|_{\mathrm{TV}})$ be the Banach space of finite signed measures $\mu$ on $\mathcal{X}$ equipped with the total variation norm $\|\mu\|_{\mathrm{TV}} := |\mu|(\mathcal{X})$. Let $(C_b(\mathcal{X}), \| \cdot \|_\infty)$ be the Banach space of bounded continuous functions from $\mathcal{X} \subset \mathbb{R}^d$ to $(\mathbb{R}, |\cdot|)$, where $|\cdot|$ is the Euclidean norm. For any $(f, m) \in C_b(\mathcal{X}) \times \mathcal{M}(\mathcal{X})$, we define the duality pairing $\langle \cdot, \cdot \rangle : C_b(\mathcal{X}) \times \mathcal{M}(\mathcal{X}) \to \mathbb{R}$ by

$$\langle f, m \rangle := \int_{\mathcal{X}} f(x)m(\mathrm{d}x). \tag{6}$$

Next, we define the notion of convex conjugate of $h : \mathcal{M}(\mathcal{X}) \to \mathbb{R}$ relative to the duality pairing (6).

**Definition 3.1** (Convex conjugate). *Let $h : \mathcal{M}(\mathcal{X}) \to \mathbb{R}$ be a function. Then the map $h^* : C_b(\mathcal{X}) \to \mathbb{R}$ given by*

$$h^*(f) := \sup_{m \in \mathcal{M}(\mathcal{X})} \{\langle f, m \rangle - h(m)\}$$

*is called the convex conjugate of $h$.*

Regardless of the convexity of $h$, it follows from (Bonnans & Shapiro, 2000, Theorem 2.112) that $h^*$ is convex on $C_b(\mathcal{X})$, i.e., for all $\lambda \in [0, 1]$ and all $f', f \in C_b(\mathcal{X})$, we have that $h^*((1 - \lambda)f + \lambda f') \leq (1 - \lambda)h^*(f) + \lambda h^*(f')$. In Examples H.2 and H.4, we provide in Examples 1.3 and 1.4 the explicit form of $h^*$ when $h$ is the relative entropy and the chi-squared divergence, respectively.

Analogous to the characterization of the convexity of $h$ on $\mathcal{P}(\mathcal{X})$ via flat derivatives, we can characterize the convexity of $h^*$ on $C_b(\mathcal{X})$ using its Fréchet derivative (cf. Definition H.1). We say $h^* : C_b(\mathcal{X}) \to \mathbb{R}$ is Fréchet-convex if for any $f, f' \in C_b(\mathcal{X})$,

$$h^*(f') - h^*(f) \geq \nabla_{\mathcal{F}} h^*(f)[f' - f].$$

As shown in Examples H.3 and H.5, when $h$ is chosen as the relative entropy or the chi-squared divergence, its convex conjugate $h^*$ admits the Fréchet derivatives $\nabla_{\mathcal{F}} h^*(f)$.

Furthermore, using the Fréchet characterization of convexity, we can define the Bregman divergence between $f$ and $f'$ on the dual space $C_b(\mathcal{X})$.

**Definition 3.2** (Dual Bregman divergence). *Let $h^* : C_b(\mathcal{X}) \to \mathbb{R}$ be the convex conjugate of $h$. The dual $h^*$-Bregman divergence is the map $D_{h^*} : C_b(\mathcal{X}) \times C_b(\mathcal{X}) \to [0, \infty)$ given by*

$$D_{h^*}(f', f) := h^*(f') - h^*(f) - \nabla_{\mathcal{F}} h^*(f)[f' - f].$$

Before stating the next assumption on the dual space, we introduce two additional assumptions on $F$. These conditions are required only for the analysis of the alternating scheme, as they allow us to control the extra asymmetric terms arising from the updates in (2).

**Assumption 3.3** (Relative smoothness of $F$). *Assume the function $F$ is $(L_\nu, L_\mu)$-smooth relative to $h$, i.e., there exist $L_\nu, L_\mu > 0$ such that, for any $\nu, \nu' \in \mathcal{C}$ and $\mu, \mu' \in \mathcal{D}$, we have*

$$D_{F(\cdot, \mu)}(\nu', \nu) = F(\nu', \mu) - F(\nu, \mu) - \int_{\mathcal{X}} \frac{\delta F}{\delta \nu}(\nu, \mu, x)(\nu' - \nu)(\mathrm{d}x) \leq L_\nu D_h(\nu', \nu),$$

$$D_{F(\nu, \cdot)}(\mu', \mu) = F(\nu, \mu') - F(\nu, \mu) - \int_{\mathcal{X}} \frac{\delta F}{\delta \mu}(\nu, \mu, y)(\mu' - \mu)(\mathrm{d}y) \geq -L_\mu D_h(\mu', \mu).$$

**Assumption 3.4** (Uniform boundedness of $F$). *Suppose that there exists $M > 0$ such that, for all $(\nu, \mu) \in \mathcal{C} \times \mathcal{D}$, we have*

$$|F(\nu, \mu)| \leq M.$$

In Proposition C.1 and D.1, we verify that Assumptions 3.3 and 3.4 are satisfied by Examples 1.2, D and E. In Lemma B.4, we show that Assumption 3.3 corresponds to the intuition we have from Euclidean optimization where relative smoothness is equivalent to the Hessian of $F$ being upper and lower bounded by the Hessian of $h$ weighted by the smoothness constants $L_\mu$ and $L_\nu$, respectively.

The following uniform boundedness assumption on the third-order Fréchet derivative $\nabla_{\mathcal{F}}^3 h^*(f)$ at $f \in C_b(\mathcal{X})$ (cf. Definition H.10) will turn out to be crucial for showing the improvement in the convergence rate of Algorithm 2 compared to the simultaneous algorithm. We note that this assumption is the infinite-dimensional counterpart of the condition in (Wibisono et al., 2022, Theorem 3.2), where the authors require the third-order derivative of the dual Bregman potential (a tensor-valued map) to be uniformly bounded in operator norm.

**Assumption 3.5** (Uniform boundedness of $\nabla_{\mathcal{F}}^3 h^*(f)$). *Suppose that $(C_b(\mathcal{X}) \times C_b(\mathcal{X}) \times C_b(\mathcal{X})) \ni (g, g, g) \mapsto \nabla_{\mathcal{F}}^3 h^*(f)[g][g][g] \in \mathbb{R}$ is uniformly bounded, i.e, there exists $L_{h^*} > 0$ such that for all $g \in C_b(\mathcal{X})$,*

$$\left| \nabla_{\mathcal{F}}^3 h^*(f)[g][g][g] \right| \leq L_{h^*} \|g\|_\infty^3.$$

In Examples H.11 and H.12 and Propositions H.14 and H.15, we provide the explicit form of the third Fréchet derivative $\nabla_{\mathcal{F}}^3 h^*(f)$ and verify Assumption 3.5 in the case where $h$ is the relative entropy and the $\chi^2$-divergence, respectively.

Now, we are ready to state the second main result of the paper.

**Theorem 3.6** (Convergence of the alternating MDA Algorithm 2). *Let $(\nu^0, \mu^0)$ be such that $\sup_{\nu \in \mathcal{C}} D_h(\nu, \nu^0) + \sup_{\mu \in \mathcal{D}} D_h(\mu, \mu^0) < \infty$ (cf. Remark 2.3). Let Assumptions 1.1, 1.5, 1.6, 3.3, 3.4 and 3.5 hold. Suppose that $\tau L \leq \frac{1}{2}$, with $L := \max\{L_\nu, L_\mu\}$, set $\kappa_1 := \frac{1}{6}(C_1^3 + C_2^3)$ and $\kappa_2 := C_1^2 + C_2^2$. If $\tau = \Theta\left(N^{-1/3}\right)$, then*

$$\mathrm{NI}\left(\frac{1}{N} \sum_{n=0}^{N-1} \nu^{n+1}, \frac{1}{N} \sum_{n=0}^{N-1} \mu^n\right) \leq \frac{1}{(2N)^{2/3}} \left(3 \left(\sup_{\nu \in \mathcal{C}} D_h(\nu, \nu^0) + \sup_{\mu \in \mathcal{D}} D_h(\mu, \mu^0)\right)^{2/3} \times \right.$$

$$\left. \times \left(\frac{\kappa_1 L_{h^*}}{2} + \frac{8\kappa_2 L}{\alpha}\right)^{1/3} + 4^{1/3} M\right). \quad (7)$$

**Remark 3.7** (Bilinear games). *In particular, if $F(\nu, \mu) = \int_{\mathcal{X}} \int_{\mathcal{X}} f(x, y)\nu(\mathrm{d}x)\mu(\mathrm{d}y)$, for a bounded function $f : \mathcal{X} \times \mathcal{X} \to \mathbb{R}$, then Assumptions 1.5, 1.6, 3.3, 3.4 are satisfied and in Assumption 3.3 we have $L_\nu = L_\mu = 0$. Therefore, $L = 0$ in (7), and hence Theorem 3.6 is consistent with the already known convergence rate $\mathcal{O}\left(N^{-2/3}\right)$ of the MDA algorithm for min-max games with strategies in compact convex subsets of $\mathbb{R}^d$ and bilinear payoff function; see (Wibisono et al., 2022, Theorem 3.2 and Corollary 3.3). Since we work in an infinite-dimensional setting with a non-linear convex-concave objective function $F$, Theorem 3.6 substantially generalizes the results of (Wibisono et al., 2022).*

### 3.1 SKETCH OF PROOF OF THEOREM 3.6

The main challenge in the proof, compared to the proof for the simultaneous updates, is in controlling the additional term $F(\nu^{n+1}, \mu^n) - F(\nu^n, \mu^n)$ produced by the non-symmetric update in Algorithm 2. Using Assumption 3.3, we combine this term with $\int_{\mathcal{X}} \frac{\delta F}{\delta \nu}(\nu^n, \mu^n, x)(\nu^n - \nu^{n+1})(\mathrm{d}x)$ and $\int_{\mathcal{X}} \frac{\delta F}{\delta \mu}(\nu^{n+1}, \mu^n, y)(\mu^{n+1} - \mu^n)(\mathrm{d}y)$, which yields the Bregman commutators $D_h(\nu^n, \nu^{n+1}) - D_h(\nu^{n+1}, \nu^n)$ and $D_h(\mu^n, \mu^{n+1}) - D_h(\mu^{n+1}, \mu^n)$. Proceeding as in the simultaneous case leads to

$$
\mathrm{NI}\left(\frac{1}{N}\sum_{n=0}^{N-1}\nu^{n+1}, \frac{1}{N}\sum_{n=0}^{N-1}\mu^n\right) \leq \mathcal{O}\left(\frac{1}{N\tau}\right) + \mathcal{O}(\tau^2) + \mathcal{O}\left(\frac{M}{N}\right)
$$
$$
+ \frac{1}{2N\tau}\sum_{n=0}^{N-1}\left(D_h(\nu^n, \nu^{n+1}) - D_h(\nu^{n+1}, \nu^n) + D_h(\mu^n, \mu^{n+1}) - D_h(\mu^{n+1}, \mu^n)\right),
$$

where the final term uses Assumption 3.4. We transport the commutators to the dual space via Lemma I.5 and obtain

$$
D_h(\nu^n, \nu^{n+1}) - D_h(\nu^{n+1}, \nu^n) = D_{h^*}\left(\frac{\delta h}{\delta \nu}(\nu^{n+1}, \cdot), \frac{\delta h}{\delta \nu}(\nu^n, \cdot)\right) - D_{h^*}\left(\frac{\delta h}{\delta \nu}(\nu^n, \cdot), \frac{\delta h}{\delta \nu}(\nu^{n+1}, \cdot)\right).
$$

Using the third-order Fréchet derivative of $h^*$ (Definition H.10) and its uniform boundedness (Assumption 3.5), we obtain

$$
D_{h^*}\left(\frac{\delta h}{\delta \nu}(\nu^{n+1}, \cdot), \frac{\delta h}{\delta \nu}(\nu^n, \cdot)\right) - D_{h^*}\left(\frac{\delta h}{\delta \nu}(\nu^n, \cdot), \frac{\delta h}{\delta \nu}(\nu^{n+1}, \cdot)\right) \leq \frac{L_{h^*}}{6}\left\|\frac{\delta h}{\delta \nu}(\nu^{n+1}, \cdot) - \frac{\delta h}{\delta \nu}(\nu^n, \cdot)\right\|_\infty^3,
$$

By Proposition B.3, the first-order optimality condition for the $\nu$-update yields (up to an additive constant)

$$
\frac{\delta h}{\delta \nu}(\nu^{n+1}, x) - \frac{\delta h}{\delta \nu}(\nu^n, x) = -\tau\frac{\delta F}{\delta \nu}(\nu^n, \mu^n, x),
$$

for all $x \in \mathcal{X}$ $\nu^{n+1}$-a.e. Hence, by Assumption 1.6,

$$
D_{h^*}\left(\frac{\delta h}{\delta \nu}(\nu^{n+1}, \cdot), \frac{\delta h}{\delta \nu}(\nu^n, \cdot)\right) - D_{h^*}\left(\frac{\delta h}{\delta \nu}(\nu^n, \cdot), \frac{\delta h}{\delta \nu}(\nu^{n+1}, \cdot)\right) \leq \frac{L_{h^*}}{6}\tau^3 C_1^3,
$$

and the same argument applies to the $\mu$-commutator. Therefore,

$$
\mathrm{NI}\left(\frac{1}{N}\sum_{n=0}^{N-1}\nu^{n+1}, \frac{1}{N}\sum_{n=0}^{N-1}\mu^n\right) \leq \mathcal{O}\left(\frac{1}{N\tau}\right) + \mathcal{O}\left(\frac{\tau^3}{\tau}\right) + \mathcal{O}(\tau^2) + \mathcal{O}\left(\frac{M}{N}\right).
$$

Choosing $\tau = \Theta\left(N^{-1/3}\right)$ yields the conclusion.

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

## 4 APPENDIX

The appendix is organized into several sections (A–K), each providing supporting technical material, extended proofs and further context for the main text. Section A contains the complete proofs of the paper's two main results, namely Theorem 2.1 and Theorem 3.6. Section B contains proofs for supplementary lemmas and propositions referenced earlier in the paper. In Section C, we check that Assumptions 1.5, 1.6, 3.3 and 3.4 hold in Example 1.2. In Sections D and E we provide additional applications of our framework and prove that these satisfy Assumptions 1.5, 1.6, 3.3 and 3.4. In Section F we describe all implementation details, experimental settings and parameter choices.

Section G collects standard definitions concerning differentiability with respect to probability measures that are used throughout the paper. Section H complements the previous section by treating differentiability in the dual space and supporting the duality arguments used in the proof of Theorem 3.6. Section I gathers duality identities used particularly in the proof of Theorem 3.6. Section J shows that an implicit MDA scheme achieves the same sublinear convergence rate as the continuous-time dynamics under convex–concave assumptions on $F$. Finally, Section K collects references and related literature not included in the main text.

## CONTENTS

## A  PROOFS OF THEOREM 2.1 AND THEOREM 3.6

This section is dedicated to the proofs of the main results, namely Theorem 2.1 and Theorem 3.6. We start with the proof of Theorem 2.1.

### A.1 PROOF OF THEOREM 2.1

*Proof of Theorem 2.1.* Since $\nu \mapsto \tau \int_{\mathcal{X}} \frac{\delta F}{\delta \nu}(\nu^n, \mu^n, x)(\nu - \nu^n)(\mathrm{d}x)$ is convex, applying Lemma B.1 with $\bar{\nu} = \nu^{n+1}$ and $\mu = \nu^n$ implies that, for any $\nu \in \mathcal{C}$, we have

$$\tau \int_{\mathcal{X}} \frac{\delta F}{\delta \nu}(\nu^n, \mu^n, x)(\nu - \nu^n)(\mathrm{d}x) + D_h(\nu, \nu^n) \geq \tau \int_{\mathcal{X}} \frac{\delta F}{\delta \nu}(\nu^n, \mu^n, x)(\nu^{n+1} - \nu^n)(\mathrm{d}x)$$
$$+ D_h(\nu^{n+1}, \nu^n) + D_h(\nu, \nu^{n+1}),$$

or, equivalently,

$$- \tau \int_{\mathcal{X}} \frac{\delta F}{\delta \nu}(\nu^n, \mu^n, x)(\nu - \nu^n)(\mathrm{d}x) - D_h(\nu, \nu^n) \leq -\tau \int_{\mathcal{X}} \frac{\delta F}{\delta \nu}(\nu^n, \mu^n, x)(\nu^{n+1} - \nu^n)(\mathrm{d}x)$$
$$- D_h(\nu^{n+1}, \nu^n) - D_h(\nu, \nu^{n+1}). \quad (8)$$

Similarly, since $\mu \mapsto -\tau \int_{\mathcal{X}} \frac{\delta F}{\delta \mu}(\nu^n, \mu^n, y)(\mu - \mu^n)(\mathrm{d}y)$ is convex, applying Lemma B.1 with $\bar{\nu} = \mu^{n+1}$ and $\mu = \mu^n$ implies that, for any $\mu \in \mathcal{D}$, we have

$$\tau \int_{\mathcal{X}} \frac{\delta F}{\delta \mu}(\nu^n, \mu^n, y)(\mu - \mu^n)(\mathrm{d}y) - D_h(\mu, \mu^n) \leq \tau \int_{\mathcal{X}} \frac{\delta F}{\delta \mu}(\nu^n, \mu^n, y)(\mu^{n+1} - \mu^n)(\mathrm{d}y)$$
$$- D_h(\mu^{n+1}, \mu^n) - D_h(\mu, \mu^{n+1}). \quad (9)$$

Using the convexity of $\nu \mapsto F(\nu, \mu)$ in (8), with $\nu = \nu^n$ and $\mu = \mu^n$, we have that

$$F(\nu^n, \mu^n) - F(\nu, \mu^n) - \frac{1}{\tau} D_h(\nu, \nu^n) \leq \int_{\mathcal{X}} \frac{\delta F}{\delta \nu}(\nu^n, \mu^n, x)(\nu^n - \nu^{n+1})(\mathrm{d}x)$$
$$- \frac{1}{\tau} D_h(\nu^{n+1}, \nu^n) - \frac{1}{\tau} D_h(\nu, \nu^{n+1}). \quad (10)$$

From Assumptions 1.1 and 1.6, it follows from (10) that

$$F(\nu^n, \mu^n) - F(\nu, \mu^n) - \frac{1}{\tau} D_h(\nu, \nu^n) \leq 2C_1 \operatorname{TV}(\nu^{n+1}, \nu^n) - \frac{\alpha}{2\tau} \operatorname{TV}^2(\nu^{n+1}, \nu^n) - \frac{1}{\tau} D_h(\nu, \nu^{n+1})$$
$$= -\frac{\alpha}{2\tau}\left( \operatorname{TV}(\nu^{n+1}, \nu^n) - \frac{2\tau C_1}{\alpha} \right)^2 + \frac{2\tau C_1^2}{\alpha} - \frac{1}{\tau} D_h(\nu, \nu^{n+1})$$
$$\leq \frac{2\tau C_1^2}{\alpha} - \frac{1}{\tau} D_h(\nu, \nu^{n+1}), \quad (11)$$

where the equality follows from the standard identity $-(a-b)^2 + b^2 = -a^2 + 2ab$.

Similarly, using concavity of $\mu \mapsto F(\nu, \mu)$ in (9), with $\nu = \nu^n$ and $\mu = \mu^n$, we have that

$$F(\nu^n, \mu) - F(\nu^n, \mu^n) - \frac{1}{\tau} D_h(\mu, \mu^n) \leq \int_{\mathcal{X}} \frac{\delta F}{\delta \mu}(\nu^n, \mu^n, y)(\mu^{n+1} - \mu^n)(\mathrm{d}y)$$
$$- \frac{1}{\tau} D_h(\mu^{n+1}, \mu^n) - \frac{1}{\tau} D_h(\mu, \mu^{n+1}). \quad (12)$$

From Assumptions 1.1 and 1.6, it follows from (12) that

$$F(\nu^n, \mu) - F(\nu^n, \mu^n) - \frac{1}{\tau} D_h(\mu, \mu^n) \leq 2C_2 \operatorname{TV}(\mu^{n+1}, \mu^n) - \frac{\alpha}{2\tau} \operatorname{TV}^2(\mu^{n+1}, \mu^n) - \frac{1}{\tau} D_h(\mu, \mu^{n+1})$$
$$= -\frac{\alpha}{2\tau}\left( \operatorname{TV}(\mu^{n+1}, \mu^n) - \frac{2\tau C_2}{\alpha} \right)^2 + \frac{2\tau C_2^2}{\alpha} - \frac{1}{\tau} D_h(\mu, \mu^{n+1})$$
$$\leq \frac{2\tau C_2^2}{\alpha} - \frac{1}{\tau} D_h(\mu, \mu^{n+1}). \quad (13)$$

Adding inequalities (11) and (13) implies that for any $(\nu, \mu) \in \mathcal{C} \times \mathcal{D}$ we have

$$F(\nu^n, \mu) - F(\nu, \mu^n) \leq \frac{2\tau}{\alpha}\left( C_1^2 + C_2^2 \right) + \frac{1}{\tau} D_h(\nu, \nu^n) + \frac{1}{\tau} D_h(\mu, \mu^n) - \frac{1}{\tau} D_h(\nu, \nu^{n+1}) - \frac{1}{\tau} D_h(\mu, \mu^{n+1}).$$
$$(14)$$

Summing the previous inequality over $n = 0, 1, ..., N-1$, using $D_h(\nu, \nu^N) + D_h(\mu, \mu^N) \geq 0$, for any $(\nu, \mu) \in \mathcal{C} \times \mathcal{D}$, bounding the right-hand from above by its supremum over $(\nu, \mu) \in \mathcal{C} \times \mathcal{D}$, and dividing by $N$ gives

$$\frac{1}{N} \sum_{n=0}^{N-1} (F(\nu^n, \mu) - F(\nu, \mu^n)) \leq \frac{2\tau}{\alpha} \left( C_1^2 + C_2^2 \right) + \frac{1}{N\tau} \left( \sup_{\nu \in \mathcal{C}} D_h(\nu, \nu^0) + \sup_{\mu \in \mathcal{D}} D_h(\mu, \mu^0) \right). \tag{15}$$

Since $\nu \mapsto F(\nu, \mu)$ and $\mu \mapsto -F(\nu, \mu)$ are convex, it follows by Jensen's inequality that

$$\frac{1}{N} \sum_{n=0}^{N-1} (F(\nu^n, \mu) - F(\nu, \mu^n)) = \frac{1}{N} \sum_{n=0}^{N-1} F(\nu^n, \mu) - \frac{1}{N} \sum_{n=0}^{N-1} F(\nu, \mu^n)$$

$$\geq F\left( \frac{1}{N} \sum_{n=0}^{N-1} \nu^n, \mu \right) - F\left( \nu, \frac{1}{N} \sum_{n=0}^{N-1} \mu^n \right). \tag{16}$$

Combining (15) with (16) and taking maximum over $(\nu, \mu)$ gives

$$\mathrm{NI}\left( \frac{1}{N} \sum_{n=0}^{N-1} \nu^n, \frac{1}{N} \sum_{n=0}^{N-1} \mu^n \right) \leq \frac{2\tau}{\alpha} \left( C_1^2 + C_2^2 \right) + \frac{1}{N\tau} \left( \sup_{\nu \in \mathcal{C}} D_h(\nu, \nu^0) + \sup_{\mu \in \mathcal{D}} D_h(\mu, \mu^0) \right).$$

Minimizing the right-hand side over $\tau$ amounts to taking

$$\tau = \sqrt{\frac{\alpha \left( \sup_{\nu \in \mathcal{C}} D_h(\nu, \nu^0) + \sup_{\mu \in \mathcal{D}} D_h(\mu, \mu^0) \right)}{2N \left( C_1^2 + C_2^2 \right)}},$$

and hence we obtain

$$\mathrm{NI}\left( \frac{1}{N} \sum_{n=0}^{N-1} \nu^n, \frac{1}{N} \sum_{n=0}^{N-1} \mu^n \right) \leq 2\sqrt{\frac{2 \left( C_1^2 + C_2^2 \right) \left( \sup_{\nu \in \mathcal{C}} D_h(\nu, \nu^0) + \sup_{\mu \in \mathcal{D}} D_h(\mu, \mu^0) \right)}{\alpha N}}.$$

$\square$

### A.2 Proof of Theorem 3.6

Before we proceed with the proof of Theorem 3.6, we will need an auxiliary result, which will turn out to be essential. The proof of Lemma A.1 is given in Appendix B.

**Lemma A.1.** *Let Assumptions 1.1, 1.6 and 3.3 hold. Suppose that $\tau L \leq \frac{1}{2}$, with $L := \max\{L_\nu, L_\mu\}$. Then, for both Algorithms 1 and 2, it holds, for all $n \geq 0$, that*

$$D_h(\nu^{n+1}, \nu^n) \leq \frac{16\tau^2 C_1^2}{\alpha} \quad \text{and} \quad D_h(\mu^{n+1}, \mu^n) \leq \frac{16\tau^2 C_2^2}{\alpha}.$$

*Proof of Theorem 3.6.* We start the proof by following the same calculations from Theorem 2.1. For (2), after applying Lemma B.1 and using convexity-concavity of $F$, (10) remains unchanged, i.e.,

$$F(\nu^n, \mu^n) - F(\nu, \mu^n) - \frac{1}{\tau} D_h(\nu, \nu^n) \leq \int_{\mathcal{X}} \frac{\delta F}{\delta \nu}(\nu^n, \mu^n, x)(\nu^n - \nu^{n+1})(\mathrm{d}x)$$

$$- \frac{1}{\tau} D_h(\nu^{n+1}, \nu^n) - \frac{1}{\tau} D_h(\nu, \nu^{n+1}),$$

while (12) becomes

$$F(\nu^{n+1}, \mu) - F(\nu^{n+1}, \mu^n) - \frac{1}{\tau} D_h(\mu, \mu^n) \leq \int_{\mathcal{X}} \frac{\delta F}{\delta \mu}(\nu^{n+1}, \mu^n, y)(\mu^{n+1} - \mu^n)(\mathrm{d}y)$$

$$- \frac{1}{\tau} D_h(\mu^{n+1}, \mu^n) - \frac{1}{\tau} D_h(\mu, \mu^{n+1}).$$

Adding the previous two inequalities, summing the resulting inequality over $n = 0, 1, ..., N-1$, dividing by $N$, using (16) and taking maximum over $(\nu, \mu)$ we arrive at

$$\mathrm{NI}\left(\frac{1}{N}\sum_{n=0}^{N-1}\nu^{n+1}, \frac{1}{N}\sum_{n=0}^{N-1}\mu^{n}\right) \leq \frac{1}{N}\sum_{n=0}^{N-1}\left(\int_{\mathcal{X}}\frac{\delta F}{\delta\nu}(\nu^{n}, \mu^{n}, x)(\nu^{n} - \nu^{n+1})(\mathrm{d}x)\right.$$

$$\left. + \int_{\mathcal{X}}\frac{\delta F}{\delta\mu}(\nu^{n+1}, \mu^{n}, y)(\mu^{n+1} - \mu^{n})(\mathrm{d}y)\right) + \frac{1}{N\tau}\left(\sup_{\nu\in\mathcal{C}}D_h(\nu, \nu^0) + \sup_{\mu\in\mathcal{D}}D_h(\mu, \mu^0)\right)$$

$$+ \frac{1}{N}\sum_{n=0}^{N-1}\left(F(\nu^{n+1}, \mu^n) - F(\nu^n, \mu^n)\right) - \frac{1}{N\tau}\sum_{n=0}^{N-1}\left(D_h(\nu^{n+1}, \nu^n) + D_h(\mu^{n+1}, \mu^n)\right), \quad (17)$$

where we used the fact that $D_h(\nu, \nu^N) + D_h(\mu, \mu^N) \geq 0$, for any $(\nu, \mu) \in \mathcal{C} \times \mathcal{D}$.

Note that the key difference to the estimates from Theorem 2.1 is the appearance of the term $F(\nu^{n+1}, \mu^n) - F(\nu^n, \mu^n)$ due to the non-symmetry of the flat derivatives of $F$ in (2). The idea is to combine $F(\nu^{n+1}, \mu^n) - F(\nu^n, \mu^n)$ with both $\int_{\mathcal{X}}\frac{\delta F}{\delta\nu}(\nu^n, \mu^n, x)(\nu^n - \nu^{n+1})(\mathrm{d}x)$ and $\int_{\mathcal{X}}\frac{\delta F}{\delta\mu}(\nu^{n+1}, \mu^n, y)(\mu^{n+1} - \mu^n)(\mathrm{d}y)$ via relative smoothness in order to obtain $D_h(\nu^n, \nu^{n+1}) - D_h(\nu^{n+1}, \nu^n)$ and $D_h(\mu^n, \mu^{n+1}) - D_h(\mu^{n+1}, \mu^n)$, which will prove to be of order $\mathcal{O}(\tau^3)$.

By Proposition B.3, the first-order conditions for (2) read

$$\begin{cases} \frac{\delta h}{\delta\nu}(\nu^{n+1}, x) - \frac{\delta h}{\delta\nu}(\nu^n, x) = -\tau\frac{\delta F}{\delta\nu}(\nu^n, \mu^n, x) + C_{n,1}, \\ \frac{\delta h}{\delta\mu}(\mu^{n+1}, y) - \frac{\delta h}{\delta\mu}(\mu^n, y) = \tau\frac{\delta F}{\delta\mu}(\nu^{n+1}, \mu^n, y) + C_{n,2}, \end{cases} \quad (18)$$

for all $x \in \mathcal{X}$ $\nu^{n+1}$-a.e. and $y \in \mathcal{X}$ $\mu^{n+1}$-a.e., where $C_{n,1}, C_{n,2} \in \mathbb{R}$. It can be shown directly from Definition 1.2 that

$$\int_{\mathcal{X}}\left(\frac{\delta h}{\delta\nu}(\nu', x) - \frac{\delta h}{\delta\nu}(\nu, x)\right)(\nu' - \nu)(\mathrm{d}x) = D_h(\nu', \nu) + D_h(\nu, \nu'), \quad (19)$$

for all $\nu, \nu' \in \mathcal{C}$, and analogously for $D_h(\mu', \mu) + D_h(\mu, \mu')$. Then, using (18) and (19) we obtain that

$$-\int_{\mathcal{X}}\frac{\delta F}{\delta\nu}(\nu^n, \mu^n, x)(\nu^{n+1} - \nu^n)(\mathrm{d}x) = \frac{1}{\tau}\int_{\mathcal{X}}\left(\frac{\delta h}{\delta\nu}(\nu^{n+1}, x) - \frac{\delta h}{\delta\nu}(\nu^n, x)\right)(\nu^{n+1} - \nu^n)(\mathrm{d}x)$$

$$= \frac{1}{\tau}\left(D_h(\nu^{n+1}, \nu^n) + D_h(\nu^n, \nu^{n+1})\right), \quad (20)$$

and similarly

$$\int_{\mathcal{X}}\frac{\delta F}{\delta\mu}(\nu^{n+1}, \mu^n, y)(\mu^{n+1} - \mu^n)(\mathrm{d}y) = \frac{1}{\tau}\int_{\mathcal{X}}\left(\frac{\delta h}{\delta\mu}(\mu^{n+1}, y) - \frac{\delta h}{\delta\mu}(\mu^n, y)\right)(\mu^{n+1} - \mu^n)(\mathrm{d}y)$$

$$= \frac{1}{\tau}\left(D_h(\mu^{n+1}, \mu^n) + D_h(\mu^n, \mu^{n+1})\right). \quad (21)$$

Therefore, using (20) and (21) in (17), we obtain that

$$\mathrm{NI}\left(\frac{1}{N}\sum_{n=0}^{N-1}\nu^{n+1}, \frac{1}{N}\sum_{n=0}^{N-1}\mu^n\right) \leq \frac{1}{N\tau}\left(\sup_{\nu\in\mathcal{C}}D_h(\nu, \nu^0) + \sup_{\mu\in\mathcal{D}}D_h(\mu, \mu^0)\right)$$

$$+ \frac{1}{N\tau}\sum_{n=0}^{N-1}\left(D_h(\nu^{n+1}, \nu^n) + D_h(\nu^n, \nu^{n+1}) + D_h(\mu^{n+1}, \mu^n) + D_h(\mu^n, \mu^{n+1})\right)$$

$$+ \frac{1}{N}\sum_{n=0}^{N-1}\left(F(\nu^{n+1}, \mu^n) - F(\nu^n, \mu^n)\right) - \frac{1}{N\tau}\sum_{n=0}^{N-1}\left(D_h(\nu^{n+1}, \nu^n) + D_h(\mu^{n+1}, \mu^n)\right). \quad (22)$$

Then, we observe that

$$D_h(\nu^n, \nu^{n+1}) = \frac{1}{2}\left(D_h(\nu^n, \nu^{n+1}) - D_h(\nu^{n+1}, \nu^n)\right) + \frac{1}{2}\left(D_h(\nu^n, \nu^{n+1}) + D_h(\nu^{n+1}, \nu^n)\right),$$

$$(23)$$

and a similar representation holds for $D_h(\mu^n, \mu^{n+1})$. Similarly, we can write

$$
F(\nu^{n+1}, \mu^n) - F(\nu^n, \mu^n) = \frac{1}{2} \left( F(\nu^{n+1}, \mu^n) - F(\nu^n, \mu^n) \right)
$$
$$
+ \frac{1}{2} \left( F(\nu^{n+1}, \mu^n) - F(\nu^{n+1}, \mu^{n+1}) + F(\nu^{n+1}, \mu^{n+1}) - F(\nu^n, \mu^n) \right). \quad (24)
$$

Therefore, putting (23) and (24) into (22) gives

$$
\mathrm{NI}\left( \frac{1}{N} \sum_{n=0}^{N-1} \nu^{n+1}, \frac{1}{N} \sum_{n=0}^{N-1} \mu^n \right) \leq \frac{1}{N\tau} \left( \sup_{\nu \in \mathcal{C}} D_h(\nu, \nu^0) + \sup_{\mu \in \mathcal{D}} D_h(\mu, \mu^0) \right)
$$
$$
+ \frac{1}{2N\tau} \sum_{n=0}^{N-1} \left( D_h(\nu^n, \nu^{n+1}) - D_h(\nu^{n+1}, \nu^n) + D_h(\mu^n, \mu^{n+1}) - D_h(\mu^{n+1}, \mu^n) \right)
$$
$$
+ \frac{1}{2N} \sum_{n=0}^{N-1} \left( \frac{1}{\tau}(D_h(\nu^n, \nu^{n+1}) + D_h(\nu^{n+1}, \nu^n)) + F(\nu^{n+1}, \mu^n) - F(\nu^n, \mu^n) \right)
$$
$$
+ \frac{1}{2N} \sum_{n=0}^{N-1} \left( \frac{1}{\tau}(D_h(\mu^n, \mu^{n+1}) + D_h(\mu^{n+1}, \mu^n)) + F(\nu^{n+1}, \mu^n) - F(\nu^{n+1}, \mu^{n+1}) \right.
$$
$$
\left. + F(\nu^{n+1}, \mu^{n+1}) - F(\nu^n, \mu^n) \right). \quad (25)
$$

Combining the fact that $\nu \mapsto F(\nu, \mu)$ is $L_\nu$-smooth relative to $h$ with the first-order condition (18), we have that

$$
F(\nu^{n+1}, \mu^n) - F(\nu^n, \mu^n) \leq \int_{\mathcal{X}} \frac{\delta F}{\delta \nu}(\nu^n, \mu^n, x)(\nu^{n+1} - \nu^n)(\mathrm{d}x) + L_\nu D_h(\nu^{n+1}, \nu^n)
$$
$$
= -\frac{1}{\tau} \int_{\mathcal{X}} \left( \frac{\delta h}{\delta \nu}(\nu^{n+1}, x) - \frac{\delta h}{\delta \nu}(\nu^n, x) \right)(\nu^{n+1} - \nu^n)(\mathrm{d}x) + L_\nu D_h(\nu^{n+1}, \nu^n)
$$
$$
= -\frac{1}{\tau} \left( D_h(\nu^{n+1}, \nu^n) + D_h(\nu^n, \nu^{n+1}) \right) + L_\nu D_h(\nu^{n+1}, \nu^n), \quad (26)
$$

where the last equality follows from (19).

Similarly, using $L_\mu$-smoothness of $\mu \mapsto F(\nu, \mu)$ relative to $h$ together with (18), we can show that

$$
F(\nu^{n+1}, \mu^n) - F(\nu^{n+1}, \mu^{n+1}) + \frac{1}{\tau} \left( D_h(\mu^n, \mu^{n+1}) + D_h(\mu^{n+1}, \mu^n) \right) \leq L_\mu D_h(\mu^{n+1}, \mu^n). \quad (27)
$$

Therefore, using (26) and (27) in (25), and recalling that $L = \max\{L_\nu, L_\mu\}$ gives

$$
\mathrm{NI}\left( \frac{1}{N} \sum_{n=0}^{N-1} \nu^{n+1}, \frac{1}{N} \sum_{n=0}^{N-1} \mu^n \right) \leq \frac{1}{N\tau} \left( \sup_{\nu \in \mathcal{C}} D_h(\nu, \nu^0) + \sup_{\mu \in \mathcal{D}} D_h(\mu, \mu^0) \right)
$$
$$
+ \frac{1}{2N\tau} \sum_{n=0}^{N-1} \left( D_h(\nu^n, \nu^{n+1}) - D_h(\nu^{n+1}, \nu^n) + D_h(\mu^n, \mu^{n+1}) - D_h(\mu^{n+1}, \mu^n) \right)
$$
$$
+ \frac{L}{2N} \sum_{n=0}^{N-1} \left( D_h(\nu^{n+1}, \nu^n) + D_h(\mu^{n+1}, \mu^n) \right) + \frac{1}{2N} \left( F\left(\nu^N, \mu^N\right) - F(\nu^0, \mu^0) \right). \quad (28)
$$

Since, by Lemma A.1,

$$
D_h(\nu^{n+1}, \nu^n) + D_h(\mu^{n+1}, \mu^n) \leq \frac{16\tau^2 \kappa_2}{\alpha}, \quad (29)
$$

where $\kappa_2 := C_1^2 + C_2^2$, it suffices to show that $D_h(\nu^n, \nu^{n+1}) - D_h(\nu^{n+1}, \nu^n) + D_h(\mu^n, \mu^{n+1}) - D_h(\mu^{n+1}, \mu^n)$ is of order $\mathcal{O}(\tau^3)$. Indeed, we could then choose $\tau = \mathcal{O}\left(N^{-1/3}\right)$, and since by Assumption 3.4, $\left| F\left(\nu^N, \mu^N\right) \right| \leq M$, we would obtain that

$$
\mathrm{NI}\left( \frac{1}{N} \sum_{n=0}^{N-1} \nu^{n+1}, \frac{1}{N} \sum_{n=0}^{N-1} \mu^n \right) \leq \mathcal{O}\left( \frac{1}{N^{2/3}} \right) + \mathcal{O}\left( \frac{1}{N} \right) = \mathcal{O}\left( \frac{1}{N^{2/3}} \right),
$$

because $\frac{1}{N} \le \frac{1}{N^{2/3}}$, for all $N \ge 1$.

In order to show that $D_h(\nu^n, \nu^{n+1}) - D_h(\nu^{n+1}, \nu^n) + D_h(\mu^n, \mu^{n+1}) - D_h(\mu^{n+1}, \mu^n)$ is $\mathcal{O}(\tau^3)$, we will leverage the connection between Bregman divergence and dual Bregman divergence given by Lemma I.5 together with Assumptions 1.6 and 3.5.

If we denote $f^n := \frac{\delta h}{\delta \nu}(\nu^n, \cdot)$, for any $n \ge 0$, then by Lemma I.5, we have that $D_h(\nu^n, \nu^{n+1}) = D_{h^*}(f^{n+1}, f^n)$. For any $\varepsilon \in [0, 1]$ denote $f^{\varepsilon,n} = \varepsilon f^{n+1} + (1-\varepsilon)f^n$ and $\phi(\varepsilon) = h^*(f^{\varepsilon,n})$. Note that $\phi(0) = h^*(f^n)$ and $\phi(1) = h^*(f^{n+1})$. We have

$$\phi'(\varepsilon) = \nabla_{\mathcal{F}} h^*(f^{\varepsilon,n})[f^{n+1} - f^n], \quad \phi''(\varepsilon) = \nabla_{\mathcal{F}}^2 h^*(f^{\varepsilon,n})[f^{n+1} - f^n][f^{n+1} - f^n].$$

Note that $\phi'(0) = \nabla_{\mathcal{F}} h^*(f^n)[f^{n+1} - f^n]$. By the fundamental theorem of calculus and integration by parts, we have

$$\phi(1) - \phi(0) = \int_0^1 \phi'(\varepsilon)\mathrm{d}\varepsilon = [(t-1)\phi'(\varepsilon)]|_{\varepsilon=0}^{\varepsilon=1} - \int_0^1 (\varepsilon-1)\phi''(\varepsilon)\mathrm{d}\varepsilon = \phi'(0) + \int_0^1 (1-\varepsilon)\phi''(\varepsilon)\mathrm{d}\varepsilon.$$

Hence,

$$h^*(f^{n+1}) - h^*(f^n) - \nabla_{\mathcal{F}} h^*(f^n)[f^{n+1} - f^n] = \int_0^1 (1-\varepsilon)\nabla_{\mathcal{F}}^2 h^*(f^{\varepsilon,n})[f^{n+1} - f^n][f^{n+1} - f^n]\mathrm{d}\varepsilon$$

By Definition 3.2, we have that

$$D_{h^*}(f^{n+1}, f^n) = \int_0^1 (1-\varepsilon)\nabla_{\mathcal{F}}^2 h^*(f^{\varepsilon,n})[f^{n+1} - f^n][f^{n+1} - f^n]\mathrm{d}\varepsilon$$

Similarly, by Lemma I.5, we have that $D_h(\nu^{n+1}, \nu^n) = D_{h^*}(f^n, f^{n+1})$, and hence

$$D_{h^*}(f^n, f^{n+1}) = \int_0^1 (1-\varepsilon)\nabla_{\mathcal{F}}^2 h^*(f^{1-\varepsilon,n})[f^{n+1} - f^n][f^{n+1} - f^n]\mathrm{d}\varepsilon.$$

Therefore, we obtain that

$$D_{h^*}(f^{n+1}, f^n) - D_{h^*}(f^n, f^{n+1}) = \int_0^1 (1-\varepsilon)\left(\nabla_{\mathcal{F}}^2 h^*(f^{\varepsilon,n}) - \nabla_{\mathcal{F}}^2 h^*(f^{1-\varepsilon,n})\right)[f^{n+1} - f^n][f^{n+1} - f^n]\mathrm{d}\varepsilon.$$

Note that $f^{\varepsilon,n} - f^{1-\varepsilon,n} = (1-2\varepsilon)(f^{n+1} - f^n)$. If we denote $g^{\varepsilon,\gamma,n} = f^{1-\varepsilon,n} + \gamma(f^{\varepsilon,n} - f^{1-\varepsilon,n})$, then applying the fundamental theorem of calculus again gives

$$D_{h^*}(f^{n+1}, f^n) - D_{h^*}(f^n, f^{n+1}) = \int_0^1 (1-\varepsilon)(1-2\varepsilon)\int_0^1 \nabla_{\mathcal{F}}^3 h^*(g^{\varepsilon,\gamma,n})[f^{n+1} - f^n][f^{n+1} - f^n][f^{n+1} - f^n]\mathrm{d}\gamma\mathrm{d}\varepsilon.$$

Using Assumption 3.5, we further obtain

$$D_{h^*}(f^{n+1}, f^n) - D_{h^*}(f^n, f^{n+1}) \le L_{h^*}\|f^{n+1} - f^n\|_\infty^3 \int_0^1 (1-\varepsilon)(1-2\varepsilon)\mathrm{d}\varepsilon \le \frac{L_{h^*}}{6}\|f^{n+1} - f^n\|_\infty^3,$$

The first-order condition for the minimizing player in (18) can be rewritten as

$$f^{n+1}(x) - f^n(x) = -\tau\frac{\delta F}{\delta \nu}(\nu^n, \mu^n, x), \tag{30}$$

for all $x \in \mathcal{X}$ $\nu^{n+1}$-a.e. By Assumption 1.6, there exists $C_1 > 0$ such that $\left\|\frac{\delta F}{\delta \nu}(\nu^n, \mu^n, \cdot)\right\|_\infty \le C_1$, for any $n \ge 0$. Hence, we obtain that

$$D_{h^*}(f^{n+1}, f^n) - D_{h^*}(f^n, f^{n+1}) \le \frac{L_{h^*}}{6}\|f^{n+1} - f^n\|_\infty^3 = \frac{L_{h^*}}{6}\tau^3\left\|\frac{\delta F}{\delta \nu}(\nu^n, \mu^n, \cdot)\right\|_\infty^3 \le \frac{L_{h^*}}{6}\tau^3 C_1^3,$$

Similarly, denoting $g^n := \frac{\delta h}{\delta \mu}(\mu^n, \cdot)$, for any $n \ge 0$, and repeating the steps above, we can prove that

$$D_{h^*}(g^{n+1}, g^n) - D_{h^*}(g^n, g^{n+1}) \le \frac{L_{h^*}}{6}\|g^{n+1} - g^n\|_\infty^3 = \frac{L_{h^*}}{6}\tau^3\left\|\frac{\delta F}{\delta \mu}(\nu^{n+1}, \mu^n, \cdot)\right\|_\infty^3 \le \frac{L_{h^*}}{6}\tau^3 C_2^3,$$

where $C_2 > 0$ exists due to Assumption 1.6.

Set $\kappa_1 := \frac{1}{6} \left( C_1^3 + C_2^3 \right) > 0$. Then,

$$D_h(\nu^n, \nu^{n+1}) - D_h(\nu^{n+1}, \nu^n) + D_h(\mu^n, \mu^{n+1}) - D_h(\mu^{n+1}, \mu^n) \leq \kappa_1 L_{h^*} \tau^3. \qquad (31)$$

Hence, using (29), (31) and Assumption 3.4, estimate (28) becomes

$$\mathrm{NI}\left( \frac{1}{N} \sum_{n=0}^{N-1} \nu^{n+1}, \frac{1}{N} \sum_{n=0}^{N-1} \mu^n \right) \leq \frac{1}{N\tau} \left( \sup_{\nu \in \mathcal{C}} D_h(\nu, \nu^0) + \sup_{\mu \in \mathcal{D}} D_h(\mu, \mu^0) \right)$$

$$+ \frac{1}{2N\tau} \sum_{n=0}^{N-1} \left( \left( D_h(\nu^n, \nu^{n+1}) - D_h(\nu^{n+1}, \nu^n) \right) + \left( D_h(\mu^n, \mu^{n+1}) - D_h(\mu^{n+1}, \mu^n) \right) \right)$$

$$+ \frac{L}{2N} \sum_{n=0}^{N-1} \left( D_h(\nu^{n+1}, \nu^n) + D_h(\mu^{n+1}, \mu^n) \right) + \frac{1}{2N} \left( F\left( \nu^N, \mu^N \right) - F(\nu^0, \mu^0) \right)$$

$$= \frac{1}{N\tau} \left( \sup_{\nu \in \mathcal{C}} D_h(\nu, \nu^0) + \sup_{\mu \in \mathcal{D}} D_h(\mu, \mu^0) \right) + \left( \frac{\kappa_1 L_{h^*}}{2} + \frac{8\kappa_2 L}{\alpha} \right) \tau^2 + \frac{M}{N}.$$

Minimizing the right-hand side over $\tau$ amounts to taking

$$\tau = \left( \frac{\sup_{\nu \in \mathcal{C}} D_h(\nu, \nu^0) + \sup_{\mu \in \mathcal{D}} D_h(\mu, \mu^0)}{2N} \right)^{1/3} \left( \frac{\kappa_1 L_{h^*}}{2} + \frac{8\kappa_2 L}{\alpha} \right)^{-1/3},$$

and since $\frac{1}{N} \leq \frac{1}{N^{2/3}}$, for any $N \geq 1$, it follows that

$$\mathrm{NI}\left( \frac{1}{N} \sum_{n=0}^{N-1} \nu^{n+1}, \frac{1}{N} \sum_{n=0}^{N-1} \mu^n \right) \leq \frac{1}{(2N)^{2/3}} \left( 3 \left( \sup_{\nu \in \mathcal{C}} D_h(\nu, \nu^0) + \sup_{\mu \in \mathcal{D}} D_h(\mu, \mu^0) \right)^{2/3} \times \right.$$

$$\left. \times \left( \frac{\kappa_1 L_{h^*}}{2} + \frac{8\kappa_2 L}{\alpha} \right)^{1/3} + 4^{1/3} M \right).$$

$\square$

## B  PROOFS OF ADDITIONAL RESULTS

In this section, we present the proofs of the additional results of the paper. We start with the proofs of Lemma A.1 and Lemma B.1, which play a key role in proving the main results. Then we continue with the proofs of some auxiliary results.

### B.1  PROOF OF LEMMA A.1

*Proof of Lemma A.1.* We will only prove the lemma for Algorithm 1 since the argument for Algorithm 2 is almost identical. From $L_\nu$-relative smoothness and the definition of $\nu^{n+1}$ in (1), for any $\nu \in \mathcal{C}$, it follows that

$$F(\nu^{n+1}, \mu^n) \leq F(\nu^n, \mu^n) + \int_{\mathcal{X}} \frac{\delta F}{\delta \nu}(\nu^n, \mu^n, x)(\nu^{n+1} - \nu^n)(\mathrm{d}x) + \left( \frac{1}{\tau} + L_\nu - \frac{1}{\tau} \right) D_h(\nu^{n+1}, \nu^n)$$

$$\leq F(\nu^n, \mu^n) + \int_{\mathcal{X}} \frac{\delta F}{\delta \nu}(\nu^n, \mu^n, x)(\nu - \nu^n)(\mathrm{d}x) + \frac{1}{\tau} D_h(\nu, \nu^n) + \left( L_\nu - \frac{1}{\tau} \right) D_h(\nu^{n+1}, \nu^n).$$

Setting $\nu = \nu^n$, we obtain that

$$F(\nu^{n+1}, \mu^n) \leq F(\nu^n, \mu^n) + \left( L_\nu - \frac{1}{\tau} \right) D_h(\nu^{n+1}, \nu^n).$$

Recall $L := \max\{L_\nu, L_\mu\} > 0$. By assumption, $\tau L \le \frac{1}{2}$, and so we get

$$\frac{1}{2\tau} D_h(\nu^{n+1}, \nu^n) \le F(\nu^n, \mu^n) - F(\nu^{n+1}, \mu^n)$$

$$= \int_0^1 \int_{\mathcal{X}} \frac{\delta F}{\delta \nu}(\nu^{n+1} + \varepsilon(\nu^n - \nu^{n+1}), \mu^n, x)(\nu^n - \nu^{n+1})(\mathrm{d}x)\mathrm{d}\varepsilon$$

$$\le 2C_1 \,\mathrm{TV}(\nu^{n+1}, \nu^n)$$

$$\le 2C_1 \sqrt{\frac{2}{\alpha} D_h(\nu^{n+1}, \nu^n)},$$

where the penultimate inequality follows from Assumption 1.6 and the last inequality follows from Assumption 1.1. Hence, since $D_h(\nu^{n+1}, \nu^n) \ge 0$, for all $n \ge 0$, we obtain that

$$D_h(\nu^{n+1}, \nu^n) \le \frac{16\tau^2 C_1^2}{\alpha}.$$

From $L_\mu$-relative smoothness and the definition of $\mu^{n+1}$ in (1), for any $\mu \in \mathcal{D}$, it follows that

$$F(\nu^n, \mu^{n+1}) \ge F(\nu^n, \mu^n) + \int_{\mathcal{X}} \frac{\delta F}{\delta \mu}(\nu^n, \mu^n, y)(\mu^{n+1} - \mu^n)(\mathrm{d}y) - \left(\frac{1}{\tau} + L_\mu - \frac{1}{\tau}\right) D_h(\mu^{n+1}, \mu^n)$$

$$\ge F(\nu^n, \mu^n) + \int_{\mathcal{X}} \frac{\delta F}{\delta \mu}(\nu^n, \mu^n, y)(\mu - \mu^n)(\mathrm{d}y) - \frac{1}{\tau} D_h(\mu, \mu^n) - \left(L_\mu - \frac{1}{\tau}\right) D_h(\mu^{n+1}, \mu^n).$$

Setting $\mu = \mu^n$, we obtain that

$$F(\nu^n, \mu^{n+1}) \ge F(\nu^n, \mu^n) - \left(L_\mu - \frac{1}{\tau}\right) D_h(\mu^{n+1}, \mu^n).$$

Using again the assumption $\tau L \le \frac{1}{2}$, we get

$$\frac{1}{2\tau} D_h(\mu^{n+1}, \mu^n) \le F(\nu^n, \mu^{n+1}) - F(\nu^n, \mu^n) \le 2C_2 \sqrt{\frac{2}{\alpha} D_h(\mu^{n+1}, \mu^n)},$$

where the last inequality follows from Assumptions 1.6 and 1.1. Hence, since $D_h(\mu^{n+1}, \mu^n) \ge 0$, for all $n \ge 0$, we obtain that

$$D_h(\mu^{n+1}, \mu^n) \le \frac{16\tau^2 C_2^2}{\alpha}.$$

$\square$

## B.2 PROOF OF LEMMA B.1

**Lemma B.1** (Three-point inequality). *Let Assumption 1.1 hold. Let $G : \mathcal{E} \to \mathbb{R}$ be convex and $G \in \mathfrak{C}^1(\mathcal{E})$. For all $\mu \in \mathcal{E}$, suppose that there exists $\bar{\nu} \in \mathcal{E}$ such that*

$$\bar{\nu} \in \underset{\nu \in \mathcal{E}}{\arg\min}\{G(\nu) + D_h(\nu, \mu)\}.$$

*Then, for any $\nu \in \mathcal{E}$, we have*

$$G(\nu) + D_h(\nu, \mu) \ge G(\bar{\nu}) + D_h(\bar{\nu}, \mu) + D_h(\nu, \bar{\nu}).$$

*Proof.* From Definition 1.2, we have

$$D_h(\nu, \mu) = h(\nu) - h(\mu) - \int_{\mathcal{X}} \frac{\delta h}{\delta \mu}(\mu, y)(\nu - \mu)(\mathrm{d}y),$$

and hence, for any $\mu \in \mathcal{K}$, and all $y \in \mathcal{X}$ Lebesgue a.e., we have

$$\left(\frac{\delta D_h}{\delta \nu}(\nu, \mu, y)\right)\bigg|_{\nu=\bar{\nu}} = \frac{\delta h}{\delta \nu}(\bar{\nu}, y) - \frac{\delta h}{\delta \mu}(\mu, y).$$

Therefore, for any $\mu \in \mathcal{K}$, we have that

$$
D_{D_h(\cdot,\mu)}(\nu, \bar{\nu}) = D_h(\nu, \mu) - D_h(\bar{\nu}, \mu) - \int_{\mathcal{X}} \left( \frac{\delta D_h}{\delta \nu}(\nu, \mu, y) \right) \Big|_{\nu = \bar{\nu}} (\nu - \bar{\nu})(\mathrm{d}y)
$$

$$
= D_h(\nu, \mu) - D_h(\bar{\nu}, \mu) - \int_{\mathcal{X}} \frac{\delta h}{\delta \nu}(\bar{\nu}, y)(\nu - \bar{\nu})(\mathrm{d}y) + \int_{\mathcal{X}} \frac{\delta h}{\delta \mu}(\mu, y)(\nu - \bar{\nu})(\mathrm{d}y)
$$

$$
= h(\nu) - h(\bar{\nu}) - \int_{\mathcal{X}} \frac{\delta h}{\delta \mu}(\mu, y)(\nu - \mu)(\mathrm{d}y) + \int_{\mathcal{X}} \frac{\delta h}{\delta \mu}(\mu, y)(\bar{\nu} - \mu)(\mathrm{d}y)
$$

$$
- \int_{\mathcal{X}} \frac{\delta h}{\delta \nu}(\bar{\nu}, y)(\nu - \bar{\nu})(\mathrm{d}y) + \int_{\mathcal{X}} \frac{\delta h}{\delta \mu}(\mu, y)(\nu - \bar{\nu})(\mathrm{d}y)
$$

$$
= h(\nu) - h(\bar{\nu}) - \int_{\mathcal{X}} \frac{\delta h}{\delta \nu}(\bar{\nu}, y)(\nu - \bar{\nu})(\mathrm{d}y)
$$

$$
= D_h(\nu, \bar{\nu}).
$$

Given $\mu \in \mathcal{K}$, if we denote $g(\nu) := G(\nu) + D_h(\nu, \mu)$, then by linearity of flat derivative, we further obtain that

$$
D_g(\nu, \bar{\nu}) = D_{G(\cdot) + D_h(\cdot, \mu)}(\nu, \bar{\nu}) = D_G(\nu, \bar{\nu}) + D_{D_h(\cdot, \mu)}(\nu, \bar{\nu}) = D_G(\nu, \bar{\nu}) + D_h(\nu, \bar{\nu}) \geq D_h(\nu, \bar{\nu}),
$$

since $D_G(\nu, \bar{\nu}) \geq 0$ by convexity of $G$. By optimality of $\bar{\nu}$, the first-order condition $\frac{\delta g}{\delta \nu}(\bar{\nu}, y) = $ constant holds for all $y \in \mathcal{X}$ Lebesgue a.e., and hence

$$
g(\nu) - g(\bar{\nu}) - D_g(\nu, \bar{\nu}) = 0.
$$

Therefore, we obtain that

$$
g(\nu) = g(\bar{\nu}) + D_g(\nu, \bar{\nu}) \geq g(\bar{\nu}) + D_h(\nu, \bar{\nu}),
$$

which is the desired inequality. $\qquad\square$

### B.3 Proofs of auxiliary results

In this subsection, we start by proving the convexity characterization $h$ via its flat derivative.

**Lemma B.2** (Strong convexity of $h$). *Let Assumption 1.1 hold. Then there exists $\alpha > 0$ such that for any $\nu, \nu' \in \mathcal{E}$,*

$$
h(\nu') - h(\nu) \geq \int_{\mathcal{X}} \frac{\delta h}{\delta \nu}(\nu, x)(\nu' - \nu)(\mathrm{d}x) + \frac{\alpha}{2} \mathrm{TV}^2(\nu', \nu).
$$

*Proof.* Let $\varepsilon \in [0, 1]$ and $\nu^\varepsilon = \nu + \varepsilon(\nu' - \nu)$. Since $h$ is strongly convex, and by Definition G.1,

$$
\varepsilon \left( h(\nu') - h(\nu) \right) - \frac{\alpha}{2} \varepsilon (1 - \varepsilon) \mathrm{TV}^2(\nu', \nu) \geq h(\nu^\varepsilon) - h(\nu)
$$

$$
= \varepsilon \int_0^1 \int_{\mathcal{X}} \frac{\delta h}{\delta \nu}(\nu^{s\varepsilon}, x)(\nu' - \nu)(\mathrm{d}x)\mathrm{d}s.
$$

Dividing by $\varepsilon$ and passing to the limit $\varepsilon \searrow 0$ via dominated convergence theorem gives the conclusion since $\frac{\delta h}{\delta \nu}$ is bounded and continuous. $\qquad\square$

Now we prove that the mirror updates in Algorithms 1 and 2 satisfy iterative schemes on the dual space.

**Proposition B.3** (MDA dual iteration). *For each $n \geq 0$, let $F(\cdot, \mu^n) \in \mathfrak{C}^1(\mathcal{C})$, $F(\nu^n, \cdot) \in \mathfrak{C}^1(\mathcal{D})$. Moreover, let Assumption 1.1 hold. Then the minimizer-maximizer pair $(\nu^{n+1}, \mu^{n+1}) \in \mathcal{C} \times \mathcal{D}$ of each one of Algorithms 1 and 2 satisfies the corresponding dual iterative update*

$$
\begin{cases}
\frac{\delta h}{\delta \nu}(\nu^{n+1}, \cdot) - \frac{\delta h}{\delta \nu}(\nu^n, \cdot) = -\tau \frac{\delta F}{\delta \nu}(\nu^n, \mu^n, \cdot) + C_{n,1}, & \nu^{n+1} - a.e., \\
\frac{\delta h}{\delta \mu}(\mu^{n+1}, \cdot) - \frac{\delta h}{\delta \mu}(\mu^n, \cdot) = \tau \frac{\delta F}{\delta \mu}(\nu^n, \mu^n, \cdot) + C_{n,2}, & \mu^{n+1} - a.e.,
\end{cases}
$$

$$
\begin{cases}
\frac{\delta h}{\delta \nu}(\nu^{n+1}, \cdot) - \frac{\delta h}{\delta \nu}(\nu^n, \cdot) = -\tau \frac{\delta F}{\delta \nu}(\nu^n, \mu^n, \cdot) + C_{n,3}, & \nu^{n+1} - a.e., \\
\frac{\delta h}{\delta \mu}(\mu^{n+1}, \cdot) - \frac{\delta h}{\delta \mu}(\mu^n, \cdot) = \tau \frac{\delta F}{\delta \mu}(\nu^{n+1}, \mu^n, \cdot) + C_{n,4}, & \mu^{n+1} - a.e.,
\end{cases}
$$

*where $C_{n,1}, C_{n,2}, C_{n,3}, C_{n,4} \in \mathbb{R}$.*

*Proof.* We present the proof only for Algorithm 1, as the proof for Algorithm 2 is identical. For convenience, define

$$G(\nu) := \int_{\mathcal{X}} \frac{\delta F}{\delta \nu}(\nu^n, \mu^n, x)(\nu - \nu^n)(\mathrm{d}x), \quad \nu \in \mathcal{C}.$$

Since $\nu^{n+1}$ is the minimizer in Algorithm 1, we have

$$G(\nu^{n+1}) + \frac{1}{\tau}D_h(\nu^{n+1}, \nu^n) \le G(\nu) + \frac{1}{\tau}D_h(\nu, \nu^n), \quad \forall \nu \in \mathcal{C}.$$

Because $\mathcal{C}$ is convex, fix any $\tilde{\nu} \in \mathcal{C}$ and take $\nu = \nu^{n+1} + \varepsilon(\tilde{\nu} - \nu^{n+1}) \in \mathcal{C}$. Then

$$G(\nu^{n+1}) + \frac{1}{\tau}D_h(\nu^{n+1}, \nu^n) \le G(\nu^{n+1} + \varepsilon(\tilde{\nu} - \nu^{n+1})) + \frac{1}{\tau}D_h(\nu^{n+1} + \varepsilon(\tilde{\nu} - \nu^{n+1}), \nu^n),$$

which, using linearity of $G$ and rearranging, becomes

$$\varepsilon(G(\tilde{\nu}) - G(\nu^{n+1})) + \frac{1}{\tau}\left(D_h(\nu^{n+1} + \varepsilon(\tilde{\nu} - \nu^{n+1}), \nu^n) - D_h(\nu^{n+1}, \nu^n)\right) \ge 0.$$

Dividing by $\varepsilon$ and letting $\varepsilon \searrow 0$, Definition G.1 yields

$$G(\tilde{\nu}) - G(\nu^{n+1}) + \frac{1}{\tau}\int_{\mathcal{X}} \frac{\delta D_h(\cdot, \nu^n)}{\delta \nu}(\nu^{n+1}, x)(\tilde{\nu} - \nu^{n+1})(\mathrm{d}x) \ge 0.$$

By the definition of Bregman divergence and flat derivative,

$$\frac{\delta D_h(\cdot, \nu^n)}{\delta \nu}(\nu^{n+1}, x) = \frac{\delta h}{\delta \nu}(\nu^{n+1}, x) - \frac{\delta h}{\delta \nu}(\nu^n, x).$$

Hence,

$$\int_{\mathcal{X}} \left(\frac{\delta F}{\delta \nu}(\nu^n, \mu^n, x) + \frac{1}{\tau}\left(\frac{\delta h}{\delta \nu}(\nu^{n+1}, x) - \frac{\delta h}{\delta \nu}(\nu^n, x)\right)\right)(\tilde{\nu} - \nu^{n+1})(\mathrm{d}x) \ge 0.$$

Since $\tilde{\nu}$ is arbitrary, we conclude that

$$\frac{\delta F}{\delta \nu}(\nu^n, \mu^n, \cdot) + \frac{1}{\tau}\left(\frac{\delta h}{\delta \nu}(\nu^{n+1}, \cdot) - \frac{\delta h}{\delta \nu}(\nu^n, \cdot)\right) = \text{constant}, \quad \nu^{n+1}\text{-a.e.}$$

An analogous argument gives the optimality condition for the maximizer $\mu^{n+1}$. $\square$

We show that, under Assumption 1.5 and 3.3, the second-order flat derivatives $\frac{\delta^2 F}{\delta \nu^2}, -\frac{\delta^2 F}{\delta \mu^2}$ are non-negative and bounded above by $\frac{\delta^2 h}{\delta \nu^2}, \frac{\delta^2 h}{\delta \mu^2}$ multiplied by the respective smoothness constants.

**Lemma B.4** (Uniform boundedness of second order flat derivatives of $F$). *Let Assumption 1.5 and 3.3 hold. Suppose that $F(\cdot, \mu) \in \mathfrak{C}^2(\mathcal{C})$, $F(\nu, \cdot) \in \mathfrak{C}^2(\mathcal{D})$ and $h \in \mathfrak{C}^2(\mathcal{E})$ (cf. (48)). Then,*

$$0 \le \int_0^1 \int_{\mathcal{X}} \int_0^\varepsilon \int_{\mathcal{X}} \frac{\delta^2 F}{\delta \nu^2}(\nu + \eta(\nu' - \nu), \mu, x, x')(\nu' - \nu)(\mathrm{d}x')\mathrm{d}\eta(\nu' - \nu)(\mathrm{d}x)\mathrm{d}\varepsilon$$

$$\le L_\nu \int_0^1 \int_{\mathcal{X}} \int_0^\varepsilon \int_{\mathcal{X}} \frac{\delta^2 h}{\delta \nu^2}(\nu + \eta(\nu' - \nu), x, x')(\nu' - \nu)(\mathrm{d}x')\mathrm{d}\eta(\nu' - \nu)(\mathrm{d}x)\mathrm{d}\varepsilon,$$

$$0 \le -\int_0^1 \int_{\mathcal{X}} \int_0^\varepsilon \int_{\mathcal{X}} \frac{\delta^2 F}{\delta \mu^2}(\nu, \mu + \eta(\mu' - \mu), y, y')(\mu' - \mu)(\mathrm{d}y')\mathrm{d}\eta(\mu' - \mu)(\mathrm{d}y)\mathrm{d}\varepsilon$$

$$\le L_\mu \int_0^1 \int_{\mathcal{X}} \int_0^\varepsilon \int_{\mathcal{X}} \frac{\delta^2 h}{\delta \mu^2}(\mu + \eta(\mu' - \mu), y, y')(\mu' - \mu)(\mathrm{d}y')\mathrm{d}\eta(\mu' - \mu)(\mathrm{d}y)\mathrm{d}\varepsilon.$$

*Proof.* We observe that combining relative smoothness and convexity for $\nu \mapsto F(\nu, \mu)$ gives that for some $L_\nu > 0$, any $\nu, \nu' \in \mathcal{C}$ and any $\mu, \mu' \in \mathcal{D}$, we have

$$0 \le F(\nu', \mu) - F(\nu, \mu) - \int_{\mathcal{X}} \frac{\delta F}{\delta \nu}(\nu, \mu, x)(\nu' - \nu)(\mathrm{d}x) \le L_\nu D_h(\nu', \nu). \tag{32}$$

Since $\nu \mapsto F(\nu, \mu)$, $\mu \mapsto F(\nu, \mu)$, and $h$ admit second-order flat derivative (cf. (48)) on $\mathcal{C}, \mathcal{D}$ and $\mathcal{E}$, respectively, from (32), we obtain

$$
0 \leq \int_0^1 \int_{\mathcal{X}} \int_0^{\varepsilon} \int_{\mathcal{X}} \frac{\delta^2 F}{\delta \nu^2} \left(\nu + \eta(\nu' - \nu), \mu, x, x'\right) (\nu' - \nu)(\mathrm{d}x') \mathrm{d}\eta (\nu' - \nu)(\mathrm{d}x) \mathrm{d}\varepsilon
$$

$$
\leq L_\nu \int_0^1 \int_{\mathcal{X}} \int_0^{\varepsilon} \int_{\mathcal{X}} \frac{\delta^2 h}{\delta \nu^2} \left(\nu + \eta(\nu' - \nu), x, x'\right) (\nu' - \nu)(\mathrm{d}x') \mathrm{d}\eta (\nu' - \nu)(\mathrm{d}x) \mathrm{d}\varepsilon.
$$

The analogous inequalities are similarly obtained for relative smoothness and relative concavity. $\quad\square$

When $F$ is strongly-convex-strongly-concave relative to $h$ and Assumption 1.1 holds, it can be shown that $(\nu^*, \mu^*)$ is the unique MNE of (1) (see the proof of (Lascu et al., 2025, Lemma 6)). Moreover, based on relative convexity-concavity of $F$, we prove in Lemma B.6 that the NI error satisfies a type of "quadratic growth" inequality relative to $h$.

**Assumption B.5** (Relative convexity-concavity). *Assume $F$ is $(\ell_\nu, \ell_\mu)$-strongly convex-concave relative to $h$, i.e., there exist $\ell_\nu, \ell_\mu > 0$ such that for any $\nu, \nu' \in \mathcal{C}$ and $\mu, \mu' \in \mathcal{D}$, we have*

$$
D_{F(\cdot, \mu)}(\nu', \nu) = F(\nu', \mu) - F(\nu, \mu) - \int_{\mathcal{X}} \frac{\delta F}{\delta \nu}(\nu, \mu, x)(\nu' - \nu)(\mathrm{d}x) \geq \ell_\nu D_h(\nu', \nu), \quad (33)
$$

$$
D_{F(\nu, \cdot)}(\mu', \mu) = F(\nu, \mu') - F(\nu, \mu) - \int_{\mathcal{X}} \frac{\delta F}{\delta \mu}(\nu, \mu, y)(\mu' - \mu)(\mathrm{d}y) \leq -\ell_\mu D_h(\mu', \mu). \quad (34)
$$

**Lemma B.6** ("Quadratic growth" of NI error relative to $h$). *Suppose that Assumption 1.1 and B.5 hold. Then, for all $(\nu, \mu) \in \mathcal{C} \times \mathcal{D}$, it holds that*

$$
\mathrm{NI}(\nu, \mu) \geq \ell \left(D_h(\nu, \nu^*) + D_h(\mu, \mu^*)\right),
$$

*where $\ell := \min\{\ell_\nu, \ell_\mu\}$.*

**Remark B.7.** *We refer to Lemma B.6 as "quadratic growth" of NI error relative to $h$ due to the similar notion of quadratic growth of a convex function relative to the squared Euclidean norm on $\mathbb{R}^d$ (see e.g. (Anitescu, 2000)).*

*Proof.* Let $(\nu, \mu) \in \mathcal{C} \times \mathcal{D}$. Since $F$ is $\ell_\nu$-strongly convex in $\nu$ and $\ell_\mu$-strongly concave in $\mu$, it follows that

$$
F(\nu, \mu^*) - F(\nu^*, \mu^*) \geq \int_{\mathcal{X}} \frac{\delta F}{\delta \nu}(\nu^*, \mu^*, x)(\nu - \nu^*)(\mathrm{d}x) + \ell_\nu D_h(\nu, \nu^*),
$$

$$
F(\nu^*, \mu) - F(\nu^*, \mu^*) \leq \int_{\mathcal{X}} \frac{\delta F}{\delta \mu}(\nu^*, \mu^*, y)(\mu - \mu^*)(\mathrm{d}y) - \ell_\mu D_h(\mu, \mu^*).
$$

Since $(\nu^*, \mu^*)$ is the MNE of $F$, we have

$$
\frac{\delta F}{\delta \nu}(\nu^*, \mu^*, x) = \text{constant}, \quad \frac{\delta F}{\delta \mu}(\nu^*, \mu^*, y) = \text{constant},
$$

for all $(x, y) \in \mathcal{X} \times \mathcal{X}$ $(\nu^*, \mu^*)$-a.e. Hence, adding the inequalities above and using the definition of NI error, we get

$$
\mathrm{NI}(\nu, \mu) \geq \ell \left(D_h(\nu, \nu^*) + D_h(\mu, \mu^*)\right).
$$

$\quad\square$

By Lemma B.6, the time-averaged iterates $\left(\frac{1}{N} \sum_{n=0}^{N-1} \nu^n, \frac{1}{N} \sum_{n=0}^{N-1} \mu^n\right)$ converge in Bregman divergence to the unique MNE $(\nu^*, \mu^*)$ of (1) with the rates proved in Theorem 2.1 and Theorem 3.6, respectively.

We now check that the condition $\sup_{\nu \in \mathcal{C}} D_h(\nu, \nu^0) + \sup_{\mu \in \mathcal{D}} D_h(\mu, \mu^0) < \infty$ required in Theorems 2.1 and 3.6 is satisfied in the specific cases of Examples 1.3 and 1.4.

**Lemma B.8.** *Let Assumption 1.6 hold and let $h$ denote the relative entropy from Example 1.3. Suppose $\nu_0, \mu_0 \in \mathcal{E}$. Then the iterates produced by Algorithms 1 and 2 satisfy*

$$(\nu^n, \mu^n)_{n \geq 0} \subset \mathcal{E}.$$

*Furthermore,*

$$\sup_{\nu \in \mathcal{C}} \mathrm{KL}(\nu, \nu^0) + \sup_{\mu \in \mathcal{D}} \mathrm{KL}(\mu, \mu^0) < \infty.$$

*Proof.* We provide the proof only for Algorithm 1, as the argument for the other algorithm is essentially the same. Since $\nu_0, \mu_0 \in \mathcal{E}$, there exists $\beta_0 > 0$ such that $\nu_0, \mu_0 \in \mathcal{E}_{\beta_0}$. Using the flat derivative formula (5), the first-order optimality condition in Proposition B.3 gives

$$\begin{cases} \log \frac{\nu^1(x)}{\pi(x)} - \log \frac{\nu^0(x)}{\pi(x)} = -\tau \frac{\delta F}{\delta \nu}(\nu^0, \mu^0, x) - \log \int_{\mathcal{X}} e^{-\tau \frac{\delta F}{\delta \nu}(\nu^0, \mu^0, x)} \frac{\nu^0(x)}{\pi(x)} \pi(x) \mathrm{d}x, \\ \log \frac{\mu^1(y)}{\pi(y)} - \log \frac{\mu^0(y)}{\pi(y)} = \tau \frac{\delta F}{\delta \mu}(\nu^0, \mu^0, y) - \log \int_{\mathcal{X}} e^{\tau \frac{\delta F}{\delta \mu}(\nu^0, \mu^0, y)} \frac{\mu^0(y)}{\pi(y)} \pi(y) \mathrm{d}y, \end{cases}$$

for all $x, y \in \mathcal{X}$ a.e. Taking the sup-norm on both sides over $x, y$ and using the assumptions gives

$$\left\| \log \frac{\nu^1(\cdot)}{\pi(\cdot)} \right\|_{L^\infty(\mathcal{X})} \leq 2\beta_0 + 2\tau C_1,$$

$$\left\| \log \frac{\mu^1(\cdot)}{\pi(\cdot)} \right\|_{L^\infty(\mathcal{X})} \leq 2\beta_0 + 2\tau C_2,$$

so $(\nu^1, \mu^1) \subset \mathcal{E}_{\beta_1}$, with $\beta_1 = 2\beta_0 + 2\tau \max\{C_1, C_2\}$, and inductively, $(\nu^n, \mu^n)_{n \geq 0} \subset \mathcal{E}$. Therefore, for any $\nu, \mu \in \mathcal{E}$, there exists $\hat{\beta} > 0$ such that $\nu, \mu \in \mathcal{E}_{\hat{\beta}}$, and so

$$\begin{aligned} \mathrm{KL}(\nu, \nu^0) + \mathrm{KL}(\mu, \mu^0) &= \int_{\mathcal{X}} \left( \log \frac{\nu(x)}{\pi(x)} - \log \frac{\nu^0(x)}{\pi(x)} \right) \nu(x) \mathrm{d}x \\ &\quad + \int_{\mathcal{X}} \left( \log \frac{\mu(y)}{\pi(y)} - \log \frac{\mu^0(y)}{\pi(y)} \right) \mu(y) \mathrm{d}y \\ &\leq \hat{\beta} + \beta_0, \end{aligned}$$

hence the conclusion. $\square$

**Lemma B.9.** *Let Assumption 1.6 hold and let $h$ denote the $\chi^2$-divergence from Example 1.4. Suppose $\nu_0, \mu_0 \in \mathcal{F}$. Then the iterates produced by Algorithms 1 and 2 satisfy*

$$(\nu^n, \mu^n)_{n \geq 0} \subset \mathcal{F}.$$

*Furthermore,*

$$\frac{1}{2} \sup_{\nu \in \mathcal{C}} \left\| \frac{\nu(\cdot)}{\pi(\cdot)} - \frac{\nu^0(\cdot)}{\pi(\cdot)} \right\|_{L_\pi^2(\mathcal{X})}^2 + \frac{1}{2} \sup_{\mu \in \mathcal{D}} \left\| \frac{\mu(\cdot)}{\pi(\cdot)} - \frac{\mu^0(\cdot)}{\pi(\cdot)} \right\|_{L_\pi^2(\mathcal{X})}^2 < \infty.$$

*Proof.* We provide the proof only for Algorithm 1, as the argument for the other algorithm is essentially the same. Since $\nu_0, \mu_0 \in \mathcal{F}$, there exists $\eta_0 > 0$ such that $\nu_0, \mu_0 \in \mathcal{F}_{\eta_0}$. The first-order condition (see e.g., (Bonnans & Shapiro, 2000, Section 5.1.1)) shows that for a.e. $x, y \in \mathcal{X}$,

$$\left\langle \frac{\delta F}{\delta \nu}(\nu^0, \mu^0, \cdot) + \frac{1}{\tau} \left( \frac{\mathrm{d}\nu^1}{\mathrm{d}\pi} - \frac{\mathrm{d}\nu^0}{\mathrm{d}\pi} \right), \phi - \frac{\mathrm{d}\nu^1}{\mathrm{d}\pi} \right\rangle_{L_\pi^2} \geq 0, \quad \forall \phi \in \mathfrak{F},$$

$$\left\langle \frac{\delta F}{\delta \mu}(\nu^0, \mu^0, \cdot) - \frac{1}{\tau} \left( \frac{\mathrm{d}\mu^1}{\mathrm{d}\pi} - \frac{\mathrm{d}\mu^0}{\mathrm{d}\pi} \right), \phi - \frac{\mathrm{d}\mu^1}{\mathrm{d}\pi} \right\rangle_{L_\pi^2} \geq 0, \quad \forall \phi \in \mathfrak{F},$$

where $\langle \cdot, \cdot \rangle_{L_\pi^2}$ is the inner product on $L_\pi^2(\mathcal{X})$, and $\mathfrak{F}$ is the nonempty closed convex set defined by

$$\mathfrak{F} = \left\{ \phi \in L_\pi^2(\mathcal{X}) \,\middle|\, \phi \geq 0 \ \pi\text{-a.e. on } \mathcal{X} \text{ and } \int \phi(x) \pi(\mathrm{d}x) = 1 \right\}.$$

Define the projection map $\Pi_{\mathfrak{F}} : L^2_{\pi}(\mathcal{X}) \mapsto \mathfrak{F}$ such that $\Pi_{\mathfrak{F}}(\varphi) = \arg\min_{\phi \in \mathfrak{F}} \|\phi - \varphi\|_{L^2_{\pi}(\mathcal{X})}$ for all $\varphi \in L^2_{\pi}(\mathcal{X})$, which satisfies

$$\langle \Pi(\varphi) - \varphi, \phi - \Pi(\varphi) \rangle_{L^2_{\pi}} \geq 0, \quad \forall \phi \in \mathfrak{F}.$$

Then

$$\frac{\mathrm{d}\nu^1}{\mathrm{d}\pi} = \Pi_{\mathfrak{F}}\left( \frac{\mathrm{d}\nu^0}{\mathrm{d}\varrho} - \tau \frac{\delta F}{\delta \nu}(\nu^0, \mu^0, \cdot) \right),$$

$$\frac{\mathrm{d}\mu^1}{\mathrm{d}\pi} = \Pi_{\mathfrak{F}}\left( \frac{\mathrm{d}\mu^0}{\mathrm{d}\varrho} + \tau \frac{\delta F}{\delta \mu}(\nu^0, \mu^0, \cdot) \right).$$

Note $\|\Pi_{\mathfrak{F}}(\varphi_1) - \Pi_{\mathfrak{F}}(\varphi_2)\|_{L^2_{\pi}(\mathcal{X})} \leq \|\varphi_1 - \varphi_2\|_{L^2_{\pi}(\mathcal{X})}$ for all $\varphi_1, \varphi_2 \in L^2_{\pi}(\mathcal{X})$ (see e.g., (Ciarlet, 2013, Theorem 4.3-1)). Moreover, since $\frac{\mathrm{d}\nu^0}{\mathrm{d}\pi} = \Pi_{\mathfrak{F}}\left(\frac{\mathrm{d}\nu^0}{\mathrm{d}\pi}\right)$, $\frac{\mathrm{d}\mu^0}{\mathrm{d}\pi} = \Pi_{\mathfrak{F}}\left(\frac{\mathrm{d}\mu^0}{\mathrm{d}\pi}\right)$, for a.e. $x, y \in \mathcal{X}$,

$$\left\| \frac{\mathrm{d}\nu^1}{\mathrm{d}\pi} \right\|_{L^2_{\pi}(\mathcal{X})} \leq \left\| \frac{\mathrm{d}\nu^0}{\mathrm{d}\pi} \right\|_{L^2_{\pi}(\mathcal{X})} + \tau \left\| \frac{\delta F}{\delta \nu}(\nu^0, \mu^0, \cdot) \right\|_{L^2_{\pi}(\mathcal{X})} \leq \eta_0 + \tau C_1^2,$$

$$\left\| \frac{\mathrm{d}\mu^1}{\mathrm{d}\pi} \right\|_{L^2_{\pi}(\mathcal{X})} \leq \left\| \frac{\mathrm{d}\mu^0}{\mathrm{d}\pi} \right\|_{L^2_{\pi}(\mathcal{X})} + \tau \left\| \frac{\delta F}{\delta \mu}(\nu^0, \mu^0, \cdot) \right\|_{L^2_{\pi}(\mathcal{X})} \leq \eta_0 + \tau C_2^2,$$

so $(\nu^1, \mu^1) \subset \mathcal{F}_{\eta_1}$, with $\eta_1 = \eta_0 + \tau \max\{C_1^2, C_2^2\}$, and inductively, $(\nu^n, \mu^n)_{n \geq 0} \subset \mathcal{F}$. Therefore, for any $\nu, \mu \in \mathcal{F}$, there exists $\hat{\eta} > 0$ such that $\nu, \mu \in \mathcal{F}_{\hat{\eta}}$, and so

$$\frac{1}{2} \left\| \frac{\nu(\cdot)}{\pi(\cdot)} - \frac{\nu^0(\cdot)}{\pi(\cdot)} \right\|^2_{L^2_{\pi}(\mathcal{X})} + \frac{1}{2} \left\| \frac{\mu(\cdot)}{\pi(\cdot)} - \frac{\mu^0(\cdot)}{\pi(\cdot)} \right\|^2_{L^2_{\pi}(\mathcal{X})} \leq \hat{\eta}^2 + \eta_0^2,$$

hence the conclusion. $\qquad\square$

## C    VERIFICATION OF ASSUMPTION 1.5, 1.6, 3.3 AND 3.4 FOR EXAMPLE 1.2

In this section we verify that Assumption 1.5, 1.6, 3.3 and 3.4 are satisfied by the objective function $F$ in Example 1.2.

**Proposition C.1** (Verification of assumptions for Example 1.2). *Let $\mathcal{Y}, \mathcal{Z} \subset \mathbb{R}^d$, with $\hat{\xi} \in \mathcal{P}(\mathcal{Y})$ and $\xi \in \mathcal{P}(\mathcal{Z})$. Suppose $T_{\theta} : \mathcal{Z} \to \mathcal{Y}$ is measurable with $\theta \in \Theta \subset \mathbb{R}^d$, and $D_w : \mathcal{Y} \to \mathbb{R}$ is uniformly bounded and measurable with $w \in \mathcal{W} \subset \mathbb{R}^d$. Then Assumptions 1.5, 1.6, 3.3 and 3.4 are satisfied by the objective*

$$F(\nu, \mu) := \int_{\mathcal{W}} \int_{\Theta} f(\theta, w) \nu(\mathrm{d}\theta) \mu(\mathrm{d}w)$$

*from Example 1.2.*

*Proof.* By Definition G.1,

$$\frac{\delta F}{\delta \nu}(\nu, \mu, \theta) = \int_{\mathcal{W}} f(\theta, w) \mu(\mathrm{d}w),$$

and

$$\frac{\delta F}{\delta \mu}(\nu, \mu, w) = \int_{\Theta} f(\theta, w) \nu(\mathrm{d}\theta).$$

Therefore, Assumption 1.5 holds with equality, and Assumption 3.3 holds with equality with $L_{\nu} = L_{\mu} = 0$. Since $D_w$ is uniformly bounded by some $M_D > 0$, we have

$$|f(\theta, w)| = \left| \int_{\mathcal{Y}} D_w(y) \left( T_{\theta} \# \xi - \hat{\xi} \right)(\mathrm{d}y) \right|$$

$$\leq \int_{\mathcal{Y}} |D_w(y)| \left( T_{\theta} \# \xi \right)(\mathrm{d}y) + \int_{\mathcal{Y}} |D_w(y)| \hat{\xi}(\mathrm{d}y) \leq 2M_D,$$

where the last inequality holds because $\hat{\xi} \in \mathcal{P}(\mathcal{Y})$ and, since $\xi \in \mathcal{P}(\mathcal{Z})$, we have $T_{\theta} \# \xi \in \mathcal{P}(\mathcal{Y})$. Finally, we have

$$|F(\nu, \mu)| \leq 2M_D, \quad \left| \frac{\delta F}{\delta \nu}(\nu, \mu, \theta) \right| \leq 2M_D, \quad \left| \frac{\delta F}{\delta \mu}(\nu, \mu, w) \right| \leq 2M_D,$$

for all $\nu, \mu \in \mathcal{P}(\mathcal{X})$ and all $\theta \in \Theta, w \in \mathcal{W}$. Hence, Assumption 1.6 and 3.4 hold with $M = C_1 = C_2 = 2M_D$. $\qquad\square$

# D    ADVERSARIAL TRAINING OF MEAN-FIELD NEURAL NETWORKS

Let $\mathcal{Y} \subset \mathbb{R}$ and $\mathcal{Z} \subset \mathbb{R}^{d-1}$ be compact with $\hat{\mu} \in \mathcal{P}(\mathcal{Y} \times \mathcal{Z})$ representing the training data $(y, z) \in \mathcal{Y} \times \mathcal{Z}$. Let $(w, b) \in \mathbb{R}^{d-1} \times \mathbb{R}$ be the parameters of the neural network and let $\varphi : \mathbb{R} \to \mathbb{R}$ be a bounded, continuous, non-constant activation function. For $x := (w, b) \in \mathbb{R}^d$ and $z \in \mathbb{R}^{d-1}$, define the function $\hat{\varphi}(x, z) := \ell(b)\varphi(w \cdot z)$, where $\ell : \mathbb{R} \to [-K, K]$ is a clipping function with clipping threshold $K > 0$. The training of the two-layer neural network aims to find the optimal set of parameters $\{x_i\}_{i=1}^N$ which minimize the non-convex $L^2$-loss function

$$F_N^0(x_1, ..., x_N) := \frac{1}{2} \int_{\mathcal{Y} \times \mathcal{Z}} \left| y - \frac{1}{N} \sum_{i=1}^N \hat{\varphi}(x_i, z) \right|^2 \hat{\mu}(\mathrm{d}y, \mathrm{d}z). \tag{35}$$

Instead of solving the non-convex minimization problem (35), we lift it to space of probability measures and consider the mean-field optimization problem (see e.g. (Hu et al., 2021, Section 3) and the references therein)

$$\min_{\nu \in \mathcal{P}(\mathbb{R}^d)} F^0(\nu), \quad \text{with } F^0(\nu) := \frac{1}{2} \int_{\mathcal{Y} \times \mathcal{Z}} \left| y - \mathbb{E}^{X \sim \nu}[\hat{\varphi}(X, z)] \right|^2 \hat{\mu}(\mathrm{d}y, \mathrm{d}z).$$

To account for potential attacks by an adversary aiming to manipulate the training data $\hat{\mu}$, we minimize over the parameter distribution $\nu$, considering the "worst-case" perturbation of $\hat{\mu}$. This leads to the following mean-field min-max game

$$\min_{\nu \in \mathcal{P}(\mathbb{R}^d)} \max_{\mu \in \mathcal{P}(\mathcal{Y} \times \mathcal{Z})} F^0(\nu, \mu) - \mathrm{TV}^2(\mu, \hat{\mu}), \tag{36}$$

where $\mathrm{TV}^2$ denotes the squared total variation distance, which represents the cost incurred by the adversary to alter the original training data $\hat{\mu}$. The resulting objective $F(\nu, \mu) := F^0(\nu, \mu) - \mathrm{TV}^2(\mu, \hat{\mu})$ is a non-linear function covered by our general framework. The choice of the incurred cost in (36) is, to an extent, arbitrary, and we focus here on $\mathrm{TV}^2$ due to its convenience for verifying our assumptions. Alternative cost functions include the Wasserstein distance (Bai et al., 2023; Trillos & Trillos, 2023) and the KL divergence (Si et al., 2023).

**Proposition D.1** (Verification of assumptions for Example D). *Let Assumption 1.1 hold. Let $\mathcal{Y} \subset \mathbb{R}$ and $\mathcal{Z} \subset \mathbb{R}^{d-1}$ be compact with $\hat{\mu} \in \mathcal{P}(\mathcal{Y} \times \mathcal{Z})$. For $x := (w, b) \in \mathbb{R}^d$ and $z \in \mathbb{R}^{d-1}$, let $\hat{\varphi}(x, z) := \ell(b)\varphi(w \cdot z)$, where $\ell : \mathbb{R} \to [-K, K]$ is a clipping function with clipping threshold $K > 0$ and $\varphi : \mathbb{R} \to \mathbb{R}$ is a bounded, continuous, non-constant function. Then Assumptions 1.5, 1.6, 3.3 and 3.4 are satisfied by the objective*

$$F(\nu, \mu) = \frac{1}{2} \int_{\mathcal{Y} \times \mathcal{Z}} \left| y - \mathbb{E}^{X \sim \nu}[\hat{\varphi}(X, z)] \right|^2 \mu(\mathrm{d}y, \mathrm{d}z) - \mathrm{TV}^2(\mu, \hat{\mu}).$$

*Proof.* Observe that by linearity of the expectation in $\nu$ and convexity of $|\cdot|^2$, the function

$$F^0(\nu, \mu) = \frac{1}{2} \int_{\mathcal{Y} \times \mathcal{Z}} \left| y - \mathbb{E}^{X \sim \nu}[\hat{\varphi}(X, z)] \right|^2 \mu(\mathrm{d}y, \mathrm{d}z)$$

satisfies the flat-convexity condition

$$F^0((1 - \varepsilon)\nu + \varepsilon\nu', \mu) \leq (1 - \varepsilon)F^0(\nu, \mu) + \varepsilon F^0(\nu', \mu),$$

for any $\nu, \nu' \in \mathcal{P}(\mathbb{R}^d)$, $\mu \in \mathcal{P}(\mathcal{Y} \times \mathcal{Z})$ and any $\varepsilon \in [0, 1]$. Since $F^0(\cdot, \mu) \in \mathfrak{C}^1(\mathbb{R}^d)$, by (Hu et al., 2021, Lemma 4.1), $\nu \mapsto F(\nu, \mu)$ satisfies $D_{F(\cdot, \mu)}(\nu', \nu) \geq 0$. Again, by convexity of $|\cdot|^2$, it holds that $\mathrm{TV}^2$ is convex, that is,

$$\mathrm{TV}^2((1 - \varepsilon)\mu + \varepsilon\mu', \hat{\mu}) \leq (1 - \varepsilon)\mathrm{TV}^2(\mu, \hat{\mu}) + \varepsilon \mathrm{TV}^2(\mu', \hat{\mu}),$$

for any $\mu, \mu' \in \mathcal{P}(\mathcal{Y} \times \mathcal{Z})$ and any $\varepsilon \in [0, 1]$. Also, by linearity of $F^0$ in $\mu$, it follows that $F$ satisfies the flat concavity condition

$$F(\nu, (1 - \varepsilon)\mu + \varepsilon\mu') \geq (1 - \varepsilon)F(\nu, \mu) + \varepsilon F(\nu, \mu'),$$

for any $\mu', \mu \in \mathcal{P}(\mathcal{Y} \times \mathcal{Z})$, $\nu \in \mathcal{P}(\mathbb{R}^d)$ and any $\varepsilon \in [0, 1]$. Hence, by (Hu et al., 2021, Lemma 4.1), $\mu \mapsto F(\nu, \mu)$ satisfies $D_{F(\nu, \cdot)}(\mu', \mu) \leq 0$. Therefore, $F$ satisfies Assumption 1.5.

To verify Assumption 3.3, it is enough to show that for all $\nu', \nu \in \mathcal{P}(\mathbb{R}^d)$, $\mu \in \mathcal{P}(\mathcal{Y} \times \mathcal{Z})$ and all $x \in \mathbb{R}^d$,

$$\left| \frac{\delta F}{\delta \nu}(\nu', \mu, x) - \frac{\delta F}{\delta \nu}(\nu, \mu, x) \right| \leq C_F \operatorname{TV}(\nu', \nu)$$

for some $C_F > 0$, since by Definition G.1, this implies

$$F(\nu', \mu) - F(\nu, \mu) - \int_{\mathbb{R}^d} \frac{\delta F}{\delta \nu}(\nu, \mu, x)(\nu' - \nu)(\mathrm{d}x)$$

$$= \int_0^1 \int_{\mathbb{R}^d} \left( \frac{\delta F}{\delta \nu}(\nu + \varepsilon(\nu' - \nu), \mu, x) - \frac{\delta F}{\delta \nu}(\nu, \mu, x) \right) (\nu' - \nu)(\mathrm{d}x)\mathrm{d}\varepsilon$$

$$\leq 2C_F \int_0^1 \operatorname{TV}(\nu + \varepsilon(\nu' - \nu), \nu) \operatorname{TV}(\nu', \nu)\mathrm{d}\varepsilon$$

$$\leq 2C_F \int_0^1 \varepsilon \operatorname{TV}^2(\nu', \nu)\mathrm{d}\varepsilon$$

$$= C_F \operatorname{TV}^2(\nu', \nu) \leq \frac{2C_F}{\alpha} D_h(\nu', \nu),$$

where the last inequality follows from Assumption 1.1. Thus, $D_{F(\cdot, \mu)}(\nu', \nu) \leq L_\nu D_h(\nu', \nu)$ in Assumption 3.3 holds with $L_\nu = \frac{2C_F}{\alpha}$. The same argument applies to $D_{F(\nu, \cdot)}(\mu', \mu) \geq -L_\mu D_h(\mu', \mu)$ in Assumption 3.3.

Note that

$$\frac{\delta F}{\delta \nu}(\nu, \mu, x) = - \int_{\mathcal{Y} \times \mathcal{Z}} \left( y - \mathbb{E}^{X \sim \nu}[\hat{\varphi}(X, z)] \right) \hat{\varphi}(x, z)\mu(\mathrm{d}y, \mathrm{d}z).$$

Since $\varphi$ is bounded by $M_\varphi > 0$, we obtain

$$\left| \frac{\delta F}{\delta \nu}(\nu', \mu, x) - \frac{\delta F}{\delta \nu}(\nu, \mu, x) \right| \leq \int_{\mathcal{Y} \times \mathcal{Z}} \int_{\mathbb{R}^d} |\hat{\varphi}(x, z)| \, |\nu' - \nu| \, (\mathrm{d}x) \, |\hat{\varphi}(x, z)| \, \mu(\mathrm{d}y, \mathrm{d}z)$$

$$\leq 2K^2 M_\varphi^2 \operatorname{TV}(\nu', \nu).$$

Let $r := (y, z) \in \mathbb{R}^d$, and assume for simplicity that both $\mu, \hat{\mu}$ are absolutely continuous with respect to Lebesgue measure. We claim that

$$\frac{\delta \operatorname{TV}(\cdot, \hat{\mu})}{\delta \mu}(\mu, r) = \frac{1}{2} \operatorname{sign}(\mu(r) - \hat{\mu}(r)),$$

for $\mu \neq \hat{\mu}$ a.e. Fix $\hat{\mu}$. For any $\mu'$, any $\mu \neq \hat{\mu}$ a.e., and any $\varepsilon \in (0, 1)$, (Tsybakov, 2008, Lemma 2.1) gives

$$\lim_{\varepsilon \to 0} \frac{1}{\varepsilon} \left( \operatorname{TV}(\mu + \varepsilon(\mu' - \mu), \hat{\mu}) - \operatorname{TV}(\mu, \hat{\mu}) \right)$$

$$= \lim_{\varepsilon \to 0} \frac{1}{2\varepsilon} \int_{\mathbb{R}^d} \left( |\mu(r) - \hat{\mu}(r) + \varepsilon(\mu'(r) - \mu(r))| - |\mu(r) - \hat{\mu}(r)| \right) \mathrm{d}r.$$

Since $|\cdot|$ is differentiable at every $v \neq 0$ with derivative $\operatorname{sign}(v)$, we obtain by dominated convergence

$$\lim_{\varepsilon \to 0} \frac{1}{\varepsilon} \left( \operatorname{TV}(\mu + \varepsilon(\mu' - \mu), \hat{\mu}) - \operatorname{TV}(\mu, \hat{\mu}) \right) = \frac{1}{2} \int_{\mathbb{R}^d} \operatorname{sign}(\mu(r) - \hat{\mu}(r)) (\mu'(r) - \mu(r))(\mathrm{d}r).$$

To justify dominated convergence, note that for every $r$, the reverse triangle inequality gives

$$\left| \frac{|\mu(r) - \hat{\mu}(r) + \varepsilon(\mu'(r) - \mu(r))| - |\mu(r) - \hat{\mu}(r)|}{\varepsilon} \right| \leq |\mu'(r) - \mu(r)| \in L^1(\mathbb{R}^d).$$

If $\mu = \hat{\mu}$ a.e., then the map $\mathbb{R} \ni v \mapsto |v|$ is not differentiable at $v = 0$ but its subdifferential is the interval $[-1, 1]$. Hence, the subdifferential of TV at such measures is the interval $[-\frac{1}{2}, \frac{1}{2}]$.

Finally, by the chain rule,

$$\frac{\delta \operatorname{TV}^2(\cdot, \hat{\mu})}{\delta \mu}(\mu, r) = 2 \operatorname{TV}(\mu, \hat{\mu}) \frac{\delta \operatorname{TV}(\cdot, \hat{\mu})}{\delta \mu}(\mu, r),$$

and we immediately see that $\frac{\delta\,\mathrm{TV}^2(\cdot,\hat{\mu})}{\delta\mu}(\mu, r) = 0$ if $\mu = \hat{\mu}$ a.e.. Hence, combining both cases,

$$\frac{\delta\,\mathrm{TV}^2(\cdot,\hat{\mu})}{\delta\mu}(\mu, r) = \begin{cases} \mathrm{TV}(\mu, \hat{\mu})\,\mathrm{sign}\,(\mu(r) - \hat{\mu}(r))\,, & \mu \neq \hat{\mu} \text{ a.e.}, \\ 0, & \mu = \hat{\mu} \text{ a.e.} \end{cases}$$

Consequently,

$$\frac{\delta F}{\delta\mu}(\nu, \mu, r) = \frac{1}{2}\left|y - \mathbb{E}^{X\sim\nu}[\hat{\varphi}(X, z)]\right|^2 - \mathrm{TV}(\mu, \hat{\mu})\,\mathrm{sign}\,(\mu(r) - \hat{\mu}(r))\,.$$

Hence,

$$\left|\frac{\delta F}{\delta\mu}(\nu, \mu', ) - \frac{\delta F}{\delta\mu}(\nu, \mu, r)\right| = \left|\mathrm{TV}(\mu', \hat{\mu})\,\mathrm{sign}\,(\mu'(r) - \hat{\mu}(r)) - \mathrm{TV}(\mu, \hat{\mu})\,\mathrm{sign}\,(\mu(r) - \hat{\mu}(r))\right|\,.$$

$$(37)$$

If $\mathrm{sign}(\mu'(r) - \hat{\mu}(r)) = \mathrm{sign}(\mu(r) - \hat{\mu}(r)) > 0$ a.e. or both are $< 0$ a.e., then (37) becomes

$$\left|\frac{\delta F}{\delta\mu}(\nu, \mu', r) - \frac{\delta F}{\delta\mu}(\nu, \mu, r)\right| = \left|\mathrm{TV}(\mu', \hat{\mu}) - \mathrm{TV}(\mu, \hat{\mu})\right|$$

$$= \frac{1}{2}\left|\int_{\mathbb{R}^d}(\mu'(r) - \hat{\mu}(r))\mathrm{d}r - \int_{\mathbb{R}^d}(\mu(r) - \hat{\mu}(r))\mathrm{d}r\right|$$

$$\leq \mathrm{TV}(\mu', \mu)\,.$$

If $\mathrm{sign}(\mu'(r) - \hat{\mu}(r)) > 0$ a.e. and $\mathrm{sign}(\mu(r) - \hat{\mu}(r)) < 0$ a.e., or vice versa, then (37) becomes

$$\left|\frac{\delta F}{\delta\mu}(\nu, \mu', r) - \frac{\delta F}{\delta\mu}(\nu, \mu, r)\right| = \mathrm{TV}(\mu', \hat{\mu}) + \mathrm{TV}(\mu, \hat{\mu})$$

$$= \frac{1}{2}\int_{\mathbb{R}^d}|\mu'(r) - \hat{\mu}(r)|\,\mathrm{d}r + \frac{1}{2}\int_{\mathbb{R}^d}|\mu(r) - \hat{\mu}(r)|\,\mathrm{d}r$$

$$= \frac{1}{2}\int_{\mathbb{R}^d}(\mu'(r) - \hat{\mu}(r))\,\mathrm{d}r + \frac{1}{2}\int_{\mathbb{R}^d}(\hat{\mu}(r) - \mu(r))\,\mathrm{d}r$$

$$\leq \mathrm{TV}(\mu', \mu)\,.$$

To verify Assumptions 1.6, note that

$$\left|\frac{\delta F}{\delta\nu}(\nu, \mu, x)\right| \leq \int_{\mathcal{Y}\times\mathcal{Z}}\left|y - \mathbb{E}^{X\sim\nu}[\hat{\varphi}(X, z)]\right||\hat{\varphi}(x, z)|\,\mu(\mathrm{d}y, \mathrm{d}z)$$

$$\leq KM_\varphi\,(\mu_{\mathcal{Y}} + KM_\varphi) := C_1,$$

where

$$\mu_{\mathcal{Y}} := \int_{\mathcal{Y}\times\mathcal{Z}}|y|\mu(\mathrm{d}y, \mathrm{d}z) < \infty$$

since $\mathcal{Y} \times \mathcal{Z}$ is compact.

Similarly,

$$\left|\frac{\delta F}{\delta\mu}(\nu, \mu, r)\right| = \left|\frac{1}{2}\left|y - \mathbb{E}^{X\sim\nu}[\hat{\varphi}(X, z)]\right|^2 - \mathrm{TV}(\mu, \hat{\mu})\,\mathrm{sign}\,(\mu(r) - \hat{\mu}(r))\right|$$

$$\leq 1 + \frac{1}{2}\,(\mathrm{diam}(\mathcal{Y}) + KM_\varphi)^2 := C_2,$$

since $\mathcal{Y}$ is compact and $\mathrm{TV}(\mu, \hat{\mu}) \leq 1$.

For Assumption 3.4, observe that

$$|F(\nu, \mu)| \leq \frac{1}{2} + \frac{1}{2}\,(\mathrm{diam}(\mathcal{Y}) + KM_\varphi)^2 := M,$$

since $\mathcal{Y}$ is compact and $\mathrm{TV}(\mu, \hat{\mu}) \leq 1$. $\qquad\square$

# E  ZERO-SUM MARKOV GAMES

In this section, we illustrate how problem (1) naturally applies to zero-sum Markov games, an example of distributional min–max problems in multi-agent reinforcement learning (see, e.g., (Littman, 1994; Zhang et al., 2020; Kim et al., 2024; Cen et al., 2024) and references therein). In such games, two agents interact within a shared environment, with one player aiming to maximize long-term average reward while an adversarial opponent seeks to minimize it. This competitive structure leads directly to a min–max optimization problem over the agents' policies.

Consider an infinite-horizon discounted zero-sum Markov Game $\mathcal{G} = (S, A, B, P, r, \delta)$, where $S$ is a finite state space, $A$ and $B$ are the action spaces of agents 1 and 2, respectively, $P : S \times A \times B \to \mathcal{P}(S)$ is the state transition kernel, $r : S \times A \times B \to \mathbb{R}$ is the reward function of agent 1 (so agent 2 receives $-r$), and $\delta \in [0, 1)$ is the discount factor. Agent 1 aims to minimize the expected discounted reward, while agent 2 aims to maximize it.

At each time $t$, given the current state $s_t$, agent 1 selects an action $a_t$, according to a policy $\nu : S \to \mathcal{P}(A)$, so that $a_t \sim \nu(\cdot|s_t) \in \mathcal{P}(A)$, and agent 2 selects an action $b_t$ according to a policy $\mu : S \to \mathcal{P}(B)$, so that $b_t \sim \mu(\cdot|s_t) \in \mathcal{P}(B)$. The environment then transitions to $s_{t+1} \sim P(\cdot|s_t, a_t, b_t) \in \mathcal{P}(S)$. Under a pair of policies $(\nu, \mu)$, the value function of the game is defined by

$$V^{\nu,\mu}(s) := \mathbb{E}_s^{\nu,\mu} \left[ \sum_{t=0}^{\infty} \delta^t r(s_t, a_t, b_t) \right], \tag{38}$$

where $\mathbb{E}_s^{\nu,\mu}$ denotes the expectation over the state-action trajectory $(s_0, a_0, , b_0, s_1, a_1, b_1, ...)$ generated by policies $\nu, \mu$ and kernel $P \in \mathcal{P}(S|S, A, B)$ such that $s_0 := s$, $a_t \sim \nu(\cdot|s_t)$, $b_t \sim \mu(\cdot|s_t)$ and $s_{t+1} \sim P(\cdot|s_t, a_t, b_t)$, for all $t \geq 0$. Similarly, the Q-value function under $(\nu, \mu)$ is defined by

$$Q^{\nu,\mu}(s, a, b) = \mathbb{E}_{s,a,b}^{\nu,\mu} \left[ \sum_{t=0}^{\infty} \delta^t r(s_t, a_t, b_t) \right]. \tag{39}$$

The objective of the two agents is to find an MNE[3] of the min-max problem

$$\min_{\nu \in \mathcal{P}(A)} \max_{\mu \in \mathcal{P}(B)} V^{\nu,\mu}(s), \tag{40}$$

for every $s \in S$.

Before verifying that problem (40) satisfies Assumptions 1.5, 1.6, 3.3, and 3.4, we first derive an alternative representation of $V^{\nu,\mu}$ to (38). To do so, we introduce some standard notation from Markov decision process theory. Full details can be found in Subsection E.1.

From (39), it follows that for any $(\nu, \mu) \in \mathcal{P}(A|S) \times \mathcal{P}(B|S)$ and any $s \in S$, the Q-value function can be equivalently written as

$$Q^{\nu,\mu}(s, a, b) = r(s, a, b) + \delta \int_S V^{\nu,\mu}(s') P(\mathrm{d}s'|s, a, b). \tag{41}$$

Moreover, (39) implies that the value function satisfies

$$V^{\nu,\mu}(s) = \int_B \int_A Q^{\nu,\mu}(s, a) \nu(\mathrm{d}a|s) \mu(\mathrm{d}b|s). \tag{42}$$

For given policies $(\nu, \mu) \in \mathcal{P}(A|S) \times \mathcal{P}(B|S)$, the occupancy kernel $d^{\nu,\mu} \in \mathcal{P}(S|S)$ is defined by

$$d^{\nu,\mu}(\mathrm{d}s'|s) = (1 - \delta) \sum_{t=0}^{\infty} \delta^t P_{\nu,\mu}^t(\mathrm{d}s'|s), \tag{43}$$

where $P_{\nu,\mu}^0(\mathrm{d}s'|s) := \delta_s(\mathrm{d}s')$, for the Dirac measure $\delta_s$ at $s \in S$, $P_{\nu'\mu}^t$ is a product of kernels in the sense of (45), and the convergence of the series is understood in $b\mathcal{K}(S|S)$.

---

[3]By (Shapley, 1953; Patek, 1997), there exists an MNE $(\nu^*, \mu^*) \in \mathcal{P}(A)^{|S|} \times \mathcal{P}(B)^{|S|}$ for two-player zero-sum Markov Games.

Using (41) and (46) in (42) gives for all $s \in S$ that

$$V^{\nu,\mu}(s) = \int_B \int_A r(s,a,b)\nu(\mathrm{d}a|s)\mu(\mathrm{d}b|s) + \delta \int_S V^{\nu,\mu}(s')P_{\nu,\mu}(\mathrm{d}s'|s).$$

Applying this identity recursively and using (43) yields for all $s \in S$ that

$$V^{\nu,\mu}(s) = \frac{1}{1-\delta} \int_S \int_B \int_A r(s',a,b)\nu(\mathrm{d}a|s')\mu(\mathrm{d}b|s')d^{\nu,\mu}(\mathrm{d}s'|s).$$

**Proposition E.1** (Verification of assumptions for Example E). *Let $S$ be a finite state space and $A, B$ be Polish action spaces. Suppose $r \in C_b(S \times A \times B)$ and $\delta \in [0,1)$. Then, for any $s \in S$, Assumptions 1.5, 1.6, 3.3 and 3.4 are satisfied by the objective*

$$F(\nu,\mu) := V^{\nu,\mu}(s) = \frac{1}{1-\delta} \int_S \int_B \int_A r(s',a,b)\nu(\mathrm{d}a|s')\mu(\mathrm{d}b|s')d^{\nu,\mu}(\mathrm{d}s'|s)$$

*from Example E.*

*Proof.* Applying the policy gradient theorem for Polish action spaces (Kerimkulov et al., 2025a, Proposition A.1) to $\mathcal{P}(A) \ni \nu(\cdot|s) \mapsto F(\nu,\mu)$ and $\mathcal{P}(B) \ni \mu(\cdot|s) \mapsto F(\nu,\mu)$, respectively, gives

$$\frac{\delta F}{\delta \nu}(\nu,\mu,a,s) = \frac{1}{1-\delta} \int_S \int_B r(s',a,b)\mu(\mathrm{d}b|s')d^{\nu,\mu}(\mathrm{d}s'|s),$$

and

$$\frac{\delta F}{\delta \mu}(\nu,\mu,b,s) = \frac{1}{1-\delta} \int_S \int_A r(s',a,b)\nu(\mathrm{d}a|s')d^{\nu,\mu}(\mathrm{d}s'|s).$$

Therefore, Assumption 1.5 holds with equality, and Assumption 3.3 holds with equality with $L_\nu = L_\mu = 0$.

Since $r \in C_b(S \times A \times B)$, $(\nu,\mu) \in \mathcal{P}(A|S) \times \mathcal{P}(B|S)$ and $d^{\nu,\mu} \in \mathcal{P}(S|S)$, it follows that

$$|F(\nu,\mu)| \le \frac{\|r\|_\infty}{1-\delta}, \quad \left|\frac{\delta F}{\delta \nu}(\nu,\mu,a,s)\right| \le \frac{\|r\|_\infty}{1-\delta}, \quad \left|\frac{\delta F}{\delta \mu}(\nu,\mu,b,s)\right| \le \frac{\|r\|_\infty}{1-\delta},$$

for all $(\nu,\mu) \in \mathcal{P}(A|S) \times \mathcal{P}(B|S)$ and all $a \in A, b \in B, s \in S$. Hence, Assumption 1.6 and 3.4 hold with $M = C_1 = C_2 = \frac{\|r\|_\infty}{1-\delta}$. $\qquad\square$

### E.1 NOTATION FOR MARKOV DECISION PROCESSES AND MARKOV GAMES

Let $(E,d)$ denote a Polish space, i.e., a complete separable metric space. Let $B_b(E)$ denote the space of bounded measurable functions $f : E \to \mathbb{R}$ endowed with the supremum norm $|f|_{B_b(E)} = \sup_{x \in E} |f(x)|$. Let $\mathcal{M}(E)$ denote the Banach space of finite signed measures $m$ on $E$ endowed with the total variation norm $|m|_{\mathcal{M}(E)} = |m|_{\mathrm{TV}}(E)$, where $|m|_{\mathrm{TV}}$ is the total-variation norm. We denote by $b\mathcal{K}(E|E)$ the Banach space of bounded signed kernels $k : E \to \mathcal{M}(E)$ endowed with the norm $|k|_{b\mathcal{K}(E|E)} = \sup_{x \in E} |k(x)|_{\mathcal{M}(E)}$; that is, $k(U|\cdot) : E \to \mathbb{R}$ is measurable for all $U \in \mathcal{M}(E)$ and $k(\cdot|x) \in \mathcal{M}(E)$ for all $x \in E$. Every kernel $k \in b\mathcal{K}(E|E)$ induces a bounded linear operator $T_k \in \mathcal{L}(\mathcal{M}(E), \mathcal{M}(E))$ defined by

$$T_k \eta(\mathrm{d}y) = \eta k(\mathrm{d}y) = \int_E \eta(\mathrm{d}x)k(\mathrm{d}y|x).$$

Moreover, we have

$$|k|_{b\mathcal{K}(E|E)} = \sup_{x \in E} \sup_{\substack{h \in B_b(E) \\ |h|_{B_b(E)} \le 1}} \int_E h(y)k(\mathrm{d}y|x) = |T_k|_{\mathcal{L}(\mathcal{M}(E),\mathcal{M}(E))}, \tag{44}$$

where the latter is the operator norm. Thus, $b\mathcal{K}(E|E)$ is a Banach algebra with the product defined via composition of the corresponding linear operators. In particular, for given $k \in b\mathcal{K}(E|E)$,

$$T_k^t \mu(\mathrm{d}y) = \mu k^t(\mathrm{d}y) = \int_{E^t} \mu(\mathrm{d}x_0)k(\mathrm{d}x_1|x_0)\cdots k(\mathrm{d}x_{t-1}|x_{t-2})k(\mathrm{d}y|x_{t-1}). \tag{45}$$

We denote by $\left((S \times A \times B)^{\mathbb{N}}, \mathcal{F}\right)$ a sample space, where the elements of $(S \times A \times B)^{\mathbb{N}}$ are state-action triples $(s_t, a_t, b_t)_{t=0}^{\infty}$ with $(s_t, a_t, b_t) \in S \times A \times B$, for each $t \in \mathbb{N}$, and $\mathcal{F}$ is the associated $\sigma$-algebra. By (Bertsekas & Shreve, 1978, Proposition 7.28), for a given initial distribution $\gamma \in \mathcal{P}(S)$ and policies $\nu \in \mathcal{P}(A|S)$, $\mu \in \mathcal{P}(B|S)$, there exists a unique product probability measure $\mathbb{P}_{\gamma}^{\nu, \mu}$ on $\left((S \times A \times B)^{\mathbb{N}}, \mathcal{F}\right)$ such that for every $t \in \mathbb{N}$, we have

1. $\mathbb{P}_{\gamma}^{\nu, \mu}(s_0 \in \mathcal{S}) = \gamma(\mathcal{S})$,
2. $\mathbb{P}^{\nu, \mu}(a_t \in \mathcal{A}|(s_0, a_0, b_0, \dots, s_t)) = \nu(a_t|s_t)$,
3. $\mathbb{P}^{\nu, \mu}(b_t \in \mathcal{B}|(s_0, a_0, b_0, \dots, s_t)) = \mu(b_t|s_t)$,
4. $\mathbb{P}_{\gamma}^{\nu, \mu}(s_{t+1} \in \mathcal{S}|(s_0, a_0, b_0, \dots, s_t, a_t, b_t)) = P(\mathcal{S}|s_t, a_t, b_t)$,

for all $\mathcal{S} \in \mathcal{B}(S)$ and $\mathcal{A} \in \mathcal{B}(A)$. Thus, $\{s_t\}_{t \geq 0}$ is a Markov chain with transition kernel $P_{\nu, \mu} \in \mathcal{P}(S|S)$ defined by

$$P_{\nu, \mu}(\mathrm{d}s'|s) := \int_B \int_A P(\mathrm{d}s'|s, a', b')\nu(\mathrm{d}a'|s)\mu(\mathrm{d}b'|s). \tag{46}$$

The expectation corresponding to $\mathbb{P}_{\gamma}^{\nu, \mu}$ is denoted by $\mathbb{E}_{\gamma}^{\nu, \mu}$. For given $s \in S$, we denote $\mathbb{E}_s^{\nu, \mu} := \mathbb{E}_{\delta_s}^{\nu, \mu}$, where $\delta_s \in \mathcal{P}(S)$ denotes the Dirac measure at $s \in S$.

# F    NUMERICAL EXPERIMENTS

In this section, we outline how to implement the infinite-dimensional algorithms 1 and 2 in the case where $h$ is the relative entropy. For brevity, we present the derivations only for Algorithm 1, as the arguments for Algorithm 2 are entirely analogous. The complete algorithms for both the simultaneous and alternating MDA schemes can be found in Algorithm 3 and Algorithm 4 in Section F.3.

## F.1    SIMULATION OF INFINITE-DIMENSIONAL MDA

As shown in Example 1.3, by taking $h$ to be the entropy, the corresponding $h$-Bregman divergence is exactly the KL divergence. Moreover, using the flat derivative formula (5), the first-order optimality condition in Proposition B.3 gives

$$\begin{cases} \log \nu^{n+1}(x) - \log \nu^n(x) = -\tau \frac{\delta F}{\delta \nu}(\nu^n, \mu^n, x) + C, \\ \log \mu^{n+1}(y) - \log \mu^n(y) = \tau \frac{\delta F}{\delta \mu}(\nu^n, \mu^n, y) + C', \end{cases}$$

for every $n \geq 0$ and, for all $x, y \in \mathcal{X}$ Lebesgue a.e., where $C, C' \in \mathbb{R}$. By summing over $n$ and exponentiating both sides, we obtain

$$\begin{cases} \nu^n(x) \propto \nu^0(x)e^{-\tau \sum_{k=0}^{n-1} \frac{\delta F}{\delta \nu}(\nu^k, \mu^k, x)}, \\ \mu^n(y) \propto \mu^0(y)e^{\tau \sum_{k=0}^{n-1} \frac{\delta F}{\delta \mu}(\nu^k, \mu^k, y)}, \end{cases}$$

where the constants $C, C'$ are absorbed into the normalizations.

For simplicity, suppose the initial samples $(X_j, Y_j)_{j=1}^J$ are drawn uniformly, so that $(\nu^0, \mu^0)$ are uniform densities. We set $(X_{j,0}, Y_{j,0})_{j=1}^J = (X_j, Y_j)_{j=1}^J$ and sample from $(\nu^1, \mu^1)$ via Langevin dynamics:

$$X_{j,t+1} = X_{j,t} - \gamma\nabla\frac{\delta F}{\delta \nu}(\nu^0, \mu^0, X_{j,t}) + \sqrt{\frac{2\gamma}{\tau}}\mathcal{N}_{j,t},$$

$$Y_{j,t+1} = Y_{j,t} + \gamma\nabla\frac{\delta F}{\delta \mu}(\nu^0, \mu^0, Y_{j,t}) + \sqrt{\frac{2\gamma}{\tau}}\mathcal{N}_{j,t},$$

for $1 \leq j \leq J$ and $0 \leq t \leq T - 1$, where $\gamma > 0$ is the step size and $\mathcal{N}_{j,t}$ are i.i.d standard Gaussian variables. For sufficiently large $J$ and $T$, the terminal particles $(X_{j,T}, Y_{j,T})_{j=1}^J$ approximate samples from $(\nu^1, \mu^1)$. Repeating this procedure recursively then yields samples from $(\nu^2, \mu^2), \dots, (\nu^n, \mu^n)$.

## F.2    TRAINING GANS BY MDA

We train the mean-field GAN from Example 1.2 using simultaneous and alternating MDA-GAN (Algorithms (5) and (6)) on the 8-Gaussian mixture and Swiss Roll datasets (Gulrajani et al., 2017). Full algorithmic details, including hyperparameters and network architectures, are in Section F.3. Both methods are run for 2000 iterations, with performance assessed by visualizing generated samples at 400, 1000, and 2000 iterations.

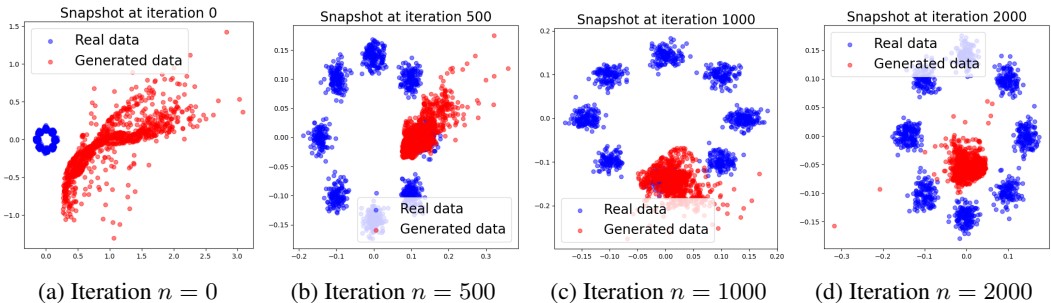

(a) Iteration $n = 0$        (b) Iteration $n = 500$        (c) Iteration $n = 1000$        (d) Iteration $n = 2000$

Figure 1: Simultaneous MDA-GAN (Algorithm 5) learning an 8-Gaussian mixture

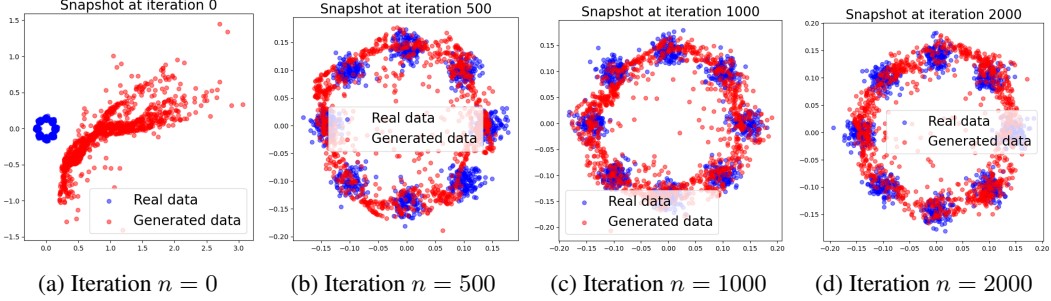

(a) Iteration $n = 0$        (b) Iteration $n = 500$        (c) Iteration $n = 1000$        (d) Iteration $n = 2000$

Figure 2: alternating MDA-GAN (Algorithm 6) learning an 8-Gaussian mixture

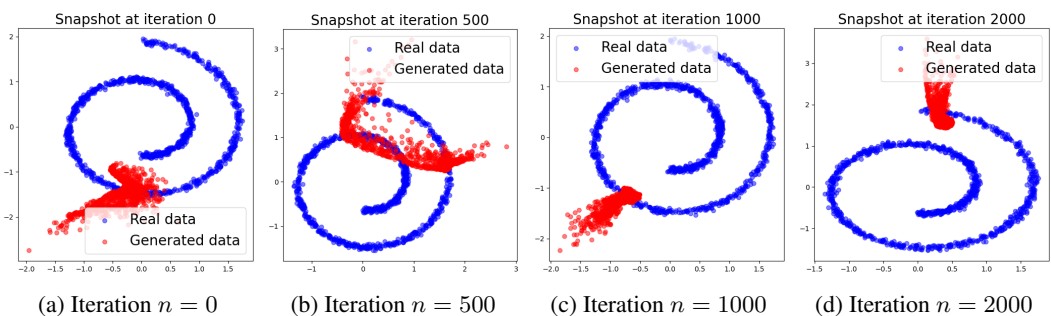

(a) Iteration $n = 0$        (b) Iteration $n = 500$        (c) Iteration $n = 1000$        (d) Iteration $n = 2000$

Figure 3: Simultaneous MDA-GAN (Algorithm 5) learning the Swiss Roll

Figures 1 and 2 show the training dynamics of simultaneous and alternating MDA-GANs on the 8-Gaussian mixture, with analogous results on the Swiss Roll in Figures 3 and 4. In both settings, generated samples start far from the data but the alternating variant captures the multi-modal structure and the spiral geometry of the Swiss Roll more clearly and at earlier iterations. In Figure 5, we plot the $L^1$-Wasserstein distance $W_1\left(T_{\theta^n}\#\xi, \hat{\xi}\right)$ for both tasks over iterations $n$, confirming the faster convergence of alternating MDA-GAN.

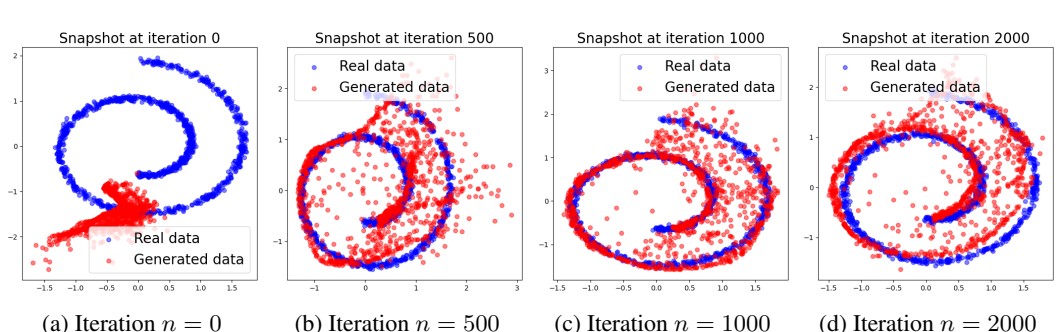

(a) Iteration $n = 0$     (b) Iteration $n = 500$     (c) Iteration $n = 1000$     (d) Iteration $n = 2000$

Figure 4: alternating MDA-GAN (Algorithm 6) learning the Swiss Roll

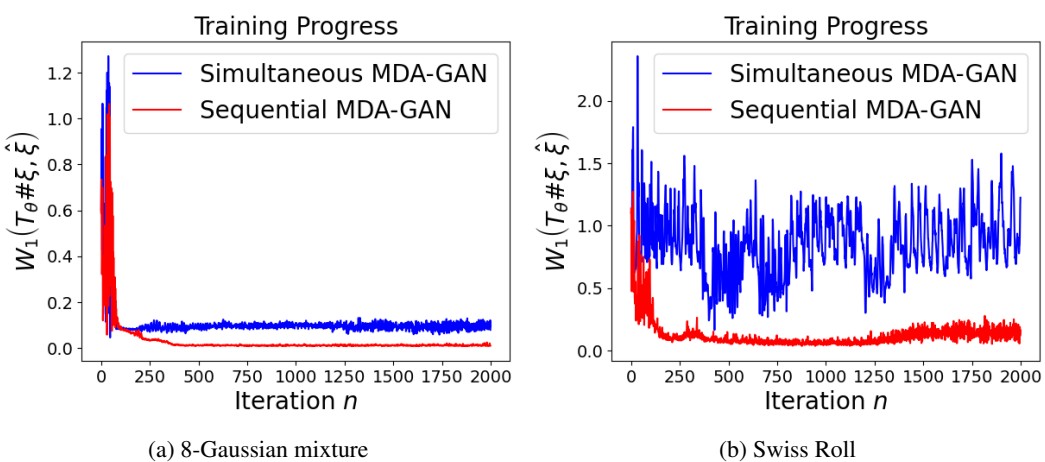

(a) 8-Gaussian mixture     (b) Swiss Roll

Figure 5: $L^1$-Wasserstein distance between generated and real data for the 8-Gaussian mixture and Swiss Roll

## F.3 Details on numerical experiments

In this section, we present the additional details of the numerical experiments. We begin by summarizing the implementable versions of the simultaneous and alternating MDA algorithms introduced in Section F. We now turn to Algorithms 3 and 4 in the setting where $F$ corresponds to the GAN

---

**Algorithm 3:** Implementable simultaneous MDA

**Input:** objective function $F$, initial measures $(\nu^0, \mu^0)$, stepsize $\tau, \gamma > 0$, time horizons $K, N$
      and number of particles $J$

Generate i.i.d $\left(X_j^0, Y_j^0\right)_{j=1}^J \sim (\nu^0, \mu^0)$

Set $\left(X_{j,0}^0, Y_{j,0}^0\right)_{j=1}^J = \left(X_j^0, Y_j^0\right)_{j=1}^J$

**for** $n = 0, 1, \ldots, N-1$ **do**
    **for** $k = 0, 1, \ldots, K-1$ **do**
        Generate independent Gaussian random variables $\mathcal{N}_{j,k}^n$
        **for** $j = 1, 2, \ldots, J$ **do**
            $X_{j,k+1}^n = X_{j,k}^n - \gamma \nabla \frac{\delta F}{\delta \nu}(\nu^n, \mu^n, X_{j,k}^n) + \sqrt{\frac{2\gamma}{\tau}} \mathcal{N}_{j,k}^n$
            $Y_{j,k+1}^n = Y_{j,k}^n + \gamma \nabla \frac{\delta F}{\delta \mu}(\nu^n, \mu^n, Y_{j,k}^n) + \sqrt{\frac{2\gamma}{\tau}} \mathcal{N}_{j,k}^n$
    **for** $j = 1, 2, \ldots, J$ **do**
        $X_{j,0}^{n+1} = X_{j,K}^n, \quad Y_{j,0}^{n+1} = Y_{j,K}^n$
    $\nu^n = \frac{1}{J} \sum_{j=1}^J \delta_{X_{j,0}^n}, \quad \mu^n = \frac{1}{J} \sum_{j=1}^J \delta_{Y_{j,0}^n}$

**Output:** $\left(\frac{1}{N} \sum_{n=0}^{N-1} \nu^n, \frac{1}{N} \sum_{n=0}^{N-1} \mu^n\right)$

---

objective introduced in Example 1.2. Recall that $F$ takes the form

$$F(\nu, \mu) = \int_{\mathcal{W}} \int_{\Theta} \int_{\mathcal{Y}} D_w(y) \left(T_\theta \# \xi - \hat{\xi}\right)(\mathrm{d}y) \nu(\mathrm{d}\theta) \mu(\mathrm{d}w)$$

$$= \int_{\mathcal{W}} \int_{\Theta} \int_{\mathcal{Y}} D_w(y) \left(T_\theta \# \xi\right)(\mathrm{d}y) \nu(\mathrm{d}\theta) \mu(\mathrm{d}w) - \int_{\mathcal{W}} \int_{\mathcal{Y}} D_w(y) \hat{\xi}(\mathrm{d}y) \mu(\mathrm{d}w)$$

By Definition G.1, we have

$$\frac{\delta F}{\delta \nu}(\nu, \mu, \theta) = \int_{\mathcal{W}} \int_{\mathcal{Y}} D_w(y) \left(T_\theta \# \xi\right)(\mathrm{d}y) \mu(\mathrm{d}w),$$

$$\frac{\delta F}{\delta \mu}(\nu, \mu, w) = \int_{\Theta} \int_{\mathcal{Y}} D_w(y) \left(T_\theta \# \xi\right)(\mathrm{d}y) \nu(\mathrm{d}\theta) - \int_{\mathcal{Y}} D_w(y) \hat{\xi}(\mathrm{d}y).$$

The flat derivatives can be approximated using empirical averages. For a batch of real data $\{\xi_1^{\text{real}}, \ldots, \xi_M^{\text{real}}\} \sim \hat{\xi}$, we have

$$\int_{\mathcal{Y}} D_w(y) \hat{\xi}(\mathrm{d}y) \approx \frac{1}{M} \sum_{i=1}^M D_w(\xi_i^{\text{real}}).$$

For the term in $\frac{\delta F}{\delta \mu}(\nu, \mu, w)$ that involves integration with respect to both $\nu$ and the generated data $T_\theta \# \xi$, we approximate via sampling as follows. We sample

$$\{\theta^{(1)}, \theta^{(2)}, ..., \theta^{(J)}\} \sim \nu, \quad \left\{Z_i^{(j)}\right\}_{i=1}^M \sim T_{\theta^{(j)}} \# \xi,$$

leading to the estimator

$$\int_{\Theta} \int_{\mathcal{Y}} D_w(y) \left(T_\theta \# \xi\right)(\mathrm{d}y) \nu(\mathrm{d}\theta) \approx \frac{1}{JM} \sum_{i=1}^M \sum_{j=1}^J D_w\left(X_i^{(j)}\right).$$

---

**Algorithm 4:** IMPLEMENTABLE ALTERNATING MDA

---

**Input:** objective function $F$, initial measures $(\nu^0, \mu^0)$, stepsize $\tau, \gamma > 0$, time horizons $K, N$ and number of particles $J$

Generate i.i.d $\left(X_j^0, Y_j^0\right)_{j=1}^J \sim (\nu^0, \mu^0)$

Set $\left(X_{j,0}^0, Y_{j,0}^0\right)_{j=1}^J = \left(X_j^0, Y_j^0\right)_{j=1}^J$

**for** $n = 0, 1, \ldots, N-1$ **do**

    **for** $k = 0, 1, \ldots, K-1$ **do**

        Generate independent Gaussian random variables $\mathcal{N}_{j,t}^n$

        **for** $j = 1, 2, \ldots, J$ **do**

            $X_{j,k+1}^n = X_{j,k}^n - \gamma \nabla \frac{\delta F}{\delta \nu}(\nu^n, \mu^n, X_{j,k}^n) + \sqrt{\frac{2\gamma}{\tau}} \mathcal{N}_{j,k}^n$

    **for** $j = 1, 2, \ldots, J$ **do**

        $X_{j,0}^{n+1} = X_{j,K}^n$

    $\nu^{n+1} = \frac{1}{J} \sum_{j=1}^J \delta_{X_{j,0}^{n+1}}$

    **for** $k = 0, 1, \ldots, K-1$ **do**

        Generate independent Gaussian random variables $\mathcal{N}_{j,k}^n$

        **for** $j = 1, 2, \ldots, J$ **do**

            $Y_{j,k+1}^n = Y_{j,k}^n + \gamma \nabla \frac{\delta F}{\delta \mu}(\nu^{n+1}, \mu^n, Y_{j,k}^n) + \sqrt{\frac{2\gamma}{\tau}} \mathcal{N}_{j,k}^n$

    **for** $j = 1, 2, \ldots, J$ **do**

        $Y_{j,0}^{n+1} = Y_{j,K}^n$

    $\mu^n = \frac{1}{J} \sum_{j=1}^J \delta_{Y_{j,0}^n}$

**Output:** $\left(\frac{1}{N} \sum_{n=0}^{N-1} \nu^{n+1}, \frac{1}{N} \sum_{n=0}^{N-1} \mu^n\right)$

---

Analogously, for $\frac{\delta F}{\delta \nu}(\nu, \mu, \theta)$ we sample

$$\{w^{(1)}, w^{(2)}, ..., w^{(J)}\} \sim \mu, \quad \{Z_i\}_{i=1}^M \sim T_\theta \# \xi,$$

and approximate

$$\int_{\mathcal{W}} \int_{\mathcal{Y}} D_w(y)\,(T_\theta \# \xi)\,(\mathrm{d}y)\mu(\mathrm{d}w) \approx \frac{1}{JM} \sum_{i=1}^M \sum_{j=1}^J D_{w^{(j)}}(Z_i).$$

To mitigate the computational cost of Algorithms 3 and 4, we follow the approach of (Hsieh et al., 2019) and employ Langevin dynamics with exponential damping (see also their Algorithm 3). Below, we present this algorithm in both the simultaneous and alternating variants used in our experiments.

In all experiments, we closely follow the specifications from (Hsieh et al., 2019). We adopt the gradient-penalized discriminator of (Gulrajani et al., 2017) as a soft-constraint alternative to the original Wasserstein GAN formulation to increase stability. The gradient penalty parameter is set to $\lambda = 0.1$. For our Simultaneous and alternating MDA-GANs, we fix the damping factor to $\beta = 0.8$. The scheduling of the parameters $K^n, \gamma^n$, and $\tau^n$ is $K^n = \lfloor (1 + 10^{-5})^n \rfloor$, $\gamma^n = \gamma(1 - 10^{-5})^n$, with $\gamma = 0.01$, and $\tau^n = \tau(1 - 5 \times 10^{-5})^{-t}$, with $\tau = 100$. The number of samples per batch is $M = 1024$. For both the 8-Gaussian mixture and Swiss Roll datasets, we use fully connected networks for the generator and discriminator, each consisting of two-hidden-layers with $J = 512$ neurons on each layer. The generator and discriminator networks use ReLU activations, except for the output layer of the discriminator, which employs a tanh activation. All network parameters are initialized from a normal distribution $\mathcal{N}(0, 0.01)$.

---

**Algorithm 5:** SIMULTANEOUS MDA-GAN

**Input:** Initial parameters $w^0, \theta^0$, step sizes $\{\gamma^n\}_{n=0}^{N-1}$, $\{\tau^n\}_{n=0}^{N-1}$, time horizon $\{K^n\}_{n=0}^{N-1}$, averaging parameter $\beta \in [0,1]$, source probability measure $\xi$

**for** $n = 0, 1, \ldots, N-1$ **do**

    Set $\bar{w}^n, w_0^n = w^n$ and $\bar{\theta}^n, \theta_0^n = \theta^n$;

    **for** $k = 0, 1, \ldots, K_t - 1$ **do**

        $A = \{Z_1, \ldots, Z_M\} \sim T_{\theta_k^n} \# \xi$;

$$\theta_{k+1}^n = \theta_k^n - \frac{\gamma^n}{M} \nabla_\theta \sum_{Z_i \in A} D_{w^n}(Z_i) + \sqrt{\frac{2\gamma^n}{\tau^n}} \mathcal{N}_k^n;$$

        $B = \{\xi_1^{\text{real}}, \ldots, \xi_M^{\text{real}}\} \sim \hat{\xi}$;
        $B' = \{Z_1', \ldots, Z_M'\} \sim T_{\theta^n} \# \xi$;

$$w_{k+1}^n = w_k^n + \frac{\gamma^n}{M} \nabla_w \sum_{Z_i' \in B'} D_{w_k^n}(Z_i') - \frac{\gamma^n}{M} \nabla_w \sum_{\xi_i^{\text{real}} \in B} D_{w_k^n}(\xi_i^{\text{real}}) + \sqrt{\frac{2\gamma^n}{\tau^n}} \mathcal{N}_k^n;$$

        $\bar{w}^n = (1-\beta)\bar{w}^n + \beta w_{k+1}^n, \quad \bar{\theta}^n = (1-\beta)\bar{\theta}^n + \beta \theta_{k+1}^n$;

    $w^{n+1} = (1-\beta)w^t + \beta \bar{w}^n, \quad \theta^{n+1} = (1-\beta)\theta^n + \beta \bar{\bar{\theta}}^n$;

**Output:** $w^N, \theta^N$

---

**Algorithm 6:** ALTERNATING MDA-GAN

**Input:** Initial parameters $w^0, \theta^0$, step sizes $\{\gamma^n\}_{n=0}^{N-1}$, $\{\tau^n\}_{n=0}^{N-1}$, time horizon $\{K^n\}_{n=0}^{N-1}$, averaging parameter $\beta \in [0,1]$, source probability measure $\xi$

**for** $n = 0, 1, \ldots, N-1$ **do**

    Set $\bar{w}^n, w_0^n = w^n$ and $\bar{\theta}^n, \theta_0^n = \theta^n$;

    **for** $k = 0, 1, \ldots, K_t - 1$ **do**

        $A = \{Z_1, \ldots, Z_M\} \sim T_{\theta_k^n} \# \xi$;

$$\theta_{k+1}^n = \theta_k^n - \frac{\gamma^n}{M} \nabla_\theta \sum_{Z_i \in A} D_{w^n}(Z_i) + \sqrt{\frac{2\gamma^n}{\tau^n}} \mathcal{N}_k^n;$$

        $\bar{\theta}^n = (1-\beta)\bar{\theta}^n + \beta \theta_{k+1}^n$;

    $\theta^{n+1} = (1-\beta)\theta^n + \beta \bar{\theta}^n$;

    **for** $k = 0, 1, \ldots, K_t - 1$ **do**

        $B = \{\xi_1^{\text{real}}, \ldots, \xi_M^{\text{real}}\} \sim \hat{\xi}$;
        $B' = \{Z_1', \ldots, Z_M'\} \sim T_{\theta^{n+1}} \# \xi$;

$$w_{k+1}^n = w_k^n + \frac{\gamma^n}{M} \nabla_w \sum_{Z_i' \in B'} D_{w_k^n}(Z_i') - \frac{\gamma^n}{M} \nabla_w \sum_{\xi_i^{\text{real}} \in B} D_{w_k^n}(\xi_i^{\text{real}}) + \sqrt{\frac{2\gamma^n}{\tau^n}} \mathcal{N}_k^n;$$

        $\bar{w}^n = (1-\beta)\bar{w}^n + \beta w_{k+1}^n$;

    $w^{n+1} = (1-\beta)w^t + \beta \bar{w}^n$;

**Output:** $w^N, \theta^N$

---

# G  DIFFERENTIABILITY ON THE PRIMAL SPACE

In this section, following (Carmona & Delarue, 2018, Definition 5.43) and (Santambrogio, 2015, Definition 7.12), we introduce the notion of differentiability on the space of measures that we utilize throughout the paper.

**Definition G.1.** *For any $\mathcal{X} \subset \mathbb{R}^d$, let $\mathcal{K} \subseteq \mathcal{P}(\mathcal{X})$ be convex and let $F : \mathcal{P}(\mathcal{X}) \to \mathbb{R}$. We say $F \in \mathfrak{C}^1(\mathcal{K})$, if there exists a measurable function $\frac{\delta F}{\delta \nu} : \mathcal{K} \times \mathcal{X} \to \mathbb{R}$ such that, for any $\nu, \nu' \in \mathcal{K}$, there exists $C > 0$ such that, for all $x \in \mathcal{X}$, we have $\left| \frac{\delta F}{\delta \nu}(\nu, x) \right| \leq C$, and it holds that*

$$\lim_{\varepsilon \to 0} \frac{F(\nu + \varepsilon(\nu' - \nu)) - F(\nu)}{\varepsilon} = \int_{\mathcal{X}} \frac{\delta F}{\delta \nu}(\nu, x)(\nu' - \nu)(\mathrm{d}x). \tag{47}$$

*The functional $\frac{\delta F}{\delta \nu}$ is called the flat derivative of $F$ on $\mathcal{K}$. We note that $\frac{\delta F}{\delta \nu}$ exists up to an additive constant, and thus we make the normalizing convention $\int_{\mathcal{X}} \frac{\delta F}{\delta \nu}(\nu, x)\nu(\mathrm{d}x) = 0$.*

If, for any fixed $x \in \mathcal{X}$, the map $\nu \mapsto \frac{\delta F}{\delta \nu}(\nu, x)$ satisfies Definition G.1, we say $F \in \mathfrak{C}^2(\mathcal{K})$, i.e., it admits a second-order flat derivative denoted by $\frac{\delta^2 F}{\delta \nu^2}$. Consequently, by Definition G.1, there exists a measurable functional $\frac{\delta^2 F}{\delta \nu^2} : \mathcal{K} \times \mathcal{X} \times \mathcal{X} \to \mathbb{R}$ such that

$$\lim_{\varepsilon \to 0} \frac{1}{\varepsilon} \left( \frac{\delta F}{\delta \nu}(\nu + \varepsilon(\nu' - \nu), x) - \frac{\delta F}{\delta \nu}(\nu, x) \right) = \int_{\mathcal{X}} \frac{\delta^2 F}{\delta \nu^2}(\nu, x, x')(\nu' - \nu)(\mathrm{d}x'). \tag{48}$$

**Remark G.2.** *One can show that if $F : \mathcal{P}(\mathbb{R}^d) \to \mathbb{R}^d$ admits a flat derivative $\frac{\delta F}{\delta \mu}$, then for all $\mu, \mu' \in \mathcal{P}(\mathbb{R}^d)$, the function $[0,1] \ni \varepsilon \mapsto F(\mu^\varepsilon)$ is continuous on $[0,1]$ and differentiable on $(0,1)$ with derivative $\frac{\mathrm{d}}{\mathrm{d}\varepsilon} F(\mu^\varepsilon) = \int_{\mathbb{R}^d} \frac{\delta F}{\delta \mu}(\mu^\varepsilon, x)(\mu' - \mu)(\mathrm{d}x)$ (see (Jourdain & Tse, 2020, Theorem 2.3)). Hence, by the fundamental theorem of calculus, $F(\mu') - F(\mu) = \int_0^1 \int_{\mathbb{R}^d} \frac{\delta F}{\delta \mu}(\mu^\varepsilon, x)(\mu' - \mu)(\mathrm{d}x)\mathrm{d}\varepsilon$, provided that $\varepsilon \mapsto \int \frac{\delta F}{\delta \mu}(\mu^\varepsilon, x)(\mu' - \mu)(\mathrm{d}x)$ is integrable.*

## H   DIFFERENTIABILITY ON THE DUAL SPACE

In this section, we start by recalling the notions of Fréchet and Gâteaux derivative for functions $H : C_b(\mathcal{X}) \to \mathbb{R}$, where $(C_b(\mathcal{X}), \|\cdot\|_\infty)$ is the Banach space of real-valued bounded continuous functions on $\mathcal{X} \subset \mathbb{R}^d$ ; see e.g. Chapters 7, 1, 3 in (Aliprantis & Border, 2007; Ambrosetti & Prodi, 1995; Ortega & Rheinboldt, 1970), respectively. Based on these notions of differentiablity, we will introduce the notions of first and second variation for functions $H$.

### H.1   PRELIMINARIES ON FRÉCHET AND GÂTEAUX DERIVATIVES

For $\mathcal{X} \subset \mathbb{R}^d$, let $\mathcal{L}(C_b(\mathcal{X}), \mathbb{R})$ and $\mathcal{L}(C_b(\mathcal{X}))$ denote the space of continuous linear maps from $C_b(\mathcal{X})$ to $\mathbb{R}$, and from $C_b(\mathcal{X})$ to itself, respectively.

**Definition H.1** (Fréchet differentiability)**.** *Let $\mathcal{U} \subset C_b(\mathcal{X})$ be open. Given $f \in \mathcal{U}$, the function $H : \mathcal{U} \to \mathbb{R}$ is Fréchet differentiable at $f$ if there exists $T \in \mathcal{L}(C_b(\mathcal{X}), \mathbb{R})$ such that, for all $g \in C_b(\mathcal{X})$,*

$$\lim_{\|g\|_\infty \to 0} \frac{|H(f + g) - H(f) - T[g]|}{\|g\|_\infty} = 0.$$

*If it exists, the map $T$ is unique, we write $T = \nabla_{\mathcal{F}} H(f)$, and call $\nabla_{\mathcal{F}} H(f)$ the Fréchet derivative of $H$ at $f$. If $H$ is Fréchet differentiable at every $f \in \mathcal{U}$, then we say that $H$ is Fréchet differentiable on $\mathcal{U}$.*

**Example H.2** (Convex conjugate of the relative entropy)**.** *If $h$ is the relative entropy in Example 1.3, then a straightforward calculation directly from Definition 3.1 shows that its dual $h^*$ is given by*

$$h^*(f) = \log \left( \int_{\mathcal{X}} e^{f(z)} \pi(\mathrm{d}z) \right).$$

**Example H.3** (Fréchet derivative of the relative entropy)**.** *A straightforward calculation directly from Definition H.1 shows that $h^*$ is Fréchet differentiable on $C_b(\mathcal{X})$ with Fréchet derivative given by*

$$\nabla_{\mathcal{F}} h^*(f)[g] = \int_{\mathcal{X}} g(z) \frac{e^{f(z)}}{\int_{\mathcal{X}} e^{f(y)} \pi(\mathrm{d}y)} \pi(\mathrm{d}z),$$

*for all $g \in C_b(\mathcal{X})$.*

**Example H.4** (Convex conjugate of the $\chi^2$-divergence). *If $h$ is the $\chi^2$-divergence in Example 1.4, then (Polyanskiy & Wu, 2025, Example 7.4) shows that its dual $h^*$ is given by*

$$h^*(f) = \frac{1}{2} \int_{\mathcal{X}} f(z)\pi(\mathrm{d}z) + \frac{1}{8} \int_{\mathcal{X}} f^2(z)\pi(\mathrm{d}z).$$

**Example H.5** (Fréchet derivative of the relative $\chi^2$-divergence). *A straightforward calculation directly from Definition H.1 shows that $h^*$ is Fréchet differentiable on $C_b(\mathcal{X})$ with Fréchet derivative given by*

$$\nabla_{\mathcal{F}} h^*(f)[g] = \frac{1}{2} \int_{\mathcal{X}} g(z)\pi(\mathrm{d}z) + \frac{1}{4} \int_{\mathcal{X}} g(z)f(z)\pi(\mathrm{d}z),$$

*for all $g \in C_b(\mathcal{X})$.*

**Definition H.6** (Gâteaux differentiability). *Let $\mathcal{U} \subset C_b(\mathcal{X})$ be open. Given $f \in \mathcal{U}$, the function $H : \mathcal{U} \to \mathbb{R}$ is Gâteaux differentiable at $f$ if there exists $T \in \mathcal{L}(C_b(\mathcal{X}), \mathbb{R})$ such that for any direction $f' \in C_b(\mathcal{X})$,*

$$\lim_{\varepsilon \downarrow 0} \frac{H(f + \varepsilon f') - H(f)}{\varepsilon} = T[f'].$$

*If it exists, the map $T$ is unique, we write $T = \nabla_{\mathcal{G}} H(f)$, and call $\nabla_{\mathcal{G}} H(f)$ the Gâteaux derivative of $H$ at $f$. If $H$ is Gâteaux differentiable at every $f \in \mathcal{U}$, then we say that $H$ is Gâteaux differentiable on $\mathcal{U}$.*

As observed in Chapter $1, 3$ in (Ambrosetti & Prodi, 1995; Ortega & Rheinboldt, 1970), if $H$ is Fréchet differentiable, then it is automatically Gâteaux differentiable and the two derivatives coincide, i.e., $\nabla_{\mathcal{F}} H = \nabla_{\mathcal{G}} H$. Moreover, (Ortega & Rheinboldt, 1970, Proposition 3.1.6) proves that Fréchet differentiability of $H$ at $f \in \mathcal{U}$ implies that $H$ is continuous at $f$, whereas in the case of Gâteaux differentiability, this does not necessarily hold; see (Ortega & Rheinboldt, 1970, Proposition 3.1.4).

Following the discussions in (Aliprantis & Border, 2007; Ambrosetti & Prodi, 1995; Ortega & Rheinboldt, 1970), it is possible to extend Definition H.1 to higher-order Fréchet derivatives.

**Definition H.7** (Second-order Fréchet differentiability). *Let $\mathcal{U} \subset C_b(\mathcal{X})$ be open and let $f \in \mathcal{U}$. Suppose that $H : \mathcal{U} \to \mathbb{R}$ is Fréchet differentiable (cf. Definition H.1) at $f$, and admits Fréchet derivative $\nabla_{\mathcal{F}} H(f)$. Then $\nabla_{\mathcal{F}} H(f)$ is Fréchet differentiable at $f$, if there exists $T \in \mathcal{L}(C_b(\mathcal{X}), \mathcal{L}(C_b(\mathcal{X}), \mathbb{R}))$ such that for all $f', f'' \in C_b(\mathcal{X})$,*

$$\lim_{\|f''\|_{\infty} \to 0} \frac{|\nabla_{\mathcal{F}} H(f + f'')[f'] - \nabla_{\mathcal{F}} H(f)[f'] - T[f''][f']|}{\|f''\|_{\infty}} = 0.$$

*If it exists, the map $T$ is unique, we write $T = \nabla_{\mathcal{F}}^2 H(f)$, and call $\nabla_{\mathcal{F}}^2 H(f)$ the second Fréchet derivative of $H$ at $f$.*

**Example H.8** (Second-order Fréchet derivative of the relative entropy). *If $h$ is the relative entropy in Example 1.3, using Example H.3, we can show that $\nabla_{\mathcal{F}} h^*(f)$ is Fréchet differentiable on $C_b(\mathcal{X})$ with Fréchet derivative given by*

$$\nabla_{\mathcal{F}}^2 h^*(f)[g'][g] = \int_{\mathcal{X}} g(x)g'(x)\varphi(f)(\mathrm{d}x) - \left(\int_{\mathcal{X}} g'(z)\varphi(f)(\mathrm{d}z)\right)\left(\int_{\mathcal{X}} g(z)\varphi(f)(\mathrm{d}z)\right)$$
$$= \mathrm{Cov}_{\varphi(f)}(g', g),$$

*for all $g, g' \in C_b(\mathcal{X})$, where*

$$\varphi(f)(\mathrm{d}x) := \frac{e^{f(x)}}{\int_{\mathcal{X}} e^{f(y)}\pi(\mathrm{d}y)}\pi(\mathrm{d}x)$$

*If $g' = g$, then*

$$\nabla_{\mathcal{F}}^2 h^*(f)[g][g] = \mathrm{Var}_{\varphi(f)}(g).$$

**Example H.9** (Second-order Fréchet derivative of the $\chi^2$-divergence). *If $h$ is the $\chi^2$-divergence in Example 1.4, using Example H.5, we can show that $\nabla_{\mathcal{F}} h^*(f)$ is Fréchet differentiable on $C_b(\mathcal{X})$ with Fréchet derivative given by*

$$\nabla_{\mathcal{F}}^2 h^*(f)[g'][g] = \frac{1}{4} \int_{\mathcal{X}} g(z)g'(z)\pi(\mathrm{d}z),$$

*for all $g, g' \in C_b(\mathcal{X})$.*

**Definition H.10** (Third-order Fréchet differentiability). *Let $\mathcal{U} \subset C_b(\mathcal{X})$ be open and let $f \in \mathcal{U}$. Suppose that $H : \mathcal{U} \to \mathbb{R}$ is twice Fréchet differentiable (cf. Definition H.7) at $f$, and admits second-order Fréchet derivative $\nabla^2_{\mathcal{F}} H(f)$. Then $\nabla^2_{\mathcal{F}} H(f)$ is Fréchet differentiable at $f$, if there exists $T \in \mathcal{L}(C_b(\mathcal{X}), \mathcal{L}(C_b(\mathcal{X}), \mathcal{L}(C_b(\mathcal{X}), \mathbb{R})))$ such that for all $g, g', g'' \in C_b(\mathcal{X})$,*

$$\lim_{\|f''\|_\infty \to 0} \frac{|\nabla_{\mathcal{F}} H(f + g'')[g'][g] - \nabla_{\mathcal{F}} H(f)[g'][g] - T[g''][g'][g]|}{\|g''\|_\infty} = 0.$$

*If it exists, the map $T$ is unique, we write $T = \nabla^3_{\mathcal{F}} H(f)$, and call $\nabla^3_{\mathcal{F}} H(f)$ the third Fréchet derivative of $H$ at $f$.*

**Example H.11** (Third-order Fréchet derivative of the relative entropy). *If $h$ is the relative entropy in Example 1.3, using Example H.8, we can show that $\nabla^2_{\mathcal{F}} h^*(f)$ is Fréchet differentiable on $C_b(\mathcal{X})$ with Fréchet derivative $\nabla^3_{\mathcal{F}} h^*(f)$. We differentiate the variance $\mathrm{Var}_{\varphi(f)}(g)$ with respect to $f$ in a direction $g'$. Using the identity*

$$\frac{\mathrm{d}}{\mathrm{d}\varepsilon}\bigg|_{\varepsilon=0} \left( \int_{\mathcal{X}} G(x)\varphi(f + \varepsilon g')(\mathrm{d}x) \right) = \mathrm{Cov}_{\varphi(f)}(G, g'),$$

*we obtain*

$$\nabla^3_{\mathcal{F}} h^*(f)[g][g][g'] = \frac{\mathrm{d}}{\mathrm{d}\varepsilon}\bigg|_{\varepsilon=0} \left( \int_{\mathcal{X}} g(x)^2 \varphi(f + \varepsilon g')(\mathrm{d}x) - \left( \int_{\mathcal{X}} g(x)\varphi(f + \varepsilon g')(\mathrm{d}x) \right)^2 \right)$$

$$= \mathrm{Cov}_{\varphi(f)}(g^2, g') - 2\,\mathrm{Cov}_{\varphi(f)}(g, g') \int_{\mathcal{X}} g(x)\varphi(f)(\mathrm{d}x),$$

*for all $g, g' \in C_b(\mathcal{X})$. If $g' = g$, then*

$$\nabla^3_{\mathcal{F}} h^*(f)[g][g][g] = \int_{\mathcal{X}} \left( g(x) - \int_{\mathcal{X}} g(y)\varphi(f)(\mathrm{d}y) \right)^3 \varphi(f)(\mathrm{d}x).$$

**Example H.12** (Third-order Fréchet derivative of the $\chi^2$-divergence). *If $h$ is the $\chi^2$-divergence in Example 1.4, using Example H.9, we observe that since $\nabla^2_{\mathcal{F}} h^*(f)$ is independent of $f$, the third-order Fréchet derivative of $h^*$ is given by*

$$\nabla^3_{\mathcal{F}} h^*(f)[g][g][g] = 0,$$

*for all $g \in C_b(\mathcal{X})$.*

The motivation behind working with Fréchet instead of Gâteaux differentiability is that the higher-order derivatives in the case of the former could be identified with continuous symmetric multilinear maps. As proved in Section 3 of Chapter 1 from (Ambrosetti & Prodi, 1995), the space $\mathcal{L}(C_b(\mathcal{X}), \mathcal{L}(C_b(\mathcal{X}), \mathcal{L}(C_b(\mathcal{X}), \mathbb{R})))$ is isometrically isomorphic to $\mathcal{L}_3(C_b(\mathcal{X}), \mathbb{R})$, i.e., the space of continuous trilinear maps from $C_b(\mathcal{X}) \times C_b(\mathcal{X}) \times C_b(\mathcal{X})$ to $\mathbb{R}$, and therefore, we could naturally view the third-order Fréchet derivative of $H$, if it exists, as a continuous trilinear map. Furthermore, due to (Ambrosetti & Prodi, 1995, Theorem 3.5), we have that the third-order Fréchet derivative is always symmetric. On the contrary, the second-order Gâteaux derivative is not necessarily symmetric as noted on page 78 in (Ortega & Rheinboldt, 1970).

**Remark H.13.** *If we replace $C_b(\mathcal{X})$ with $\mathbb{R}^d$, then the first and second-order Fréchet derivatives are precisely the gradient and Hessian matrix of $H$ at $f$.*

**Proposition H.14** (Verification of Assumption 3.5 for the relative entropy in Example 1.3). *For $h$ being the relative entropy in Example 1.3, its third Frechet derivative in Example H.11 satisfies Assumption 3.5.*

*Proof.* Let $g \in C_b(\mathcal{X})$. Recall that

$$\nabla^3_{\mathcal{F}} h^*(f)[g][g][g] = \int_{\mathcal{X}} \left( g(x) - \int_{\mathcal{X}} g(y)\varphi(f)(\mathrm{d}y) \right)^3 \varphi(f)(\mathrm{d}x).$$

Note that since $\varphi(f) \in \mathcal{P}(\mathcal{X})$ we have

$$\left| g(x) - \int_{\mathcal{X}} g(y)\varphi(f)(\mathrm{d}y) \right| \le 2\|g\|_\infty,$$

and hence
$$|\nabla_{\mathcal{F}}^3 h^*(f)[g][g][g]| \le 8\|g\|_\infty^3.$$

Using the fact that
$$\left\|\nabla_{\mathcal{F}}^3 h^*(f)\right\|_{\mathcal{L}_3(C_b(\mathcal{X}),\mathbb{R})} := \sup_{\|g\|_\infty = 1} \frac{|\nabla_{\mathcal{F}}^3 h^*(f)[g][g][g]|}{\|g\|_\infty^3},$$

we conclude that
$$\left\|\nabla_{\mathcal{F}}^3 h^*(f)\right\|_{\mathcal{L}_3(C_b(\mathcal{X}),\mathbb{R})} \le 8,$$

for all $f \in C_b(\mathcal{X})$. $\qquad\square$

**Proposition H.15** (Verification of Assumption 3.5 for the relative $\chi^2$-divergence in Example 1.4). *For h being the $\chi^2$-divergence in Example 1.4, its third Frechet derivative in Example H.12 satisfies Assumption 3.5.*

*Proof.* Recall that
$$\nabla_{\mathcal{F}}^3 h^*(f)[g][g][g] = 0,$$

hence the conclusion is immediate. $\qquad\square$

## H.2 FIRST AND SECOND VARIATIONS

Following Chapter 2 from (Abraham et al., 2012), we introduce the notions of first and second variation for Fréchet differentiable functions $H$, relative to the duality pairing (6).

**Definition H.16** (First variation of $H$). *Let $H : C_b(\mathcal{X}) \to \mathbb{R}$ be Fréchet differentiable at $f \in C_b(\mathcal{X})$. If it exists, the first variation of $H$ at $f$ is the element $\frac{\delta H}{\delta f}(f) \in \mathcal{M}(\mathcal{X})$ such that, for all $g \in C_b(\mathcal{X})$,*
$$\left\langle g, \frac{\delta H}{\delta f}(f) \right\rangle := \nabla_{\mathcal{F}} H(f)[g].$$

**Example H.17** (First variation of the dual of relative entropy). *From Example H.2, we observe that the first variation $\frac{\delta h^*}{\delta f}(f) \in \mathcal{P}(\mathcal{X}) \subset \mathcal{M}(\mathcal{X})$ of $h^*$ at $f$ is given by*
$$\frac{\delta h^*}{\delta f}(f)(\mathrm{d}z) = \varphi(f)(\mathrm{d}z).$$

**Example H.18** (First variation of the dual of $\chi^2$-divergence). *From Example H.4, we observe that the first variation $\frac{\delta h^*}{\delta f} \in \mathcal{M}(\mathcal{X})$ of $h^*$ at $f$ is given by*
$$\frac{\delta h^*}{\delta f}(f)(\mathrm{d}z) = \left(1 + \frac{1}{2}f(z)\right)\pi(\mathrm{d}z).$$

Assuming that $H : C_b(\mathcal{X}) \to \mathbb{R}$ is Fréchet differentiable at $f \in C_b(\mathcal{X})$ with Fréchet derivative $\nabla_{\mathcal{F}} H(f)$, then it is Gâteaux differentiable (cf. Definition H.6) with the same derivative, and therefore the first variation of $H$ at $f$ can be characterized as
$$\left\langle g, \frac{\delta H}{\delta f}(f) \right\rangle = \lim_{\varepsilon \downarrow 0} \frac{1}{\varepsilon}\left(H\left(f + \varepsilon g\right) - H\left(f\right)\right), \tag{49}$$

for all $g \in C_b(\mathcal{X})$.

With the definition of first variation at hand, we can introduce necessary and sufficient conditions for $H$ to have an extremum at $f \in C_b(\mathcal{X})$.

**Lemma H.19** (Necessary first-order condition on $C_b(\mathcal{X})$). *Suppose $H : C_b(\mathcal{X}) \to \mathbb{R}$ admits first variation at $f$. If $H$ has an extremum at $f^*$, then it holds that*
$$\frac{\delta H}{\delta f}(f^*) = 0.$$

*Proof.* For a proof, see (Abraham et al., 2012, Proposition 2.4.22). $\qquad\square$

**Lemma H.20** (Sufficient first-order condition on $C_b(\mathcal{X})$)**.** *Let $\mathcal{U} \subset C_b(\mathcal{X})$ be non-empty and convex. Suppose that $H : \mathcal{U} \to \mathbb{R}$ admits first variation on $\mathcal{U}$ and is convex in the sense that, for all $\lambda \in [0,1]$, and all $f, g \in \mathcal{U}$, it holds that $H((1-\lambda)f + \lambda g) \leq (1-\lambda)H(f) + \lambda H(g)$. If $\frac{\delta H}{\delta f}(f^*) = 0$, for some $f^* \in \mathcal{U}$, then $f^*$ is a global minimum of $H$.*

**Remark H.21.** *An analogous result can be identically proved for concave functions and global maxima, so we will give the proof only for the convex case.*

*Proof.* Since $H$ is convex and admits first variation, following the argument in Lemma B.2, it can be showed that for any $f, g \in \mathcal{U}$

$$H(g) \geq H(f) + \left\langle g - f, \frac{\delta H}{\delta f}(f) \right\rangle.$$

For $f = f^*$ and using the assumption that $\frac{\delta H}{\delta f}(f^*) = 0$, we get

$$H(g) \geq H(f^*),$$

for all $g \in \mathcal{U}$, i.e., $f^*$ is a global minimum. $\qquad\square$

## I    Technical results on duality

In this section we state and prove some technical results which are central to the proof technique via dual Bregman divergence that we developed in Subsection 3.

**Proposition I.1.** *Let Assumption 1.1 hold. Let $h^* : C_b(\mathcal{X}) \to \mathbb{R}$ be the convex conjugate of $h$. Then, the following are equivalent:*

1. *The supremum of $\mathcal{M}(\mathcal{X}) \ni m \mapsto \langle g^*, m \rangle - h(m) \in \mathbb{R}$ is attained at $m = m^*$.*

2. *We have the first-order condition $g^*(x) - \frac{\delta h}{\delta m}(m^*, x) = 0$, for all $x \in \mathcal{X}$, $m^*$-a.e.*

3. *The supremum of $C_b(\mathcal{X}) \ni g \mapsto \langle g, m^* \rangle - h^*(g) \in \mathbb{R}$ is attained at $g = g^*$.*

4. *It holds that $m^* = \frac{\delta h^*}{\delta g}(g^*)$.*

*Proof.* (1) $\implies$ (2): Suppose that (1) holds. Then the supremum of $m \mapsto \langle g^*, m \rangle - h(m)$ is attained at the maximizer $m^* = \arg\max_{m \in \mathcal{M}(\mathcal{X})} \{\langle g^*, m \rangle - h(m)\}$. Hence,

$$\langle g^*, m^* - m \rangle - (h(m^*) - h(m)) \geq 0,$$

for all $m \in \mathcal{M}(\mathcal{X})$. Let $\tilde{m} \in \mathcal{M}(\mathcal{X})$ and set $m := m^* + t(\tilde{m} - m^*)$, for $t \in [0,1]$. Then

$$-t\langle g^*, \tilde{m} - m^* \rangle + (h(m^* + t(\tilde{m} - m^*)) - h(m^*)) \geq 0.$$

Dividing by $t$ and letting $t \searrow 0$ gives

$$-\langle g^*, \tilde{m} - m^* \rangle + \int_{\mathcal{X}} \frac{\delta h}{\delta m}(m^*, x)(\tilde{m} - m^*)(\mathrm{d}x) \geq 0,$$

or equivalently

$$\left\langle -g^* + \frac{\delta h}{\delta m}(m^*, \cdot), \tilde{m} - m^* \right\rangle \geq 0.$$

Since $\tilde{m}$ is arbitrary, $m^*$ satisfies the first-order condition

$$g^*(x) - \frac{\delta h}{\delta m}(m^*, x) = 0,$$

for all $x \in \mathcal{X}$, $m^*$-a.e.

(2) $\implies$ (1): Suppose that (2) holds. Observe that the map $m \mapsto \langle g^*, m \rangle - h(m)$ is strictly concave due to the strict convexity of $h$ and the linearity of $m \mapsto \langle g^*, m \rangle$. Therefore, $m^*$ is the maximizer of the map $\mathcal{M}(\mathcal{X}) \ni m \mapsto \langle g^*, m \rangle - h(m) \in \mathbb{R}$, and so (1) holds.

(3) $\implies$ (4): Suppose that (3) holds. Then the supremum in $g \mapsto \langle g, m^* \rangle - h^*(g)$ is attained at a maximizer $g^* \in \arg\max_{g \in C_b(\mathcal{X})} \{\langle g, m^* \rangle - h^*(g)\}$. Hence, by Lemma H.19, it follows that $g^*$ satisfies the first-order condition

$$m^* = \frac{\delta h}{\delta g}(g^*).$$

(4) $\implies$ (3): Suppose that (4) holds. Observe that $C_b(\mathcal{X})$ is convex and the map $g \mapsto \langle g, m^* \rangle - h^*(g)$ is concave due to the convexity of $h^*$ and the linearity of $g \mapsto \langle g, m^* \rangle$. Hence, by Lemma H.20, it follows that $g^*$ is a maximizer of the map $C_b(\mathcal{X}) \ni g \mapsto \langle g, m^* \rangle - h^*(g) \in \mathbb{R}$, and so (3) holds.

(1) $\implies$ (3): Suppose that (1) holds. Then, by Definition 3.1, we have that $h^*(g) = \langle g, m^* \rangle - h(m^*)$, and equivalently $h(m^*) = \langle g, m^* \rangle - h^*(g)$. Clearly, $\mathcal{M}(\mathcal{X})$ is convex and $(\mathcal{M}(\mathcal{X}), \mathrm{TV})$ is Hausdorff since it is a metric space, hence we can apply the Fenchel-Moreau theorem (Zalinescu, 2002, Theorem 2.3.3) to conclude that $h^{**} = h$, i.e., $h(m^*) = \sup_{g \in C_b(\mathcal{X})}\{\langle g, m^* \rangle - h^*(g)\}$. Therefore, $h(m^*)$ is the supremum of $g \mapsto \langle g, m^* \rangle - h^*(g)$ attained at $g = g^*$.

(3) $\implies$ (1): Suppose (3) holds. Then $h^{**}(m^*) = \langle g^*, m^* \rangle - h^*(g^*)$, or equivalently $h^*(g^*) = \langle g^*, m^* \rangle - h^{**}(m^*)$. Again, by the Fenchel-Moreau theorem (Zalinescu, 2002, Theorem 2.3.3), $h^{**}(m) = h(m)$, for all $m \in \mathcal{M}(\mathcal{X})$, and hence $h^*(g^*) = \langle g^*, m^* \rangle - h(m^*)$. Hence, by Definition 3.1, the supremum of $m \mapsto \langle g^*, m \rangle - h(m)$ is realized at $m = m^*$. $\qquad\square$

**Corollary I.2.** *Let $h^* : C_b(\mathcal{X}) \to \mathbb{R}$ be the convex conjugate of $h$. If Assumption 1.1 holds and $h^*$ admits the first variation $\frac{\delta h^*}{\delta f}(f)$ (cf. (49)) on $C_b(\mathcal{X})$, then*

$$\frac{\delta h^*}{\delta f}(f) = \arg\max_{m \in \mathcal{M}(\mathcal{X})} \{\langle f, m \rangle - h(m)\}. \tag{50}$$

**Remark I.3** (Bregman divergence via first variation). *Definition 3.2 can be relaxed as follows. Provided that $h^*$ admits a first variation (see Examples H.17 and H.18), Corollary I.2 shows that if Assumption 1.1 holds, then the first variation $\frac{\delta h^*}{\delta f}(f)$ of $h^*$ at $f$ is the unique maximizer of $m \mapsto \langle f, m \rangle - h(m)$. Consequently, from Definition H.16, since $f, f' \in C_b(\mathcal{X})$ and $\frac{\delta h^*}{\delta f}(f) \in \mathcal{M}(\mathcal{X})$, it follows that $\nabla_{\mathcal{F}} h^*(f)[f' - f] = \langle f' - f, \frac{\delta h^*}{\delta f}(f)\rangle$. Moreover, because $h^*$ is Fréchet-convex, $D_{h^*}(f', f) \geq 0$, for all $f', f \in C_b(\mathcal{X})$.*

**Lemma I.4.** *Let Assumption 1.1 hold. Let $h^* : C_b(\mathcal{X}) \to \mathbb{R}$ be the convex conjugate of $h$. Fix $f, g \in C_b(\mathcal{X})$ and $\mu, \mu' \in \mathcal{E}$. If $f(z) = \frac{\delta h}{\delta m}(\mu, z)$ and $g(z) = \frac{\delta h}{\delta m}(\mu', z)$, for all $z \in \mathcal{X}$, $\mu$-a.e. and $\mu'$-a.e., respectively, then*

$$D_{h^*}(f, g) = D_h(\mu', \mu).$$

*Proof.* By Definition 3.2, we have that

$$D_{h^*}(f, g) = h^*(f) - h^*(g) - \int_{\mathcal{X}} (f(z) - g(z)) \frac{\delta h^*}{\delta g}(g)(\mathrm{d}z)$$

$$= \langle f, \mu \rangle - h(\mu) - \langle g, \mu' \rangle + h(\mu') - \int_{\mathcal{X}} (f(z) - g(z)) \frac{\delta h^*}{\delta g}(g)(\mathrm{d}z)$$

$$= h(\mu') - h(\mu) + \int_{\mathcal{X}} \frac{\delta h}{\delta m}(\mu, z)\mu(\mathrm{d}z) - \int_{\mathcal{X}} \frac{\delta h}{\delta m}(\mu', z)\mu'(\mathrm{d}z) - \int_{\mathcal{X}} \left(\frac{\delta h}{\delta m}(\mu, z) - \frac{\delta h}{\delta m}(\mu', z)\right)\mu'(\mathrm{d}z)$$

$$= h(\mu') - h(\mu) - \int_{\mathcal{X}} \frac{\delta h}{\delta m}(\mu, z)(\mu' - \mu)(\mathrm{d}z) = D_h(\mu', \mu),$$

where the second and third equalities follow from Lemma I.1 and Corollary I.2, while the last equality follows from the definition of the Bregman divergence. $\qquad\square$

**Lemma I.5.** *Consider Algorithms 1 and 2. Let Assumption 1.1 hold. Let $h^* : C_b(\mathcal{X}) \to \mathbb{R}$ be the convex conjugate of $h$. For each $n \geq 0$, fix $f^n, g^n \in C_b(\mathcal{X})$, $\nu^n \in \mathcal{C}$ and $\mu^n \in \mathcal{D}$. If $f^n = \frac{\delta h}{\delta \nu}(\nu^n, \cdot)$ and $g^n = \frac{\delta h}{\delta \mu}(\mu^n, \cdot)$, then, for any $n \geq 0$, we have that*

$$D_h(\nu^{n+1}, \nu^n) = D_{h^*}(f^n, f^{n+1}), \quad D_h(\nu^n, \nu^{n+1}) = D_{h^*}(f^{n+1}, f^n),$$
$$D_h(\mu^{n+1}, \mu^n) = D_{h^*}(g^n, g^{n+1}), \quad D_h(\mu^n, \mu^{n+1}) = D_{h^*}(g^{n+1}, g^n).$$

*Proof.* First, observe that due to Assumption 1.1, the pairs $(\nu^{n+1}, \mu^{n+1})$ in (1) and (2) are unique. We will only present the proof for (1) since the argument for (2) is identical. The updates in (1) can be equivalently written as

$$
\begin{aligned}
\nu^{n+1} &= \underset{\nu \in \mathcal{C}}{\arg\min} \left\{ \int_{\mathcal{X}} \frac{\delta F}{\delta \nu}(\nu^n, \mu^n, x)(\nu - \nu^n)(\mathrm{d}x) + \frac{1}{\tau} D_h(\nu, \nu^n) \right\} \\
&= \underset{\nu \in \mathcal{C}}{\arg\min} \left\{ \int_{\mathcal{X}} \tau \frac{\delta F}{\delta \nu}(\nu^n, \mu^n, x)(\nu - \nu^n)(\mathrm{d}x) + h(\nu) - h(\nu^n) - \int_{\mathcal{X}} \frac{\delta h}{\delta \nu}(\nu^n, x)(\nu - \nu^n)(\mathrm{d}x) \right\} \\
&= \underset{\nu \in \mathcal{C}}{\arg\min} \left\{ \int_{\mathcal{X}} \left( \tau \frac{\delta F}{\delta \nu}(\nu^n, \mu^n, x) - \frac{\delta h}{\delta \nu}(\nu^n, x) \right)(\nu - \nu^n)(\mathrm{d}x) + h(\nu) \right\} \\
&= \underset{\nu \in \mathcal{C}}{\arg\max} \left\{ \int_{\mathcal{X}} \left( \frac{\delta h}{\delta \nu}(\nu^n, x) - \tau \frac{\delta F}{\delta \nu}(\nu^n, \mu^n, x) \right)(\nu - \nu^n)(\mathrm{d}x) - h(\nu) \right\} \\
&= \underset{\nu \in \mathcal{C}}{\arg\max} \left\{ \int_{\mathcal{X}} \left( \frac{\delta h}{\delta \nu}(\nu^n, x) - \tau \frac{\delta F}{\delta \nu}(\nu^n, \mu^n, x) \right)\nu(\mathrm{d}x) - h(\nu) \right\},
\end{aligned}
\tag{51}
$$

and

$$
\begin{aligned}
\mu^{n+1} &= \underset{\mu \in \mathcal{D}}{\arg\max} \left\{ \int_{\mathcal{X}} \frac{\delta F}{\delta \mu}(\nu^n, \mu^n, y)(\mu - \mu^n)(\mathrm{d}y) - \frac{1}{\tau} D_h(\mu, \mu^n) \right\} \\
&= \underset{\mu \in \mathcal{D}}{\arg\max} \left\{ \int_{\mathcal{X}} \tau \frac{\delta F}{\delta \mu}(\nu^n, \mu^n, y)(\mu - \mu^n)(\mathrm{d}y) - h(\mu) + h(\mu^n) + \int_{\mathcal{X}} \frac{\delta h}{\delta \mu}(\mu^n, y)(\mu - \mu^n)(\mathrm{d}y) \right\} \\
&= \underset{\mu \in \mathcal{D}}{\arg\max} \left\{ \int_{\mathcal{X}} \left( \frac{\delta h}{\delta \mu}(\mu^n, y) + \tau \frac{\delta F}{\delta \mu}(\nu^n, \mu^n, y) \right)(\mu - \mu^n)(\mathrm{d}y) - h(\mu) \right\} \\
&= \underset{\mu \in \mathcal{D}}{\arg\max} \left\{ \int_{\mathcal{X}} \left( \frac{\delta h}{\delta \mu}(\mu^n, y) + \tau \frac{\delta F}{\delta \mu}(\nu^n, \mu^n, y) \right)\mu(\mathrm{d}y) - h(\mu) \right\}.
\end{aligned}
\tag{52}
$$

Using the notation $f^n = \frac{\delta h}{\delta \nu}(\nu^n, \cdot)$ and $g^n = \frac{\delta h}{\delta \mu}(\mu^n, \cdot)$, for each $n \geq 0$, the first-order conditions for (1) in Proposition B.3 can be equivalently written as

$$
f^{n+1}(x) - f^n(x) = -\tau \frac{\delta F}{\delta \nu}(\nu^n, \mu^n, x),
\tag{53}
$$

$$
g^{n+1}(y) - g^n(y) = \tau \frac{\delta F}{\delta \mu}(\nu^n, \mu^n, y),
\tag{54}
$$

for all $(x, y) \in \mathcal{X} \times \mathcal{X}$, $\nu^{n+1}$-a.e. and $\mu^{n+1}$-a.e., respectively. Then, using (50), (51) becomes

$$
\begin{aligned}
\nu^{n+1} &= \underset{\nu \in \mathcal{C}}{\arg\max} \left\{ \int_{\mathcal{X}} \left( f^n(x) - \tau \frac{\delta F}{\delta \nu}(\nu^n, \mu^n, x) \right)\nu(\mathrm{d}x) - h(\nu) \right\} \\
&= \underset{\nu \in \mathcal{C}}{\arg\max} \left\{ \int_{\mathcal{X}} f^{n+1}(x)\nu(\mathrm{d}x) - h(\nu) \right\} = \frac{\delta h^*}{\delta f}(f^{n+1}),
\end{aligned}
\tag{55}
$$

for all $n \geq 0$. Similarly, from (52), we have that

$$
\mu^{n+1} = \frac{\delta h^*}{\delta f}(g^{n+1}),
\tag{56}
$$

for all $n \geq 0$. The conclusion follows directly from Lemma I.4. $\qquad\square$

# J CONVERGENCE OF THE CONTINUOUS-TIME DYNAMICS AND THE MDA IMPLICIT ALGORITHM

In this section, we provide a formal calculation showing that the continuous-time gradient flow obtained by taking the limit $\tau \to 0$ in the dual iterative MDA schemes of Proposition B.3 converges at rate $\mathcal{O}(1/t)$ in NI for the time-averaged flows.

Moreover, we show that an implicit Euler discretization of this gradient flow achieves a linear convergence rate $\mathcal{O}(1/N)$, matching the continuous-time rate under the same convexity–concavity assumptions on $F$. However, this implicit scheme is not practically implementable, unlike the explicit Algorithms 1 and 2.

Formally letting $\tau \to 0$ in the updates of Proposition B.3 yields the continuous-time flow

$$\partial_t \frac{\delta h}{\delta \nu}(\nu_t, x) = -\frac{\delta F}{\delta \nu}(\nu_t, \mu_t, x), \quad \partial_t \frac{\delta h}{\delta \mu}(\mu_t, y) = \frac{\delta F}{\delta \mu}(\nu_t, \mu_t, y), \quad t > 0, \qquad (57)$$

with initial condition $(\nu_0, \mu_0) \in \mathcal{C} \times \mathcal{D}$. For convenience, we assume this flow is well-posed, i.e., it admits a unique solution $(\nu_t, \mu_t)_{t \geq 0}$.

For any $(\nu, \mu) \in \mathcal{C} \times \mathcal{D}$, and assuming the interchange of derivatives and integrals is valid, a direct calculation gives

$$\partial_t D_h(\nu, \nu_t) = \partial_t \left( h(\nu) - h(\nu_t) - \int_{\mathcal{X}} \frac{\delta h}{\delta \nu}(\nu_t, x)(\nu - \nu_t)(\mathrm{d}x) \right)$$

$$= -\partial_t h(\nu_t) - \partial_t \int_{\mathcal{X}} \frac{\delta h}{\delta \nu}(\nu_t, x)(\nu - \nu_t)(\mathrm{d}x)$$

$$= -\int_{\mathcal{X}} \frac{\delta h}{\delta \nu}(\nu_t, x) \partial_t \nu_t(\mathrm{d}x) - \int_{\mathcal{X}} \partial_t \frac{\delta h}{\delta \nu}(\nu_t, x)(\nu - \nu_t)(\mathrm{d}x) - \int_{\mathcal{X}} \frac{\delta h}{\delta \nu}(\nu_t, x) \partial_t(\nu - \nu_t)(\mathrm{d}x)$$

$$= -\int_{\mathcal{X}} \frac{\delta h}{\delta \nu}(\nu_t, x) \partial_t \nu_t(\mathrm{d}x) - \int_{\mathcal{X}} \partial_t \frac{\delta h}{\delta \nu}(\nu_t, x)(\nu - \nu_t)(\mathrm{d}x) + \int_{\mathcal{X}} \frac{\delta h}{\delta \nu}(\nu_t, x) \partial_t \nu_t(\mathrm{d}x)$$

$$= \int_{\mathcal{X}} \frac{\delta F}{\delta \nu}(\nu_t, \mu_t, x)(\nu - \nu_t)(\mathrm{d}x).$$

Following the same calculation for $D_h(\mu, \mu_t)$ we obtain

$$\partial_t D_h(\mu, \mu_t) = -\int_{\mathcal{X}} \frac{\delta F}{\delta \mu}(\nu_t, \mu_t, y)(\mu - \mu_t)(\mathrm{d}y).$$

Adding these and applying the convexity–concavity of $F$ (Assumption 1.5) yields

$$\partial_t \left( D_h(\nu, \nu_t) + D_h(\mu, \mu_t) \right) \leq F(\nu, \mu_t) - F(\nu_t, \mu_t) + F(\nu_t, \mu_t) - F(\nu_t, \mu).$$

Integrating, dividing by $t$ and applying Jensen's inequality to $F$ gives

$$F\left( \frac{1}{t} \int_0^t \nu_s \mathrm{d}s, \mu \right) - F\left( \nu, \frac{1}{t} \int_0^t \mu_s \mathrm{d}s \right) \leq \frac{1}{t} \left( \sup_{\nu \in \mathcal{C}} D_h(\nu, \nu_0) + \sup_{\mu \in \mathcal{D}} D_h(\mu, \mu_0) \right).$$

Hence, maximizing over $(\nu, \mu)$, we conclude that

$$\mathrm{NI}\left( \frac{1}{t} \int_0^t \nu_s \mathrm{d}s, \frac{1}{t} \int_0^t \mu_s \mathrm{d}s \right) \leq \frac{1}{t} \left( \sup_{\nu \in \mathcal{C}} D_h(\nu, \nu_0) + \sup_{\mu \in \mathcal{D}} D_h(\mu, \mu_0) \right),$$

establishing the $\mathcal{O}(1/t)$ rate.

We now turn to the implicit MDA scheme. For a given stepsize $\tau > 0$, and fixed initial pair of strategies $(\nu_0, \mu_0) \in \mathcal{C} \times \mathcal{D}$, for $n \geq 0$, the *implicit* MDA algorithm is defined by

---

**Algorithm 7:** IMPLICIT MDA

---

**Input:** Objective function F, initial measures $(\nu_0, \mu_0)$, stepsize $\tau > 0$

**for** $n = 0, 1, \ldots, N-1$ **do**

$\quad \nu^{n+1} = \arg\min_{\nu \in \mathcal{C}} \{ \int_{\mathcal{X}} \frac{\delta F}{\delta \nu}(\nu^n, \mu^{n+1}, x)(\nu - \nu^n)(\mathrm{d}x) + \frac{1}{\tau} D_h(\nu, \nu^n) \}$,

$\quad \mu^{n+1} = \arg\max_{\mu \in \mathcal{D}} \{ \int_{\mathcal{X}} \frac{\delta F}{\delta \mu}(\nu^{n+1}, \mu^n, y)(\mu - \mu^n)(\mathrm{d}y) - \frac{1}{\tau} D_h(\mu, \mu^n) \}$

**Output:** $\left( \frac{1}{N} \sum_{n=0}^{N-1} \nu^{n+1}, \frac{1}{N} \sum_{n=0}^{N-1} \mu^n \right)$

---

**Theorem J.1** (Convergence of the implicit MDA Algorithm 7). *Let $(\nu^0, \mu^0)$ be such that $\sup_{\nu \in \mathcal{C}} D_h(\nu, \nu^0) + \sup_{\mu \in \mathcal{D}} D_h(\mu, \mu^0) < \infty$. Let Assumption 1.1, 1.5 and 3.3 hold. Suppose that $\tau L \leq 1$, where $L := \max\{L_\nu, L_\mu\}$. Then, we have*

$$\mathrm{NI}\left(\frac{1}{N}\sum_{n=0}^{N-1}\nu^{n+1}, \frac{1}{N}\sum_{n=0}^{N-1}\mu^{n+1}\right) \leq \frac{1}{N\tau}\left(\sup_{\nu \in \mathcal{C}} D_h(\nu, \nu^0) + \sup_{\mu \in \mathcal{D}} D_h(\mu, \mu^0)\right).$$

*Proof.* Since $\nu \mapsto \tau \int \frac{\delta F}{\delta \nu}(\nu^n, \mu^{n+1}, x)(\nu - \nu^n)(\mathrm{d}x)$ is convex, applying Lemma B.1 with $\bar{\nu} = \nu^{n+1}$ and $\mu = \nu^n$ implies that, for any $\nu \in \mathcal{C}$, we have

$$\tau \int \frac{\delta F}{\delta \nu}(\nu^n, \mu^{n+1}, x)(\nu - \nu^n)(\mathrm{d}x) + D_h(\nu, \nu^n) \geq \tau \int \frac{\delta F}{\delta \nu}(\nu^n, \mu^{n+1}, x)(\nu^{n+1} - \nu^n)(\mathrm{d}x)$$
$$+ D_h(\nu^{n+1}, \nu^n) + D_h(\nu, \nu^{n+1}),$$

or, equivalently,

$$-\tau \int \frac{\delta F}{\delta \nu}(\nu^n, \mu^{n+1}, x)(\nu - \nu^n)(\mathrm{d}x) - D_h(\nu, \nu^n) \leq -\tau \int \frac{\delta F}{\delta \nu}(\nu^n, \mu^{n+1}, x)(\nu^{n+1} - \nu^n)(\mathrm{d}x)$$
$$- D_h(\nu^{n+1}, \nu^n) - D_h(\nu, \nu^{n+1}). \quad (58)$$

Similarly, since $\mu \mapsto -\tau \int \frac{\delta F}{\delta \mu}(\nu^{n+1}, \mu^n, y)(\mu - \mu^n)(\mathrm{d}y)$ is convex, applying Lemma B.1 with $\bar{\nu} = \mu^{n+1}$ and $\mu = \mu^n$ implies that, for any $\mu \in \mathcal{D}$, we have

$$\tau \int \frac{\delta F}{\delta \mu}(\nu^{n+1}, \mu^n, y)(\mu - \mu^n)(\mathrm{d}y) - D_h(\mu, \mu^n) \leq \tau \int \frac{\delta F}{\delta \mu}(\nu^{n+1}, \mu^n, y)(\mu^{n+1} - \mu^n)(\mathrm{d}y)$$
$$- D_h(\mu^{n+1}, \mu^n) - D_h(\mu, \mu^{n+1}). \quad (59)$$

Using the convexity of $\nu \mapsto F(\nu, \mu)$ in (58), with $\nu = \nu^n$ and $\mu = \mu^{n+1}$, we have that

$$F(\nu^n, \mu^{n+1}) - F(\nu, \mu^{n+1}) - \frac{1}{\tau}D_h(\nu, \nu^n) \leq \int_{\mathcal{X}} \frac{\delta F}{\delta \nu}(\nu^n, \mu^{n+1}, x)(\nu^n - \nu^{n+1})(\mathrm{d}x)$$
$$- \frac{1}{\tau}D_h(\nu^{n+1}, \nu^n) - \frac{1}{\tau}D_h(\nu, \nu^{n+1}). \quad (60)$$

From $L_\nu$-relative smoothness and the fact that $\tau L \leq 1$, it follows that

$$F(\nu^{n+1}, \mu^{n+1}) \leq F(\nu^n, \mu^{n+1}) + \int_{\mathcal{X}} \frac{\delta F}{\delta \nu}(\nu^n, \mu^{n+1}, x)(\nu^{n+1} - \nu^n)(\mathrm{d}x) + L_\nu D_h(\nu^{n+1}, \nu^n)$$
$$\leq F(\nu^n, \mu^{n+1}) + \int_{\mathcal{X}} \frac{\delta F}{\delta \nu}(\nu^n, \mu^{n+1}, x)(\nu^{n+1} - \nu^n)(\mathrm{d}x) + \frac{1}{\tau}D_h(\nu^{n+1}, \nu^n). \quad (61)$$

Hence, combining (60) with (61), we obtain that

$$F(\nu^n, \mu^{n+1}) - F(\nu, \mu^{n+1}) - \frac{1}{\tau}D_h(\nu, \nu^n) \leq F(\nu^n, \mu^{n+1}) - F(\nu^{n+1}, \mu^{n+1}) - \frac{1}{\tau}D_h(\nu, \nu^{n+1}).$$
$$(62)$$

Similarly, using concavity of $\mu \mapsto F(\nu, \mu)$ in (59), with $\nu = \nu^{n+1}$ and $\mu = \mu^n$, we have that

$$F(\nu^{n+1}, \mu) - F(\nu^{n+1}, \mu^n) - \frac{1}{\tau}D_h(\mu, \mu^n) \leq \int_{\mathcal{X}} \frac{\delta F}{\delta \mu}(\nu^{n+1}, \mu^n, y)(\mu^{n+1} - \mu^n)(\mathrm{d}y)$$
$$- \frac{1}{\tau}D_h(\mu^{n+1}, \mu^n) - \frac{1}{\tau}D_h(\mu, \mu^{n+1}). \quad (63)$$

From $L_\mu$-relative smoothness and the fact that $\tau L \leq 1$, it follows that

$$F(\nu^{n+1}, \mu^{n+1}) \geq F(\nu^{n+1}, \mu^n) + \int_{\mathcal{X}} \frac{\delta F}{\delta \mu}(\nu^{n+1}, \mu^n, y)(\mu^{n+1} - \mu^n)(\mathrm{d}y) - L_\mu D_h(\mu^{n+1}, \mu^n)$$
$$\geq F(\nu^{n+1}, \mu^n) + \int_{\mathcal{X}} \frac{\delta F}{\delta \mu}(\nu^{n+1}, \mu^n, y)(\mu^{n+1} - \mu^n)(\mathrm{d}y) - \frac{1}{\tau}D_h(\mu^{n+1}, \mu^n). \quad (64)$$

Hence, combining (63) with (64), we obtain that

$$F(\nu^{n+1}, \mu) - F(\nu^{n+1}, \mu^n) - \frac{1}{\tau} D_h(\mu, \mu^n) \leq F(\nu^{n+1}, \mu^{n+1}) - F(\nu^{n+1}, \mu^n) - \frac{1}{\tau} D_h(\mu, \mu^{n+1}). \tag{65}$$

Adding inequalities (62) and (65) implies that

$$F(\nu^{n+1}, \mu) - F(\nu, \mu^{n+1}) \leq F(\nu^{n+1}, \mu^{n+1}) - F(\nu^{n+1}, \mu^{n+1})$$
$$+ \frac{1}{\tau} D_h(\nu, \nu^n) + \frac{1}{\tau} D_h(\mu, \mu^n) - \frac{1}{\tau} D_h(\nu, \nu^{n+1}) - \frac{1}{\tau} D_h(\mu, \mu^{n+1})$$

Summing the previous inequality over $n = 0, 1, ..., N - 1$, bounding the right-hand side from above by its supremum over $(\nu, \mu)$, dividing by $N$, applying Jensen's inequality and taking maximum over $(\nu, \mu)$ in the left-hand side leads to

$$\mathrm{NI}\left(\frac{1}{N}\sum_{n=0}^{N-1} \nu^{n+1}, \frac{1}{N}\sum_{n=0}^{N-1} \mu^{n+1}\right) \leq \frac{1}{N\tau}\left(\sup_{\nu \in \mathcal{C}} D_h(\nu, \nu^0) + \sup_{\mu \in \mathcal{D}} D_h(\mu, \mu^0)\right),$$

where the last inequality follows since $D_h(\nu, \nu^N) + D_h(\mu, \mu^N) \geq 0$, for all $(\nu, \mu) \in \mathcal{C} \times \mathcal{D}$. $\qquad\square$

## K    FURTHER RELATED WORKS

Besides the vanilla MDA algorithm, (Hsieh et al., 2019) considers the entropic Mirror Prox algorithm, which requires the computation of an extra gradient at an intermediate point and two projections onto the dual space. Although it is proved in (Hsieh et al., 2019) that the Mirror Prox algorithm achieves $\mathcal{O}\left(N^{-1}\right)$ convergence rate for deterministic gradients, it is also outlined that for stochastic gradients (which one has typically access to in practice) Mirror Prox and simultaneous MDA achieve the same rate $\mathcal{O}\left(N^{-1/2}\right)$.

Another approach based on reproducing kernel Hilbert spaces (RKHS) is developed in (Dvurechensky & Zhu, 2024) and achieves the same convergence rates $\mathcal{O}\left(N^{-1}\right)$ and $\mathcal{O}\left(N^{-1/2}\right)$ for the deterministic and stochastic Mirror Prox algorithm, respectively. To our knowledge, the analysis of a alternating version of the Mirror Prox algorithm has not appeared in the literature.

