# OpenReview forum: "Mirror Descent-Ascent for mean-field min-max problems"
_ICLR.cc/2026/Conference — Submitted to ICLR 2026_

### Official Review · Reviewer_dZnt · 2025-10-26

**Soundness:** 3
**Presentation:** 2
**Contribution:** 3
**Rating:** 6
**Confidence:** 2

**Summary:**

This paper studies the convergence properties of simultaneous and sequential mirror descent for solving min-max games. The paper shows that under some technical assumptions, the convergence rates (in terms of NI error) are $O(N^{-1/2})$ in the simultaneous case and $O(N^{-2/3})$ in the sequential case. An example of training GANs is implemented with both variants of MD, and the behavior corroborates the faster convergence rate of sequential MD.

**Strengths:**

- The main technical results are clearly presented and while the notation is quite heavy, the authors do a good job of making the paper readable.
- The convergence result for sequential MD is the first such result for mean-field min-max games, and the proof technique is quite interesting (if reliant on several technical assumptions).
- The motivating example of GANs is clearly presented and the experimental results nicely corroborate the theoretical statements.

**Weaknesses:**

- A concern is how extensible the convergence results are -- it is not made clear what other problem settings would satisfy the required technical assumptions to obtain a fast convergence rate. In particular, Assumption 3.4 seems like a very restrictive condition on the second variation, and I believe it would improve the paper to show more examples of divergences and min-max problems that satisfy all technical assumptions.
- Beyond GANs and mean field neural nets, not much is said about the other concrete applications of the framework. Considering that prior work has shown connections and examples ranging from Sinkhorn and EM algorithms to reinforcement learning, a natural question which is not investigated is how sequential MD performs in these settings. I would be more positive on the paper if it can be shown that sequential MD is a superior method for other applications. As it stands, the theoretical results are novel but the significance and applicability of the results are not convincing.
- The notation in the paper is quite dense (and somewhat unavoidable given the topic), but giving some additional clarifications on the assumptions and also providing more detailed proof sketches (in particular emphasizing the way the proofs diverge from standard techniques) would help readers who might be unfamiliar with the material.

**Questions:**

- The MD variants studied are Euler discretizations of the Fisher-Rao gradient flow in continuous time. I am curious if other discrete-time algorithms which can be obtained by taking different discretizations? Does the continuous-time convergence rate given insights into the behavior of its discretizations?

---

> ### Author Response · Authors · 2025-11-28
> **Response to the reviewer**
>
> We would like to thank the reviewer for appreciating the contribution of our paper, for their feedback and questions. We would like to address the weaknesses and questions formulated by the reviewer:
> ### Weaknesses
> - Due to the technical issues raised by reviewer pJx5, we replaced Assumption 3.4 with the new Assumption 3.5, which serves as the infinite-dimensional analogue of the smoothness condition used in [Theorem 3.2, Wibisono et al. (2022), Alternating Mirror Descent for Constrained Min–Max Games]. Although Assumption 3.5 may appear restrictive, it is satisfied by many standard choices of $h$. In the revised manuscript, we added Propositions H.14 and H.15, where we explicitly verify that the duals of the entropy and chi-squared divergence admit well-defined third-order Fréchet derivatives and indeed satisfy Assumption 3.5. We have also expanded the paper with Appendix E, presenting one additional application of our framework: two-player zero-sum Markov games, a class of problems in multi-agent reinforcement learning. We verify that the corresponding objective function satisfies our assumptions.
>
> - We would like to clarify that the Sinkhorn and EM algorithms fall under the category of pure minimization problems, rather than min–max problems. Both can be formulated as min–min optimization over their respective objective functions; see [Section 4 (Applications to Sinkhorn and EM with convergence rates) in Aubin-Frankowski et al., Mirror Descent with Relative Smoothness in Measure Spaces.] As noted above, we have also added one further application in Appendix E: two-player zero-sum Markov games to demonstrate the applicability of our general framework.
>
> - In the revised version, we have added sketches of our proofs (cf. Sections 2.1 and 3.1) and provided additional comments about our assumptions, in order to improve the clarity of the paper.
>
> ### Questions:
> - Continuous-time analysis provides useful intuition about which gradient flows converge to minimizers/maximizers, under what assumptions, and at what rates. It can therefore guide the design of discrete-time algorithms. However, discrete-time analysis must ultimately be carried out independently, as convergence properties depend on the choice of discretization (such as mirror/JKO-type schemes or interacting particle systems). In particular, our paper highlights the importance of choosing between simultaneous and alternating mirror schemes.
>
> In our min–max games setting, continuous-time analysis typically reveals an "ideal" convergence rate of the gradient flow, by which we mean a rate that may not be achievable by discrete-time algorithms. This is now explained in Remark 1.7. In Appendix J, we provide a formal calculation showing that the continuous time gradient flow yields a $1/t$ convergence rate, which is faster than both the simultaneous and alternating discrete-time schemes. Other discretizations, such as the implicit MDA scheme (Appendix J), can match this rate in theory (i.e., $1/N$), but are not practically implementable since they require knowledge of the opponent’s next move.
>
> Crucially, continuous-time analysis does not require the technical tools needed in discrete-time, such as the three-point lemma or analysis in the dual space. This makes the discrete-time analysis significantly more delicate and technically involved.

---

### Official Review · Reviewer_pJx5 · 2025-10-28

**Soundness:** 2
**Presentation:** 3
**Contribution:** 3
**Rating:** 2
**Confidence:** 4

**Summary:**

This paper studies mirror descent (MD) algorithms on spaces of measures. It provides analysis for both the simultaneous and sequential (a.k.a. alternating) versions of MD for general convex-concave objectives and general objectives, generalizing existing results.

**Strengths:**

The generalization of existing results to this setting requires, at least in my understanding, a significant amount of technical work, and the paper seems to do this well. The paper is also generally well-written and clear.

**Weaknesses:**

### Issue with the spaces under consideration
I have trouble understanding on which spaces assumptions and statements are valid.

In particular, it is not clear how Assumption 3.4 can be satisfied for the entropy regularizer with the current wording.
Indeed, Assumption 3.4 requires the existence of a constant $L_{h^* }$ such that the inequality l368 holds uniformly over all bounded functions.
But the constant $L_{h^* }$ given by Prop. G.15 depends on the functions themselves (through their sup norm).
Moreover, in this proof, the authors refer to (Lascu et al., 2025, Lemma A.2) for the Lipschitzness of $\phi$ but I could not find this result in the lemma mentioned (which is several pages long). I also do not see how to obtain such a pointwise Lipschitz bound.
(This issue is the main reason behind my current rating.)

Moreover, in example 1.3, the authors appear to have to restrict the space of probability measures to those with density wthin bounded distance from a reference distribution. This seems to be restrictive compared to (Hsieh et al., 2019) for instance.

### Lack of examples of divergence
In its comparison to previous work, this paper insists on the generality of the divergences considered compared to previous works which were only on KL divergence.
However, most of the assumptions are verified only for the entropy regularizer, see Asm 2.1 and 3.4.
Moreover, for the examples 1.2 and E, the choice of divergence is not discussed, and the reader is refered to Remark 2.2. If I understand correctly, this means that these examples are only valid when $h$ satisfies the inequality l286. If this is the case, this should be made more explicit in the paper and this assumption should be better discussed.


### Minor
- l260-267 make it sound as if the form of the update is novel while, unless I am mistaken, it is standard in mirror methods.
- The appendix is quite long, it could benefit from a table of contents and an introductory section explaining its organization.

**Questions:**

Could the authors address "Issue with the spaces under consideration" and "Lack of examples of divergence" in the section above?

---

> ### Author Response · Authors · 2025-11-28
> **Response to the reviewer**
>
> We would like to thank the reviewer for appreciating the contribution of our paper, for their feedback and questions. We would like to address the weaknesses and questions formulated by the reviewer:
>
> ### Issue with the spaces under consideration
> We understand the reviewer's concern and we thank the reviewer for carefully checking the proof. Indeed, our assumption on the second variation of the dual map was a local Lipschitz assumption, rather than a global one. We also agree that the Lipschitzness of $\varphi$ is not pointwise but with respect to the $L^\infty$ norm. We fixed the assumption (the newly added Assumption 3.5) by adopting the analogous condition used in [Theorem 3.2, Wibisono et al., Alternating Mirror Descent for Constrained Min–Max Games]. Specifically, we now assume that the third-order Fréchet derivative of the dual map is uniformly bounded. In Propositions H.14 and H.15, we verify that this assumption holds for the cases of interest, namely, when $h$ is the entropy or the chi-squared divergence.
>
> We emphasize that the restriction in Example 1.3 is necessary to rigorously guarantee the existence of the flat derivative of the relative entropy (which was not fully addressed in (Hsieh et al., 2019)). In Theorem 4, part 2 of (Hsieh et al., 2019), the authors provide a heuristic argument suggesting that the entropy admits a flat derivative. Their argument relies on the approximation $\log(1+t) = t + o(t)$ as $t \to 0$, and it implicitly invokes the dominated convergence theorem without verifying its applicability, i.e., for one of the terms in their proof it reads
> \begin{equation*}
> \lim_{\epsilon \to 0} \int \rho \log\left(1+\epsilon\frac{\rho''}{\rho}\right)
> = \int \rho  \lim_{\epsilon \to 0} \log\left(1+\epsilon\frac{\rho''}{\rho}\right).
> \end{equation*}
> To apply the dominated convergence theorem, one must ensure that the integrand is dominated by an integrable function independent of $\epsilon$. Since both (Hsieh et al., 2019) and our work operate under the TV norm, where test functions are bounded and continuous, it suffices to assume that at every step of the algorithm the logarithmic term inside the integral is uniformly bounded in $\epsilon.$ This requirement is reflected by the condition stated in our Example 1.3.
>
> The issue of flat differentiability of the entropy is addressed in (Kerimkulov et al., 2025, Proposition 2.17), which shows that the entropy admits a flat derivative only for measures in the admissible class $\mathcal{E}.$ A related observation appears in [Remark 5, Aubin-Frankowski, Korba, Lejer. Mirror Descent with Relative Smoothness in Measure Spaces, with application to Sinkhorn and EM]: A sufficient condition for the relative entropy (or the entropy) to admit a flat derivative in $L^{\infty}$ at $\mu$ is that there exists $\kappa_0, \kappa_1 > 0$ such that $\kappa_0 \leq \frac{\mathrm{d}\mu}{\mathrm{d}\pi} \leq \kappa_1$ a.e. on $\mathcal{X},$ which is precisely an equivalent condition to the one defining the class $\mathcal{E}.$
>
> Of course, choosing $\pi$ as the reference measure is not necessary. We included it merely to demonstrate that Example 1.3 holds in a more general setting. One could just as well omit $\pi$ and define the entropy directly with respect to the Lebesgue measure.
>
> ### Lack of examples of divergence
> We agree with the reviewer that when we verify the assumptions for the GAN example and the NN example, we implicitly rely on the comment in Remark 2.2, namely that $h$ satisfies a Pinsker-type inequality but for Bregman divergences. We have made this more transparent in the revised version and state directly in the current Assumption 1.1 that $h$ is strongly convex with respect to $\operatorname{TV}^2,$ which essentially implies that $h$ satisfies the Bregman Pinsker-type inequality in the previous Remark 2.2. We verify that this inequality holds for relative entropy and for chi-squared divergence in Examples 1.3 and 1.4, respectively, and we also refer to other examples in [Lemma 3.2, Chizat. Convergence Rates of Gradient Methods for Convex Optimization in the Space of Measures].
>
> Two additional benefits arise from this modification. First, we no longer need to assume that $F$ is Lipschitz with respect to the Bregman divergence (Assumption 2.1 in the previous version). Second, we can now establish convergence of the simultaneous scheme without requiring relative smoothness of $F$. It suffices to assume boundedness of the flat derivatives of $F$. Moreover, the previous Assumption 3.4 (now Assumption 3.5) is verified for both the entropy and the chi-squared divergence.
>
> ### Minor
> 1. A table of contents and a description of the appendix have been added.
> 2. We agree that L260-267 are unnecessary and we decided to remove them.

---

### Official Review · Reviewer_kmEj · 2025-10-30

**Soundness:** 2
**Presentation:** 3
**Contribution:** 3
**Rating:** 6
**Confidence:** 4

**Summary:**

This paper studies the simultaneous and sequential mirror descent-ascent (MDA) algorithms in solving convex-concave min-max problems with infinite-dimensional action spaces.
For simultaneous MDA, they show that the averaged iterate converges at a rate of $O(1/\sqrt{N})$, where $N$ is the number of iterations.
For sequential MDA, they show that the averaged iterate converges at a rate of $O(1/N^{2/3})$.
Both convergence rates match the known rates for the finite-dimensional bilinear case.
They also implement many numerical experiments to prove the efficiency of the algorithms in its applications, e.g., training GANs.

**Strengths:**

*   The paper seems to be the first paper to study the sequential MDA for games with infinite action spaces;
*   This paper establishes theoretical foundations for the faster convergence of the sequential MDA than its simultanuous counterpart;
*   The paper is fairly well-written and polished.

**Weaknesses:**

*   It looks like the new result on the sequential MDA is obtained by the parallel analysis in the discrete-case. It would be better if the authors can point out the main differences between the discrete-time and continuous-time proofs;
*   It seems that in the statement of their main theorems, they missed some key assumptions:
    *   It appears that they need the reference function $h$ to be Legendre, i.e., the norm of the derivative of $h$ goes to infinity as the iterate approaches the boundary. For example, they may need that assumption to get the first-order optimality conditions in Eq. (20). If correct, this is a big limitation that should be called out in many places. Currently, the paper repeatedly emphasizes being for "general Bregman divergences."
    *   The stepsizes need to be set depending on the total number of iterations $N$ for both algorithms, while it seems that they did not include that setup in the statements of Theorem 2.4 and Theorem 3.6.
*.  The GANs motivation does not feel very compelling. Having a probability distribution over network parameters feels highly unrealistic. How would you train this?

**Questions:**

*   Is my understanding of part 3 correct?
*  How do you imagine realistically training a probability distribution over parameters in a neural network?

Minor points:
*   In Assumption 1.5, it would be clearer if the authors clearify the notations $D_{F(\cdot, \mu)}(\nu', \nu)$ and $D_{F(\nu, \cdot)}(\mu', \mu)$. As $D_h$ is used to denote Bregman divergence in the paper, maybe they can consider use alternative notations for the second-order derivatives if possible;
*   The motivation of MDA formulation in Section 1.6 can be simplified in my opinion, as it aligns with the general gradient-based optimization methods;
*   Remark 2.5 is not entirely right in my opinion, since the usual $1/\sqrt{N}$ rate does not require relative Lipschitzness.
*   In much of the related literature, "sequential" algorithms are called "alternating" rather than "sequential".
*   Typos:
    *   p18 l929: due [to] Assumption 3.5 or [by] Assumption 3.5.
    *   Why are $\mu^\*,\nu^\*$ included in Theorem 2.4? As far as I can tell they play no role?

---

> ### Author Response · Authors · 2025-11-28
> **Response to the reviewer 1**
>
> We would like to thank the reviewer for appreciating the contribution of our paper, for their feedback and questions. We would like to address the weaknesses and questions formulated by the reviewer:
>
> ### Weaknesses:
> - In principle the analysis in continuous time can give us a general idea about which gradient flows converge to the minimizer/maximizer (and under what assumptions, what rate, etc.) and hence which gradient flows can be used as the basis for constructing discrete-time algorithms, but ultimately the analysis in discrete-time has to be carried out independently because it crucially depends on the choice of the discretisation (for example we can use mirror/Jordan-Kinderlehler-Otto (JKO)-type schemes or we can use interacting particle systems). In particular, our paper demonstrates that the choice of simultaneous vs. alternating mirror scheme is important.
>
> Restricting to our min-max games setup, the continuous time analysis typically reveals an "ideal" rate of convergence of gradient flow. By "ideal" we mean that this rate may not be attained by the discrete-time algorithms. This is now explained in Remark 1.7. In Appendix J, we provide a formal calculation showing that in continuous time we arrive at a $1/t$ convergence rate, which could only be attained by an implicit MDA scheme but which, as we mention there, is not implementable in practice because it requires both players to know the next move of their opponent.
>
> From a technical perspective, the continuous time analysis does not require the technical tools we need in discrete-time such as the three-point lemma and the analysis on the dual space. Thus making the analysis in discrete-time more technically challenging.
>
> - We thank the reviewer for raising the connection with Legendre-type (essentially smooth) mirror maps. In our setting, however, we explicitly assume that the Bregman potential $h$ is flat differentiable on the entire domains $\mathcal{C}$ and $\mathcal{D}$ (Assumption 1.1). This guarantees that $h$ is differentiable everywhere in all feasible directions.
>
> In contrast, the Legendre property ensures that minimizers remain in the interior of the domain, where $h$ is differentiable, thus avoiding boundary issues. For instance, Definition 1 (Legendre functions) in [Bauschke et al., 2017, A descent Lemma beyond Lipschitz gradient
> continuity: first-order methods revisited and applications] only requires $h$ to be differentiable on the interior of $dom(h)$. Consequently, they must additionally impose the condition mentioned by the reviewer to handle potential boundary complications. This is also reflected in their definition of the Bregman divergence (their Eq. (5)):
> \begin{equation*}
>     D_h(x,y) = h(x) - h(y) - \langle \nabla h(y), x-y\rangle, \forall x \in dom(h), \forall y \in int(dom(h)).
> \end{equation*}
>
> Our framework differs in that we assume $h$ is flat differentiable on the full domains $\mathcal{C}$ and $\mathcal{D}$, not only on their interiors. Hence, $D_h$ is well defined on the entire domain without restricting $y$ to lie in the interior. The key requirement for taking derivatives and writing optimality conditions is therefore to specify domains $\mathcal{C}$ and $\mathcal{D}$ on which $h$ is everywhere differentiable.
>
> To make this precise, we provide concrete examples for the relative entropy and the chi-squared divergence by identifying admissible classes $\mathcal{E}$ and $\mathcal{F}$ where $h$ is flat differentiable and $D_h$ is well defined. Moreover, Lemmas B.8 and B.9 show that the iterates of both the simultaneous and alternating schemes remain in these admissible classes at every step. Thus, for each $n \geq 0$, the flat derivative of $D_h(\nu, \nu^n)$ at $\nu=\nu^{n+1}$ can be computed as
> \begin{equation*}
>     \frac{\delta D_h(\cdot,\nu^n)}{\delta \nu}(\nu^{n+1},\cdot) = \frac{\delta h}{\delta \nu}(\nu^{n+1},\cdot) - \frac{\delta h}{\delta \nu}(\nu^{n},\cdot), \nu^{n+1}-a.e.
> \end{equation*}
> since $\frac{\delta h}{\delta \nu}(\nu^{n+1},\cdot)$ is well-defined within its admissible class.

---

> ### Author Response · Authors · 2025-11-28
> **Response to the reviewer 2**
>
> - The conditions on the stepsizes are now explicitly included in the statements of the theorems. Regarding training in the mean-field regime, our methodology aligns with the mean-field perspective on neural networks, in which models are described by probability distributions over their parameters [Chizat and Bach, 2018; On the Global Convergence of Gradient Descent for Overparameterized Models using Optimal Transport], [Mei, Montanari, Nguyen, 2018; A mean field view of the landscape of two-layer neural networks] and [Rotskoff and Vanden-Eijnden, 2018; Neural networks as interacting particle systems:
> Asymptotic convexity of the loss landscape and universal scaling of the approximation error]. These works show that training a one-hidden-layer neural network with a sufficiently large number of parameters can be interpreted as minimizing a functional defined on the space of probability distributions over the parameters.
> From the practical point of view, such infinite-dimensional optimization problems can be solved by a variety of methods, including interacting particle systems approximating mean-field Langevin dynamics as in Section 4 of our paper (see also [Hu et al.; Mean-Field Langevin Dynamics and Energy Landscape of Neural Networks], [Nitanda, Wu, Suzuki; Convex Analysis of the Mean Field Langevin Dynamics], [Chizat; Mean-Field Langevin Dynamics: Exponential Convergence and Annealing]).
>
> ### Questions:
> Please see our responses in the Weaknesses section.
>
> ### Minor points:
> - By definition of the Bregman divergence, the notations $D_{F(\cdot,\mu)}$ and $D_{F(\nu,\cdot)}$ precisely mean the Bregman divergences of the maps $F(\cdot,\mu)$ and $F(\nu,\cdot),$ respectively. In the revised version we wrote the explicit expressions to remove any confusion.
>  - We decided to remove the paragraph on the motivation of MDA due to space constraints after revision.
>  - We refined the assumptions and in the current version we obtain $1/\sqrt{N}$ rate for the simultaneous scheme under analogous assumptions to (Bubeck 2015, Theorem 5.1).
>  - We have decided to follow the suggestion and changed the name from "sequential" to "alternating".
>  - Both typos are fixed now.

---

### Official Review · Reviewer_xMxA · 2025-11-04

**Soundness:** 3
**Presentation:** 3
**Contribution:** 3
**Rating:** 4
**Confidence:** 3

**Summary:**

This manuscript studies mirror descent/ascent algorithms for nonlinear convex–concave problems defined on the space of probability measures. The authors consider two variants of a saddle-point optimization algorithm: simultaneous and sequential. They leverage relative smoothness and Lipschitzness properties of the convex–concave objective to prove a $1/\sqrt{N}$ bound for the simultaneous algorithm. Additionally, they use Hessian Lipschitzness and an $L_{\infty}$ assumption on derivatives to improve the convergence rate to $1/N^{2/3}$ for the sequential algorithm. The results extend earlier results for bilinear objectives and can be extended to training mean-field neural networks.

**Strengths:**

- The paper is well-written.
- It tackles the technical arguments needed to extend optimization guarantees for bilinear objectives on probability spaces to nonlinear convex–concave objectives.
- It empirically verifies its theoretical results.

**Weaknesses:**

- The analysis assumes both Lipschitzness and smoothness, yet the convergence of the simultaneous algorithm is $1/\sqrt{N}$, which may be restrictive for nonlinear objectives. The cited analogous result (Bubeck 2015 Theorem 5.1) assumes only Lipschitzness. That said, the authors clearly explain the proof bottleneck, which I appreciate.
- The implementation requires an internal sampler to trace the algorithm’s trajectory, but the approximation error introduced by this sampler is not characterized.

**Questions:**

Do you think it is possible to derive a $1/N$ rate in the relative-smoothness case using an algorithm similar to (Bubeck 2015, Section 5.2.3)? What are the main technical challenges? If not, is there an impossibility result—e.g., a matching lower bound—establishing that faster rates are unattainable under these assumptions?

---

> ### Author Response · Authors · 2025-11-28
> **Response to the reviewer**
>
> We would like to thank the reviewer for appreciating the contribution of our paper, for their feedback and questions. We would like to address the weaknesses and questions formulated by the reviewer:
>
> ### Weaknesses:
> - We agree with the reviewer that, in the simultaneous scheme (the analogue of the algorithm in Bubeck 2015, Theorem 5.1), our original analysis used both the Lipschitzness and the relative smoothness of $F$ with respect to the Bregman divergence. This discrepancy between our result and Bubeck's theorem occurred because Bubeck’s theorem assumes the mirror map $h$ is strongly convex, whereas we had only assumed strict convexity. In the revised version of the paper, we decided to impose the assumption that $h$ is strongly convex with respect to $\operatorname{TV}^2$ (Assumption 1.1) since this is naturally satisfied anyway for all typical examples of $h$ (cf. the discussion in Section 1.5). Under this stronger convexity assumption, we can establish convergence of the simultaneous scheme assuming only that the flat derivatives of $F$ are uniformly bounded and that $F$ is convex–concave (i.e., we no longer need the relative smoothness).
>
> - We agree that this is an important question, however, we believe that it will require a substantial amount of additional work and hence it falls beyond the scope of the present paper. We are planning to carry out a full theoretical analysis of the algorithm from Section 4 in a future work - this will require combining the results from the present paper with a careful analysis of the approximation error. The latter would be essentially a propagation-of-chaos-type result, showing that the empirical measures for the interacting particle systems from Section 4 converge to the corresponding (simultaneous or alternating) infinite-dimensional MDA dynamics. To the best of our knowledge, such results are currently unavailable in the literature for the dynamics studied in our paper.
>
> ### Questions:
> Extending our analysis to the Mirror Prox scheme is not straightforward due to the following technical issue. To replicate the assumptions on the objective used in Bubeck (2015, Section 5.2.3), one would need to impose the following $\operatorname{TV}$-Lipschitz condition on $F$. Specifically, there exist $C_{11}, C_{12}, C_{21}, C_{22} > 0$ such that
> \begin{equation*}
>     \left|\frac{\delta F}{\delta \nu}(\nu',\mu,x') - \frac{\delta F}{\delta \nu}(\nu,\mu,x)\right| \leq C_{11}(\operatorname{TV}(\nu',\nu) + |x'-x|),
> \end{equation*}
> \begin{equation*}
>     \left|\frac{\delta F}{\delta \nu}(\nu,\mu',x') - \frac{\delta F}{\delta \nu}(\nu,\mu,x)\right| \leq C_{12}(\operatorname{TV}(\mu',\mu) + |x'-x|),
> \end{equation*}
> \begin{equation*}
>     \left|\frac{\delta F}{\delta \mu}(\nu',\mu,y') - \frac{\delta F}{\delta \mu}(\nu,\mu,y)\right| \leq C_{21}(\operatorname{TV}(\nu',\nu) + |y'-y|).
> \end{equation*}
> \begin{equation*}
>     \left|\frac{\delta F}{\delta \mu}(\nu,\mu',y') - \frac{\delta F}{\delta \mu}(\nu,\mu,y)\right| \leq C_{22}(\operatorname{TV}(\mu',\mu) + |y'-y|).
> \end{equation*}
> This $\operatorname{TV}$-Lipschitz assumption is substantially stronger than the relative smoothness condition (Assumption 3.6 in the revised version, which is now needed only for the alternating scheme but not for the simultaneous one). For instance, the bilinear game $$F(\nu, \mu) = \int \int f(x,y)\nu(dx)\mu(dy)$$ satisfies Assumption 3.6 for $L_\nu = L_\mu = 0,$ yet it is $\operatorname{TV}$-Lipschitz only when $f$ is both bounded and Lipschitz. Thus, showing that Mirror Prox achieves an $1/N$ convergence rate under the weaker relative smoothness assumption remains a challenging open question.
>
> As a side remark, we added a few comments on the comparison between mirror descent and Mirror Prox (cf. Appendix K). Mirror Prox is computationally more expensive than mirror descent since it requires two mirror descent steps per player per iteration, effectively doubling the number of gradient evaluations. Although it achieves a $1/N$ convergence rate with deterministic gradients, in the stochastic case it attains only the same $1/\sqrt{N}$ rate as simultaneous MDA. Since the stochastic setting is precisely the one arising in the particle-based algorithmic implementation, even if the theoretical infinite-dimensional Mirror Prox analysis yielded a $1/N$ rate, the corresponding particle system would still converge at best at $1/\sqrt{N}$. Finally, we note that the convergence analysis of an alternating Mirror Prox scheme remains, to our knowledge, open. Designing how to alternate the four mirror descent steps in a way that improves convergence rates appears to be technically delicate. We view this as another potential direction for future work.

---

### Author Response · Authors · 2025-11-28
**Meta-response to the reviewers**

We apologize for the delay in providing our rebuttal. We required additional time to carefully address and resolve the technical issues raised by reviewer pJx5.

We would like to summarize the main revisions made to the paper:
- **Strengthened convexity assumption.** We now assume that $h$ is strongly convex relative to $\operatorname{TV}^2$ (and verify this for common choices of $h$), rather than strictly convex as in the previous version. Consequently, our main result for simultaneous schemes (Theorem 2.1) now does not require to assume: (i) smoothness of the objective $F$ relative to $D_h$ and (ii) Lipschitzness of $F$ relative to $D_h$.
- **Additional verification of assumptions on $h$.** Our assumptions on $h$ (Assumption 1.1 and 3.5) are now verified not only for the entropy but also for another commonly used choice, the chi-squared divergence. In addition, Assumption 1.1 is showed to hold for a broader class of divergences.
- **Additional context on convergence rates.** In Remark 1.7 and Appendix J, we expand our discussion of the convergence rates for the simultaneous and alternating schemes by comparing them with an implicit discretization and we describe the connection among all three discretizations (simultaneous, alternating, implicit) and the continuous-time Bregman gradient flow.
- **Correction of Assumption 3.4.** Following reviewer pJx5’s observation that the previous Assumption 3.4 does not hold for entropy, we replaced it with the new Assumption 3.5. We rewrote Section 3 using Fréchet derivatives of the dual map (instead of first/second order variations) and proved that Assumption 3.5 holds both when $h$ is the entropy and when $h$ is the chi-squared divergence.
- **Expanded proof sketches.** We added more detailed proof sketches for the main results in Sections 2.1 and 3.1.
- **New application.** In Appendix E, we introduce an additional application to two-player zero-sum Markov games and verify that it satisfies our assumptions.
- **Numerical experiments relocated.** Due to space constraints after the revision, the entire numerical experiments section has been moved to Appendix F.

---

### Meta-Review · Area_Chair_KCSU · 2026-01-17

**Summary:**

This paper analyzes mirror descent-ascent (MDA) for solving min-max problems on the space of measures. The goal is to be able to handle nonlinear convex-concave coupling function $F$ as well as the case when the function $h$ generating the Bregman divergence is not necessarily the negative entropy. The authors show $O(1/\sqrT{T})$ for the simultaneous MDA and $O(1/T^{2/3})$ for the alternating MDA. The authors initially required strict convexity from $h$ and smoothness and Lipschitzness from $F$. The authors later changed their setting after the technical issues pointed out by Reviewer pJx5. I think that the changes proposed in the rebuttal are major and hence require a new round of review. As a result, the positioning compared to Hsieh et al., 2019 requires a new explanation with the new setup. Moreover, the fact that current submission cannot handle Mirror-Prox, unlike Hsieh et al., 2019 needs to also be explained in a better way compared to Appendix K (Indeed, the fact that stochastic MP and MDA get similar rates does not mean that getting a fast rate is MP is not desirable).

**Reviewer Concerns:**

Main concerns by Reviewer pJx5 were that the authors didn't have examples apart from the negative entropy case for their Assumptions 2.1 and 3.4. This reviewer also pointed out that Assumption 3.4 may not be satisfied for the case when $h$ is negative entropy and that the given reference (Lascu et al., Lemma A.2) did not contain the claimed proof. The authors then changed their assumptions (added Assumption 3.5) and showed that chi-squared divergence satisfies this new assumption. Even though these changes may address the concerns, they are rather major and require a new round of review.

Moreover, as pointed out by Reviewer xMxA, the submission cannot handle Mirror-Prox to obtain a faster rate. This is in contrast to Hsieh et al., 2019. The explanation in Appendix K is not convincing regarding the lack of such a result in the current submission. The text in the rebuttal is better, yet does not completely address the concern for the lack of such a result.

**Reviewer Scores:**

I think that Reviewer 9vd3 would not have increased their score because their concerns are addressed by changing the assumption and the main setup of the problem, which are major changes not suitable for the discussion period.

I think that Reviewer xMxA would not have increased their score as well, since to address their concern about the assumptions on $F$, the authors introduced a new set of assumptions. Moreover, an improved rate with MP seems to be out of reach.

I think Reviewer kmEj and dZnt would keep their weak accept recommendations.

---

### Decision · Program_Chairs · 2026-01-26

Reject